*Resource*

# Global analysis of cancer cell responses to USP9X inhibition

Philipp Schenk [1,2], Shane M Devine [1,2], Simon A Cobbold [1,2], Niall D Geoghegan[1,2], Elizabeth L Kyran[1,3,4], Ching-Seng Ang[5], Jack A Alexandrovics[1,2], Dale J Calleja [1,2], Dylan H Multari [1,2], Vineet Vaibhav[1,2], Bernadine G C Lu [1,2], Theresa A Klemm [1,2], Laura F Dagley[1,2], Kym N Lowes[1,2], Nicholas A Williamson [5], Pieter J A Eichhorn [6,7], Ashley P Ng [1,2,8,9], Rebecca Feltham [1,2] & David Komander [1,2 ✉]

## Abstract

The ubiquitin-specific protease (USP) USP9X is a human deubiquitinase (DUB) with a large number of described targets and cellular roles. In cancer, USP9X is found as an oncogene or as a tumour suppressor depending on context, and its utility as a target for cancer therapy remains unclear. We here describe WEHI-092, a piperazine-based USP9X-specific small-molecule inhibitor, which binds to a unique region in the USP9X Fingers-subdomain, distinct from known DUB-inhibitor binding sites. Using proteomics and ubiquitinomics, we show that USP9X targets distinct substrates compared to USP7, yet the substrate profile of USP9X varies significantly across cancer cell lines. We reveal a core set of 17 proteins commonly regulated by USP9X in most cell lines, which we consider as proximal biomarkers for USP9X inhibition. Consistent with proteomics, we show in unrelated cell lines that WEHI-092 treatment arrests the cell cycle in metaphase without inducing cell death. This explains growth suppression in long-term clonogenic assays in most cancer cell lines, and positions USP9X inhibitors as a new class of selective mitotic poisons.

**Keywords** USP9X; DUB Inhibitor; Cancer; Ubiquitinomics; Substrate Identification
**Subject Categories** Pharmacology & Drug Discovery; Post-translational Modifications & Proteolysis

## Introduction

Protein ubiquitination is a key regulator of protein homeostasis, and as such has many recognised roles in human disease settings, in particular cancer (Deng et al, 2020). The attachment of ubiquitin to proteins most commonly triggers their degradation through the ubiquitin proteasome system (UPS), which facilitates the clearance of old and damaged proteins but can also regulate cellular cascades such as the cell cycle (Hershko, 1997). Overall, more than 5% of the human genome encodes for proteins that attach, bind, or remove a vast array of distinct and dynamic ubiquitin modifications (Agrata and Komander, 2025). Within these proteins, more than 700 E3 ligases facilitate ubiquitination in conjunction with E1 activating and E2 conjugating enzymes, while ~100 deubiquitinases (DUBs) remove ubiquitin signals from proteins (Clague et al, 2019).

Many oncogenes are under the control of the UPS, and are ubiquitinated and degraded, often in a highly regulated fashion. The UPS is exploited pharmacologically, most prominently via targeted protein degradation (TPD) approaches using proteolysis-targeting chimeras (PROTACs) or molecular glues (Békés et al, 2022; Zhao et al, 2022). Some of the most desired cancer targets, however, remain undruggable due to a lack of efficacious small-molecule binders. An orthogonal way to drive an oncoprotein towards degradation can be achieved through inhibition of the DUB that stabilises the oncoprotein (Harrigan et al, 2018). Indeed, hundreds of papers report deregulated DUB activity in cancer, where DUB deregulation stabilises oncoproteins or destabilises tumour suppressors (Dewson et al, 2023). Hence, DUB inhibitors provide an orthogonal strategy to induce oncoprotein degradation.

Ubiquitin-specific protease (USP) DUBs have received most pharmacological attention. USP DUBs are typically promiscuous regarding the ubiquitin signals they remove but achieve target specificity via substrate-binding domains (Mevissen and Komander, 2016). The more than 50 human enzymes in the family comprise several known oncogenes (Dewson et al, 2023). However, drug discovery for USP DUBs has proven challenging, and truly specific USP inhibitors have only been reported in the last decade, for less than ten USP enzymes (reviewed in Schauer et al, 2020; Liu et al, 2025). A fascinating diversity of inhibitory mechanisms has been revealed in structural studies (Kazi et al, 2025).

[1]The Walter and Eliza Hall Institute for Medical Research, Parkville, VIC, Australia. [2]Department of Medical Biology, University of Melbourne, Parkville, VIC, Australia. [3]Cancer Research UK Cambridge Institute, Cambridge, UK. [4]School of Clinical Medicine, University of Cambridge, Cambridge, UK. [5]Bio21 Molecular Science and Biotechnology Institute, University of Melbourne, Parkville, VIC, Australia. [6]Curtin Medical School, Curtin University, Bentley, WA, Australia. [7]Curtin Health Innovation Research Institute, Curtin University, Bentley, WA, Australia. [8]Clinical Haematology Department, The Royal Melbourne Hospital and Peter MacCallum Cancer Centre, Parkville, VIC, Australia. [9]Sir Peter MacCallum Department of Oncology, The University of Melbourne, Parkville, VIC, Australia. ✉E-mail: dk@wehi.edu.au

USP9X is amongst the best-studied USP enzymes. More than 400 papers have focused on this enzyme, revealing roles in DNA damage response, cell cycle regulation, ribosomal quality control, endosomal trafficking, TGFβ and Hippo signalling, and others (reviewed in Murtaza et al, 2015; Gao et al, 2024). USP9X is essential for organism (especially brain) development (Pantaleon et al, 2001; Stegeman et al, 2013) as originally described in the *fat facets* phenotypes in Drosophila (Fischer-Vize et al, 1992) and mouse (Wood et al, 1997). Recent papers have focused on the roles of USP9X in cancer, where conflicting literature has shown USP9X to act as an oncoprotein or as a tumour suppressor in different cancer types. An oncogenic role for USP9X has been reported broadly across many cancers, and since many putative USP9X substrates are interesting pharmacological targets (such as MCL-1, β-catenin, ERG), USP9X became a promising pharmaceutical target itself (Taya et al, 1999; Schwickart et al, 2010; Wang et al, 2014; Lu et al, 2019). Conversely, USP9X was reported as a frequently mutated gene in pancreatic ductal adenocarcinoma (PDAC; Pérez-Mancera et al, 2012), and other human cancers (Hunter et al, 2015; Cheasley et al, 2021; Sisoudiya et al, 2023), suggesting tumour suppressor roles. Indeed, several putative USP9X substrates are tumour suppressors (LATS2, YAP1, FBW7; Toloczko et al, 2017; Li et al, 2018; Khan et al, 2018; Zhu et al, 2018). While the tumour suppressor role of USP9X in PDAC has been challenged (Cox *et al*, 2014a; Pal et al, 2017; Liu et al, 2017), these contrasting roles questioned the validity of USP9X as a cancer target and reinforced the need to define the molecular landscape and clinical context in which targeting USP9X would be most effective.

Discrepancies in the literature likely reflect an overemphasis on single-protein relationships (i.e. USP9X regulates a distinct target), which was sometimes exaggerated by inappropriate tools. Notably, more than 30 papers have assigned the effects of the small-molecule inhibitor Degrasyn/WP1130 and its derivatives G9/EOAI3402143 (Bartholomeusz et al, 2007a, 2007b) to USP9X inhibition, despite data showing that WP1130 is a non-specific DUB inhibitor (Kapuria et al, 2010; Ritorto et al, 2014; Peterson et al, 2015).

We recently reported FT709 as a first, highly specific USP9X inhibitor (Clancy et al, 2021). While its molecular mechanism of action has remained unclear, FT709 revealed a role for USP9X in ribosomal quality control in a cancer cell line (Clancy et al, 2021), but failed to confirm many previously reported USP9X substrates.

Here, we report WEHI-092 as a next-generation USP9X inhibitor. The simplified chemical scaffold retains excellent USP9X specificity, which we structurally explain through hydrogen-deuterium exchange mass spectrometry (HDX-MS) and mutagenesis studies. WEHI-092 is highly potent in cells, broadly arrests cell growth in a panel of cancer cell lines, and is strikingly selective in inducing cell death in one of 57 cell lines. Unbiased ubiquitinomics and proteomics experiments across four and eight cell lines, respectively, unveil novel insights into USP9X biology. We identify a set of high-confidence USP9X-regulated proteins across all cell lines studied, representing potential biomarkers for USP9X inhibition. Surprisingly, USP9X inhibition regulates a distinct set of targets in each cell line, and such pleiotropic cell-type-specific effects may explain discrepancies in the literature. Importantly, collective analysis of all proteomic data uncovers commonalities, showing that in multiple unrelated cell lines, USP9X inhibition leads to mitotic metaphase arrest without cell death, explaining growth retardation phenotypes.

# Results

## Identification of a USP9X-specific small-molecule inhibitor

While the USP9X inhibitor FT709 demonstrated potency in vitro and in cellulo, it bears a challenging chemical structure with a central symmetric bicyclic core, resulting in a complex synthesis route and high cost of production (Follows et al, 2020). During our search for DUB inhibitors within our chemical libraries, we identified WEHI-553 (Appendix Fig. S1A) as a compound that inhibited the purified catalytic domain of human USP9X (residues 1551–1970) in vitro with an $IC_{50}$ of 4.2 μM in a ubiquitin-rhodamine (Ub-Rho) based screening campaign (Klemm et al, 2020; Appendix Fig. S1B). WEHI-553 was structurally similar to FT709 but features a simpler and more chemically adaptable piperazine group as a central core. Structure-activity relationship (SAR) studies improved the $IC_{50}$ towards USP9X >16-fold with the best compound, WEHI-092 (Fig. 1A; Appendix Fig. S1A), displaying an $IC_{50}$ of 254 nM, compared to 124 nM for FT709 (Fig. 1B). An additional methyl group on the indole nitrogen (WEHI-333) was tolerated ($IC_{50}$ of 382 nM), whereas the removal of both methyl groups on the indole ring in WEHI-680 resulted in a >15-fold loss of activity in comparison to WEHI-092 ($IC_{50}$: 4.3 μM). A negative control compound, WEHI-871 ($IC_{50}$: >100 μM), was also included, which lacked the indole amide portion but retained the 2,3-dihydrobenzodioxine sulfonyl piperazine pharmacophore (Appendix Fig. S1A,B). Direct compound binding to the USP9X catalytic domain was measured by surface plasmon resonance (SPR) and revealed a $K_D$ of 69 nM for FT709 vs. 1 μM for WEHI-092 (Fig. 1C; Appendix Fig. S1C,D). Incubation of the USP9X catalytic domain with 50 μM WEHI-092 inhibited cleavage of K48-linked di-ubiquitin (Fig. 1D) and prevented the modification of USP9X with ubiquitin propargylamide (Ub-PA), a suicide substrate (Ekkebus et al, 2013) that covalently modifies the catalytic Cys1566 of USP9X (Fig. 1E). Comparison with reported inhibitors for USP9X in this assay, showed no or weak effects of WP1130 and EOAI3402143 on the purified USP9X catalytic domain (Appendix Fig. S1B,E).

We next tested compound specificity using a commercial screen against a human DUB panel (see "Methods"). At 50 μM compound concentration, WEHI-092 did not affect any of 49 human DUBs with the exception of USP9X (Fig. 1F). Compound specificity was further corroborated in cell lysates. A triple-FLAG ubiquitin-vinyl sulfone (3X-FLAG Ub-VS) suicide probe (Borodovsky et al, 2001), incubated with MCF-7 cell lysate, immuno-precipitated 51 human DUBs, which were identified and quantified by mass-spectrometry (MS). A pre-incubation of the lysate with untagged Ub-VS revealed catalytically active DUBs in the lysate, and their probe-unresponsive background levels. Out of the 51 identified DUBs, 23 responded to Ub-VS pre-incubation, suggesting that they were catalytically competent. Within these, only USP9X was affected by WEHI-092 treatment (Fig. 1G); the compound reduced activity levels of USP9X to a comparable degree as pre-incubation with unlabelled Ub-VS probe, suggesting WEHI-092 had inhibited all active USP9X in the lysate (Appendix Fig. S1F).

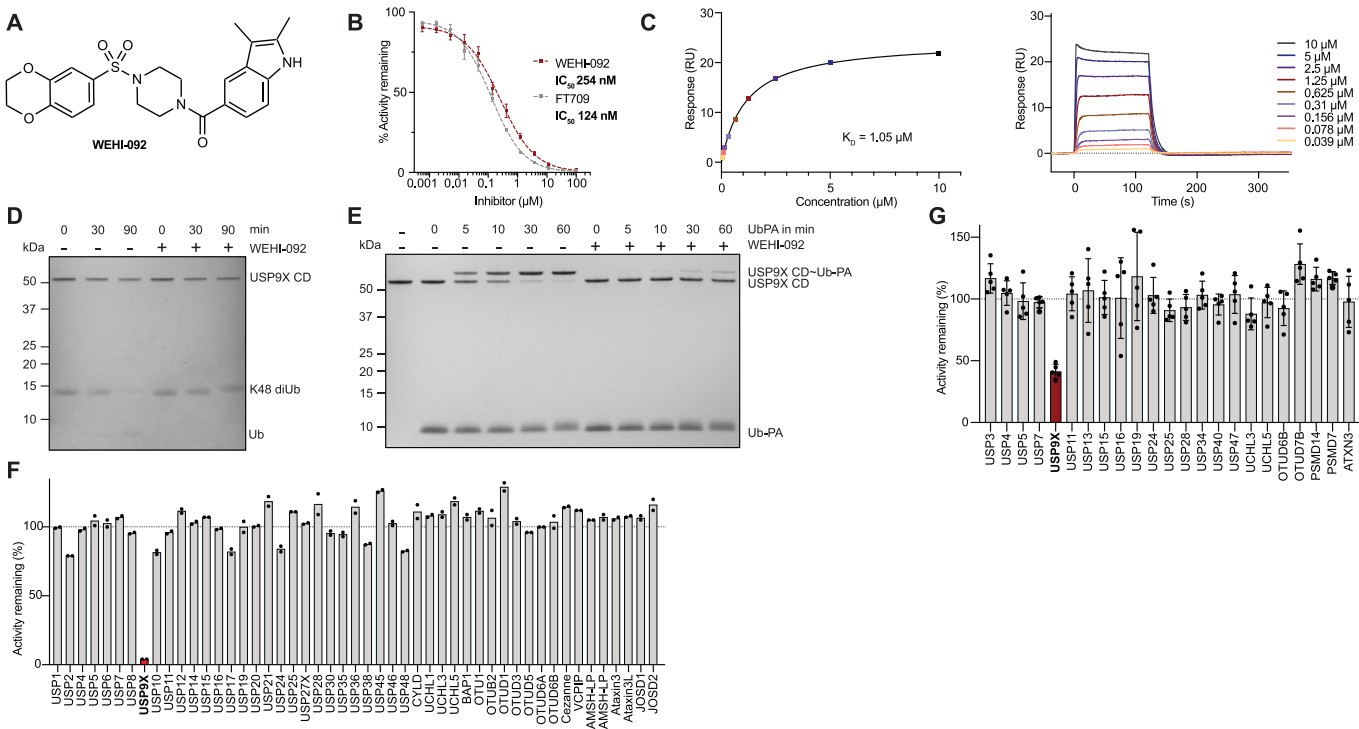

**Figure 1. Identification of WEHI-092 as a novel inhibitor for the deubiquitinase USP9X.**

(A) Chemical structure of WEHI-092. (B) WEHI-092 and FT709 activity against recombinant USP9X catalytic domain determined from Ub-Rho cleavage assay. The data shown are the mean of four independent biological repeats with two technical replicates per experiment. The curve shown is the nonlinear curve fit generated using GraphPad Prism (v10.3) and was used for calculating the $IC_{50}$ values. Error bars, s.e.m. (C) Direct binding of WEHI-092 to the USP9X catalytic domain was measured via SPR in steady-state. *Left*, fitted binding curve. Right, raw sensorgram data. $K_D$ shown is the mean of three independent biological repeats. Graphs shown are representatives from one experiment. (D) Coomassie gel of a di-ubiquitin (diUb) cleavage assay using K48-linked diUb with 50 μM WEHI-092. Representative gel of three independent biological repeats. (E) Coomassie gel of a ubiquitin-propargylamide (Ub-PA) suicide probe competition assay with 50 μM WEHI-092. Representative gel of three independent biological repeats. (F) DUB selectivity panel for WEHI-092 (50 μM) against recombinant protein performed at Ubiquigent™ (Dundee, UK). The data shown are the mean of two technical replicates, which are shown as individual datapoints. (G) DUB IP-MS selectivity panel for WEHI-092 (50 μM) in MCF-7 cell lysates. 3X-FLAG-Ub-VS labelled DUBs were subjected to quantification by MS, and % activity remaining was calculated from protein intensities relative to the untreated control condition. DUBs shown have been pre-filtered to respond to Ub-VS probe pre-incubation. Data shown is the mean of four or five biological repeats per condition, which are shown as individual datapoints; error bars, s.d. USP9X (red) was the only significantly altered (empirical Bayes moderated *t* test; adjusted *P* value < 0.05) hit in our analyses between untreated and WEHI-092 treated lysates. Source data are available online for this figure.

## WEHI-092 binds within the Fingers subdomain of USP9X

In the absence of mechanistic data for compound binding and specificity, we set out to understand where USP9X binds WEHI-092 and FT709. We used HDX-MS to study time-dependent changes in hydrogen accessibility upon compound binding. Even at the shortest deuterium exchange timepoints (6 s), compound incubation led to protection of five overlapping peptides that included residues 1757-1762 (Fig. 2A). This indicated that amide protons were shielded from solvent exchange, likely due to compound interaction. These changes were more pronounced at later timepoints of the assay (Fig. 2B); only after 10 min deuterium exchange were changes in additional regions of USP9X detected (Appendix Fig. S2A). Similar results were obtained for FT709, suggesting that both scaffolds utilise the same binding site (Appendix Fig. S2B).

Several structures of USP9X have been reported, including a crystal structure of the isolated catalytic domain (Paudel et al, 2019). We modelled a ubiquitin-bound catalytic domain USP9X using AlphaFold3 (Abramson et al, 2024; Fig. 2C) and mapped the

perturbed peptide motif onto this structure (Fig. 2D). Interestingly, the affected peptide of USP9X lies in the Fingers subdomain of the enzyme that is remote from the active site. USP catalytic domains have a characteristic 'right-hand'-like fold, in which the ubiquitin is clasped between Fingers and Palm subdomains and cleaved within a groove provided between Palm and Thumb subdomains (Hu et al, 2002; Ye et al, 2009). All known selective USP DUB inhibitors bind residues within 15 Å on the catalytic Cys, whereas the closest distance between a residue perturbed by WEHI-092 and USP9X catalytic Cys1566 is 26 Å (Appendix Fig. S3A,B). Moreover, the perturbed peptide is located at the edge of the Fingers subdomain's β-sheet, without forming a notable pocket or groove, yet some residues are in hydrogen binding distance with ubiquitin (Fig. 2E). Additional peptides within the Fingers subdomain appear to be deprotected at later timepoints, potentially suggesting long-range compound-induced conformational changes (Appendix Fig. S2A). Biochemical experiments (Fig. 1) show that compound binding affects ubiquitin interactions, which would be consistent with WEHI-092 binding to the Fingers domain as shown through an AlphaFold3 prediction with ubiquitin bound (Fig. 2E).

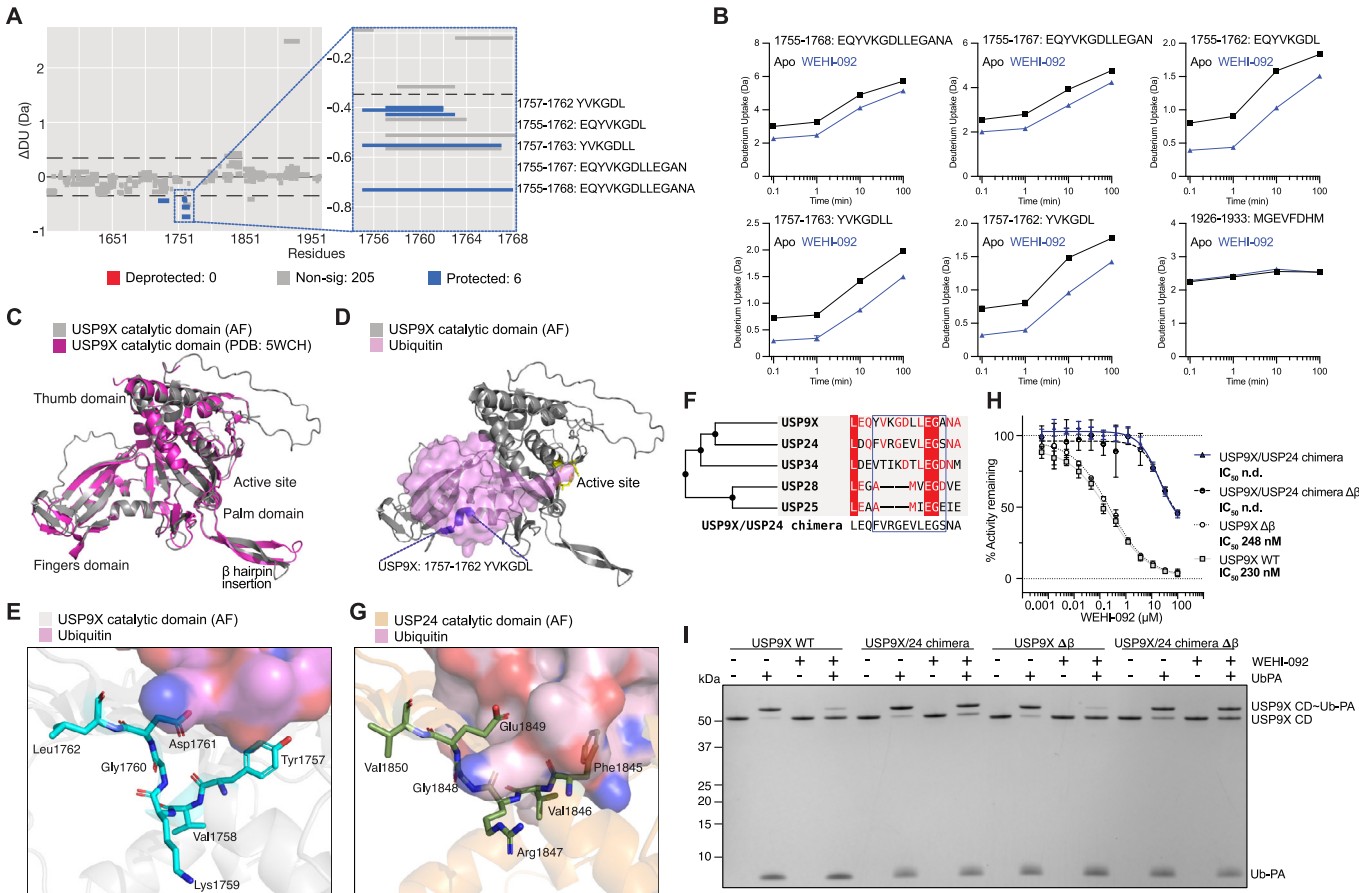

**Figure 2. Molecular basis of WEHI-092 specificity for USP9X over other closely related USPs.**

(A) Hybrid Woods differential plot highlighting significant differences (P < 0.01, in blue) in hydrogen-deuterium uptake between USP9X catalytic domain apo and WEHI-092 bound protein, for 6 s timepoint. The zoomed-in section highlights overlapping peptides. Dashed lines indicate the confidence limit at 99% (hybrid significance test, significance confirmed by a Welch's t test). Plots were generated using Deuteros software (v2.0). (B) Mean relative deuterium uptake plots for significant (Welch's t test; P < 0.01) peptides of USP9X catalytic domain apo (black) and WEHI-092 bound (blue) from (A), including a control peptide (β-hairpin loop, 1926–1933, residues MGEVFDHM). Error bars, s.d. from n = 3 (technical replicates) experiments. (C) Structure of the USP9X catalytic domain (PDB: 5WCH, shown in pink) and AlphaFold3 prediction (shown superimposed in grey). (D) Mapping of the shortest peptide from (A), YVKGDL (in blue), on the USP9X catalytic domain, with ubiquitin modelled as a pink transparent surface from an AlphaFold3 prediction. Active site residues are shown in yellow. (E) Detailed view of the WEHI-092 binding site (residues YVKGDL), with nitrogen (blue) and oxygen (red) atoms, with transparent surface of modelled ubiquitin (pink carbon atoms). (F) Phylogenetic tree of the closest USP relatives to USP9X for the WEHI-092 binding region. (G) Detailed view as in (E) of the corresponding motif in the USP24 catalytic domain from an AlphaFold3 prediction. (H) Ub-Rho cleavage assay (as in Fig. 1B) with WEHI-092 against recombinant USP9X variants: USP9X wild-type (WT), USP9X/USP24 chimera (USP9X with five USP24-like mutations, see "Methods" and (F)), USP9X Δβ (USP9X β-hairpin deletion), and USP9X/USP24 chimera Δβ (USP24-like mutant with β-hairpin deletion). The data shown are the mean of three independent biological repeats with two technical replicates per experiment. IC50 values were calculated using GraphPad Prism (v10.3). n.d., not determined; error bars, s.e.m. (I) Ub-PA assay (as in Fig. 1E) using USP9X variants from (H) with and without WEHI-092. Representative gel of three independent biological repeats. Source data are available online for this figure.

We used phylogenetic analyses and mutagenesis to corroborate the compound binding site. USP9X has a close human paralogue, USP24, a less-well-studied enzyme (Wang et al, 2020; Rossio et al, 2024) with highly similar overall domain architectures, in which the catalytic domains are embedded in expansive helical repeat scaffolds (Appendix Fig. S2E). The catalytic domain shares a sequence identity of 47% (sequence similarity 61%) with a similar overall fold, also featuring the characteristic extended β-hairpin. USP24 was included in DUB specificity panels tested against WEHI-092, but was not inhibited by the compound, neither in the commercial panel nor in cell lysates (Fig. 1F,G). The perturbed peptide within USP9X (aa: YVKGDL) differs in sequence, but not in length in USP24 (aa: FVRGEV; Fig. 2F,G; Appendix Fig. S2C).

We therefore created a chimera, in which USP9X residues were replaced with USP24 residues in a 11-mer sequence, in a wild-type (WT) catalytic domain, and in a catalytic domain in which the extended β-hairpin—a unique feature of USP9X and USP24—was removed (USP9X Δβ). Individual mutations slightly reduced USP9X activity in a Ub-Rho assay as reported previously (Paudel et al, 2019), yet each variant remained robustly active (Appendix Fig. S2D). Importantly, while WT USP9X and USP9X Δβ were inhibited by WEHI-092, the chimera mutant catalytic domains remained fully active at concentrations up to 10 μM, and partially active even at the highest compound concentrations (Fig. 2H). The same results were obtained for FT709 (Appendix Fig. S2F), again suggesting that both compounds share identical binding sites. At

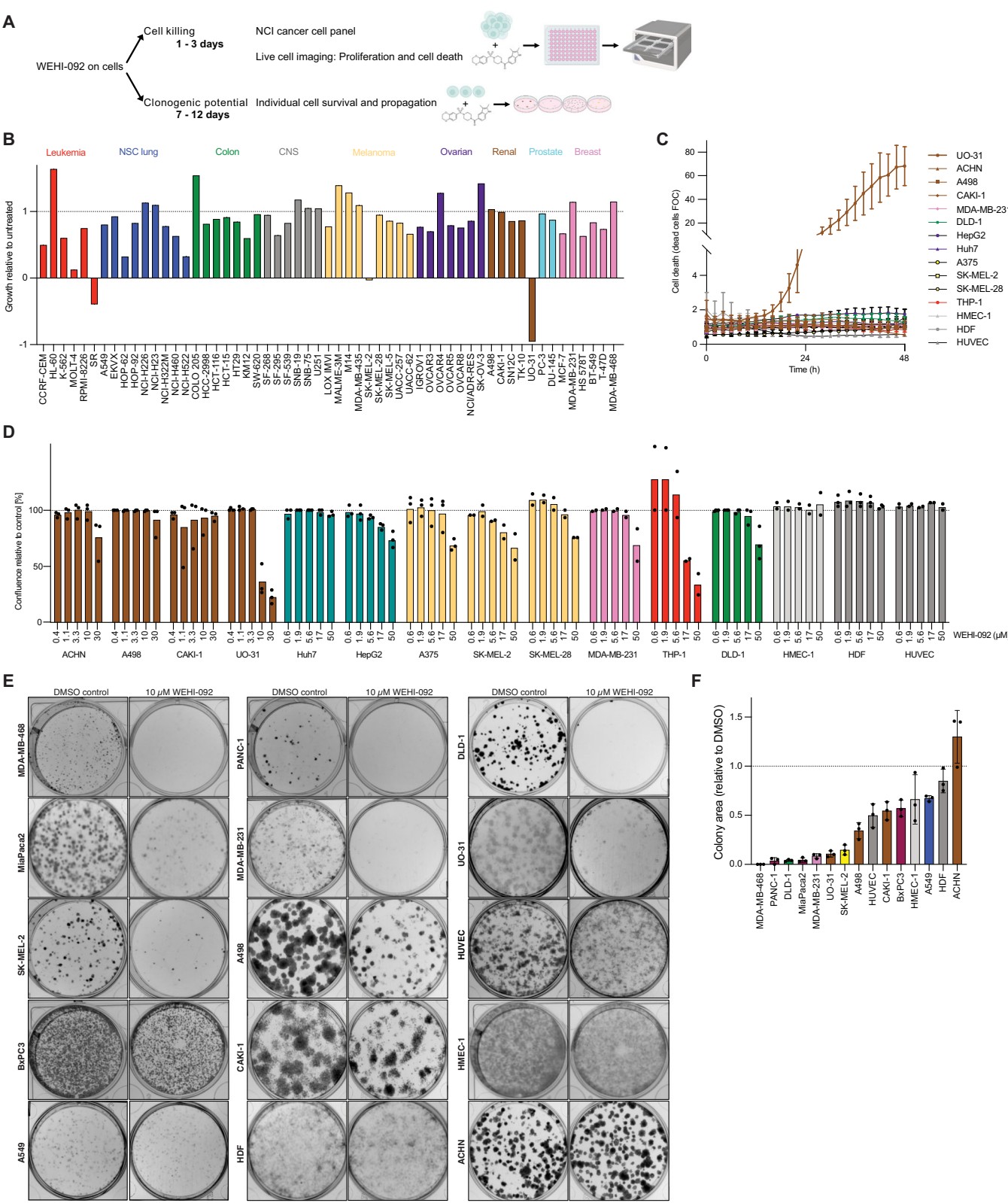

**Figure 3.   Treatment with WEHI-092 induces cell line-specific cell killing and clonogenic potential phenotypes.**

(A) Schematic of short-term cell killing and long-term clonogenic potential experiments. (B) NCI-60 cancer cell panel of 57 cancer cell lines treated with WEHI-092 (10 µM). Data shown are the normalised cell growth at 3 days post treatment relative to untreated control and relative to day 0 for each individual cell line. Negative values indicate cell death at 3 days post treatment. Cell lines are colour-coded depending on their cancer type. NSC - non-small cell. CNS - central nervous system. The data shown are the mean of two technical replicates. (C) IncuCyte live-cell imaging cell death data for cell lines treated with WEHI-092, used at 10 µM for UO-31, A498, ACHN, CAKI-1, HMEC-1, HDF cells and at 17 µM for the remaining cell lines. Cell death was measured by PI uptake. The data shown are the mean of at least two independent biological repeats with two technical replicates per experiment. FOC fold-over control, HDF human dermal fibroblasts. Error bars, s.e.m. (D) IncuCyte live-cell imaging proliferation data for cell lines treated with WEHI-092. Data shown is the mean of at least two independent biological repeats with two technical replicates per experiment at 72 h post treatment where each experiment is shown as a single datapoint. Data were normalised to the untreated condition. (E) Clonogenic potential assay for cell lines treated with WEHI-092. Representative images from three independent biological repeats. (F) Quantification of clonogenic potential assays, as shown in (E). Data shown are the mean of three independent biological repeats relative to the untreated condition (dashed line) for each cell line, where each separate experiment is shown as a single datapoint. Bars are colour-coded as in (B). Source data are available online for this figure.

50 µM compound concentration in a Ub-PA suicide probe assay, WT or USP9X Δβ catalytic domains were unable to be modified by the probe, while chimera mutants were able to bind and were modified by Ub-PA probes (Fig. 2I). Pinpointing a short stretch of amino acids to be involved in compound binding reveals a new USP9X compound binding site in the Fingers subdomain distinct from other selective USP DUB inhibitors (Appendix Fig. S3A,B).

## WEHI-092 is cytotoxic in a small subset of cancer cell lines

Next, we tested the inhibitory effects of WEHI-092 on the proliferation and survival of a panel of cancer cell lines. To ensure compound activity in cells, we tested a previously established proximal biomarker, the centrosomal protein CEP55 (Wang et al, 2017). CEP55 was destabilised by WEHI-092 in a dose-dependent fashion, and also by FT709 as reported previously (Clancy et al, 2021), with an apparent $IC_{50}$ (based on Western blotting) of 1.3 µM for WEHI-092 (Appendix Fig. S4A).

We assayed cell lines that included a broad range of cancer types, which were all found to express USP9X to varying degrees (Appendix Fig. S4B). To understand the impact of USP9X inhibition on cancer cell lines, we initially performed a series of cellular assays by treating cells with WEHI-092 (Fig. 3A). These included (i) short-term (3-day) cell killing/toxicity studies and (ii) short-term (3-day) cell growth/proliferation assessment, using the cancer cell line panel provided by the National Cancer Institute (NCI; Shoemaker, 2006) in which we tested WEHI-092 at 10 µM (a concentration at which clear target regulation was observed consistently, see below) against 57 human cancer cell lines across nine different cancer types (Fig. 3B). A tenfold less potent compound with similar chemical structure, WEHI-680, was tested for comparison (Appendix Figs. S1A and S4C). The results from the NCI panel were confirmed and expanded to additional cancer cell lines as well as non-cancerous cell lines (HUVECs and human dermal fibroblasts, HDFs), by performing in-house dose-escalation studies using IncuCyte live-cell imaging analysis, assessing cell death (Fig. 3C) and cell growth (Fig. 3D). These broad analyses of compound effects performed across multiple cell types revealed several interesting results. Firstly, WEHI-092 is non-lethal and non-toxic to the non-cancerous human cell lines studied (HUVECs and HDFs), a finding that was distinct from commonly used non-specific DUB inhibitors WP1130 and EOAI3402143, which were cytotoxic to HDFs (Appendix Fig. S4E).

Secondly, USP9X inhibition showed little to no effect in most cell lines tested in 3-day assays. With few exceptions, cell growth was only mildly impaired or not affected. However, in five cell lines across three cancer types, WEHI-092 caused more than 50% cell growth inhibition, including stalling cell growth completely in MOLT-4 and SK-MEL-2 cells (Fig. 3B). Thirdly, WEHI-092 caused cell death of the lymphoma line SR, and of the papillary renal cell carcinoma (pRCC) UO-31 cell line (Sinha et al, 2017), with the latter demonstrating unique sensitivity to WEHI-092. Consistently, UO-31 cells were the only cell line in the panel that was killed by WEHI-092 in IncuCyte assays (Fig. 3C). Other RCC cell lines, including clear cell renal cell carcinoma (ccRCC) lines but also a second pRCC line, ACHN, showed limited response to WEHI-092 treatment (Fig. 3C). Notably, UO-31 cells were highly sensitive to WEHI-092 but less sensitive to FT709 (Appendix Fig. S4F), with the slightly better in vitro activity of FT709 not translating to improved cellular activity (Appendix Fig. S4A). We performed a-Caco-2 permeability assessment of WEHI-092 and FT709 and demonstrated that although both compounds had high permeability in the basolateral-apical (B-A) direction, FT709 exhibited substantially lower permeability in the A-B direction compared to WEHI-092 (Appendix Fig. S4G). This reduced permeability results in an increased efflux ratio of 18 for FT709 as compared to 1.7 for WEHI-092 and could result in differences in cellular access between the two molecules, as is observed in the UO-31 cells.

Next, we assessed (iii) longer-term (7–12 days) effects of WEHI-092 on the clonogenic potential of cancer cells, either at a single compound concentration (10 µM, Fig. 3E,F) or in a dose-escalation study, the latter in comparison with WEHI-680 (Appendix Fig. S4D). As previously observed in the short-term assays, SK-MEL-2 and UO-31 cells were sensitive to WEHI-092 treatment (Fig. 3E). However, we also observed a strong decrease in proliferative capacity for the colorectal cancer cell line DLD-1, the pancreatic cancer cell lines MiaPaca2 and PANC-1, as well as breast cancer cell lines MDA-MB-231 and MDA-MB-468 (Fig. 3E,F). Other cancer cell lines tested, and also non-cancer cells (HUVEC and HDFs) showed limited or no effects in clonogenic potential, suggesting cell-type-specific effects of USP9X inhibition. This data suggested that extended treatment time exacerbates cell growth defects in cultured cell lines. As for the NCI panel, only 1 out of 15 cell lines (ACHN) showed an apparent increase in cell proliferation in colony formation assays (Fig. 3F). An analysis of the mutational status of the cancer cell lines tested revealed limited predictability of USP9X inhibitor sensitivity or

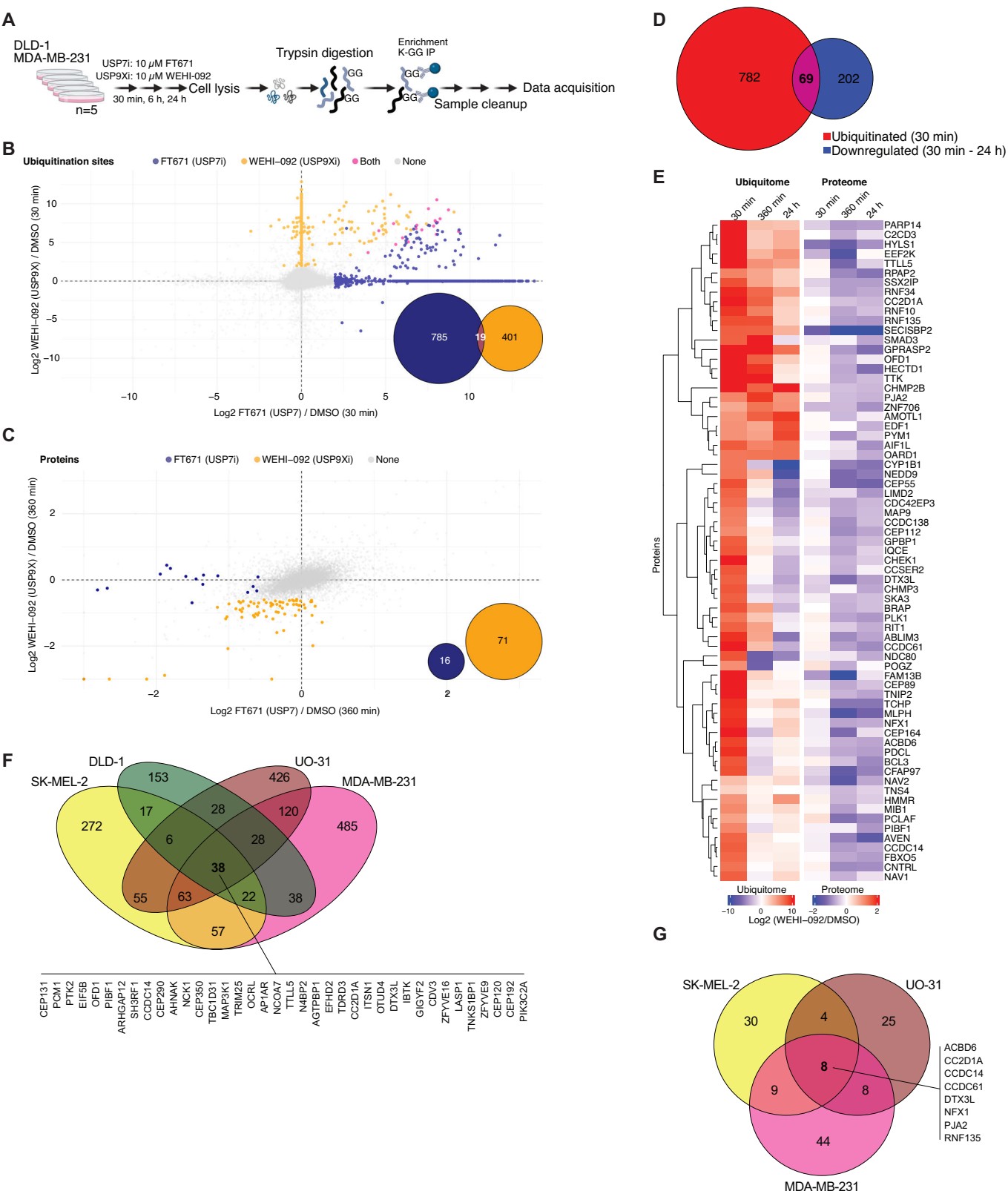

**Figure 4.  Time-resolved ubiquitome profiling reveals high-confidence USP9X substrates in a cell line-dependent manner.**

(A) Schematic overview of the ubiquitome profiling experiments. (B) USP9X vs. USP7 inhibition in DLD-1 cells treated with FT671 (USP7 inhibitor) or WEHI-092 at 10 μM for 30 min each. Each datapoint shows the mean log2 fold change for USP7i or USP9Xi treatment vs. control condition across four or five biological repeats per condition. Coloured datapoints show K-GG peptides with a fold change of log2 > 2 (adjusted *P* value < 0.05) over untreated control in blue for USP7i, in yellow for USP9Xi and in pink for both treatments. Empty values of non-detected peptides in one or the other condition were filled with zero values and are clustered on the *x* and *y* axis, respectively. The Venn diagram shows the number of K-GG peptides with a fold change of log2 > 2 (adjusted *P* value < 0.05) over the untreated control for the respective conditions. (C) USP9X vs. USP7 inhibition in DLD-1 cells treated with FT671 or WEHI-092 at 10 μM for 360 min. Each datapoint shows the mean log2 fold change upon USP7i or USP9Xi treatment vs. control condition across four or five biological repeats per condition. Coloured datapoints show proteins with a fold change of log2 < −0.585 (adjusted *P* value < 0.05) over untreated control in blue for USP7i, in yellow for USP9Xi. Non-detected proteins in one or the other condition were filled with zero values, and missing values were imputed using a mixed BPCA and Min method. Axis limits were set to −3/3, and datapoints outside of the range were plotted on the axis limit. The Venn diagram shows the number of downregulated proteins with a fold change of log2 < −0.585 (adjusted *P* value < 0.05) over the untreated control for the respective conditions. (D) Venn diagram showing the overlap of proteins with enhanced ubiquitination at 30 min (log2 >twofold, adjusted *P* value < 0.05 over untreated control) and decreased abundance at either 30 min, 360 min or 24 h WEHI-092 treatment (10 μM; log2 < −0.585, adjusted *P* value < 0.05 over untreated control) in MDA-MB-231 cells across four or five biological repeats per condition. (E) Time-resolved profiling of high-confidence USP9X substrates in MDA-MB-231 cells. Heatmap colours indicate fold change in protein ubiquitination (left) and protein expression (right) of proteins that showed significant induction (fold change log2 > 2, adjusted *P* value < 0.05 over untreated control) of at least one ubiquitination site at 30 min of WEHI-092 treatment (10 μM) and that were significantly downregulated (fold change log2 < −0.585, adjusted *P* value < 0.05 over untreated control) at 30 min/360 min/24 h. The data were averaged across four or five biological repeats per condition, and the data were matched based on gene ID level. Hierarchical clustering was performed on proteins (rows) with Euclidean distance as the similarity metric. (F) Comparison of ubiquitinated proteins (log2 >twofold increase over untreated control, adjusted *P* value < 0.05 across four or five biological repeats per cell line and condition) after 30 min of WEHI-092 treatment between MDA-MB-231, DLD-1, SK-MEL-2 and UO-31 cells. (G) Comparison of high-confidence USP9X substrates in SK-MEL-2, UO-31, and MDA-MB-231 cells. High-confidence USP9X substrates were identified in MDA-MB-231 cells (from (E)), in SK-MEL-2 cells (from Fig EV1A) and in UO-31 cells (from Fig. EV1B) and are defined as upregulated ubiquitination (fold change log2 > 2, adjusted *P* value < 0.05 over untreated control with four or five biological repeats per condition) and depleted protein levels (fold change log2 < −0.585, adjusted *P* value < 0.05 over untreated control with four or five biological repeats per condition). Source data are available online for this figure.

phenotypic response based on common cancer gene mutations (Appendix Fig. S4H).

## USP9X and USP7 have a non-overlapping set of substrates

The disparate rates of survival and cellular proliferation between cancer cell lines in response to WEHI-092 were unexpected and warranted a deeper investigation of USP9X and its regulatory roles, using a specific DUB inhibitor. We hence turned to comparative ubiquitinomics and proteomics studies to elucidate the effects of DUB inhibitors across cell lines. First, we sought to test whether different DUBs regulate the same set of ubiquitination sites within a given cell line, or whether they indeed exhibit substrate specificity, as is widely assumed in the field. To date, this has only been explored indirectly at the transcriptomic level using RNAseq, which suggested a non-overlapping set of substrates for USP7, USP14, and USP30 (Doherty et al, 2022).

Steger et al recently performed comprehensive proteomic and ubiquitinomic studies of several chemically distinct USP7 inhibitors, including the highly specific and potent FT671 (Turnbull et al, 2017), in a colorectal adenocarcinoma cell line (HCT116; Steger et al, 2021). Using FT671 in another colorectal carcinoma cell line (DLD-1) with identical concentration (10 μM), time points (30 min, 360 min), and liquid-chromatography (LC)-mass spectrometry (MS) acquisition, we detected 23,568 ubiquitinated GlyGly (K-GG) peptides that were reported by Steger et al (Fig. 4A). Filtering for K-GG peptides that were significantly increased during FT671 treatment in Steger et al revealed good quantitative reproducibility between the two studies (Appendix Fig. S5A,B), which was also apparent at the protein level for significantly reduced proteins following FT671 treatment (Appendix Fig. S5C).

Next, we treated DLD-1 cells with the USP9X inhibitor WEHI-092 at 10 μM. A distinct set of K-GG peptides were significantly increased with USP9X inhibition (using WEHI-092) as compared to USP7 inhibition (using FT671) at 30 min of treatment (Fig. 4B); 804 K-GG-modified peptides were enriched upon USP7 inhibitor treatment, 420 K-GG-peptides were enriched upon WEHI-092 treatment, and only 19 K-GG-peptides increased with either treatment (Fig. 4B). Moreover, there was no overlap in proteins that were significantly reduced in abundance between USP7 inhibitor treatment and USP9X inhibitor treatment (Fig. 4C). These results demonstrate that two distinct, highly abundant USP DUBs exhibit marked substrate specificity in cells at the protein level.

## Defining high-confidence substrates of USP9X by ubiquitinomics and proteomics

Steger et al reported that USP7 inhibition led to increased ubiquitination within 15–30 min, which corresponded to reduced abundance of the ubiquitinated proteins after 6 h of USP7 inhibition (Steger et al, 2021). To identify high-confidence substrates of USP7 and USP9X in DLD-1 cells — defined by increased ubiquitination and decreased protein abundance following DUB inhibitor treatment—proteins were filtered for both the presence of K-GG peptides that were significantly increased at 30 min (log2 >twofold change, adjusted *P* value < 0.05) and the reduction of total protein abundance after 6 h of DUB inhibition (log2 < −0.3, adjusted *P* value < 0.05; Appendix Fig. S5D,E). 26 proteins from DLD-1 cells met the criteria as high-confidence USP9X substrates (Appendix Fig. S5D). With either inhibitor (and as previously observed in Steger et al, 2021), the number of proteins with increased ubiquitination exceeded the number of subsequently degraded proteins by at least ~sixfold. This observation may indicate either non-degradative (e.g. signalling) ubiquitination events, or ubiquitination events that are not yet functional or sufficient to induce proteasomal degradation. However, ubiquitination site detection is also inherently more variable and challenging,

due to low abundance and occupancy, and 'noisier' data based on peptide-level quantification.

We expanded our analysis of USP9X inhibition by profiling the ubiquitin landscape of other cancer cell lines to interrogate the concordance of high-confidence USP9X substrates between distinct cancer cell lines. The breast cancer cell line MDA-MB-231 was treated identically (10 µM WEHI-092, measured at 30 min, 6 h, 24 h). 30 min USP9X inhibition increased the abundance of 1323 K-GG peptides on 851 proteins, while across all time points, a total of 271 proteins were significantly decreased in abundance (Fig. 4D). 69 proteins exhibited increased ubiquitination at 30 min and decreased protein levels (following 30 min, 6 h and/or 24 h of treatment) upon USP9X inhibition, indicating high-confidence USP9X substrates (Fig. 4E). Among these targets are several previously reported USP9X substrates including MIB1 (Izrailit et al, 2017), TTK (Chen et al, 2018), SMAD3 (Dupont et al, 2009), RIT1 (Riley et al, 2024), and CEP55 (Wang et al, 2017; Clancy et al, 2021) that were matched to corresponding biological processes through pathway enrichment analysis (Appendix Fig. S5F,G). Identical analyses performed in SK-MEL-2 and UO-31 cells, which were more sensitive to WEHI-092, exhibited a comparable response (increase in ubiquitination at early time points, and corresponding protein depletion at late time points; Fig. EV1A,B). However, while the number of high-confidence substrates was similar in each cell line individually, the proteins that were regulated differed substantially (Fig. 4F,G). In total, 38 ubiquitinated proteins out of ~600–1000 identified, overlapped across all datasets at 30 min (Fig. 4F), and only eight proteins (ACBD6, CC2D1A, CCDC14, CCDC61, DTX3L, NFX1, PJA2, and RNF135) were classified as overlapping high-confidence substrates across all cell lines tested (Fig. 4G). A number of cell-specific high-confidence substrates were common across two or three cell lines. Surprisingly, in each cell line, the majority of proteins appeared to experience context-specific ubiquitination and degradation by WEHI-092-mediated USP9X inhibition.

## Expansion of proteomic studies to eight cell lines reveals a core set of commonly depleted USP9X targets

To further corroborate these findings, we first validated target lists against published data generated with the USP9X inhibitor FT709. Treatment of HCT116 cells with FT709 significantly reduced CEP55, MKRN2, TTK, CEP131, PCM1 and other proteins (Clancy et al, 2021). We extended these proteomics analyses to MDA-MB-231 cells, which were treated with FT709 or WEHI-092 for 24 h. Notably, treatment with WEHI-092 also led to reduced protein levels of CEP55, MKRN2, CEP131 and PCM1, producing highly correlated datasets overall (Fig. 5A).

We next expanded the global proteomic studies to seven additional cancer cell lines, covering different phenotypes seen in the NCI panel, including cell lines with no growth defect, growth reduction, and UO-31 cells that were highly sensitive to WEHI-092 treatment (Fig. 5B, compare Fig. 3B,C). This set of proteomic experiments were performed on Bruker timsTOF MS instrumentation (see 'Methods'), which achieved proteome coverage from 5833 (UO-31) to 7310 proteins (MDA-MB-231; Fig. 5B). The re-run of MDA-MB-231 cells overlapped in 6828 out of 7310 proteins compared with the first analysis (Fig. 4D, performed on Thermo Fisher Scientific Astral MS instrument), in which 9232 proteins

were observed. All cell lines were treated with 10 µM of WEHI-092 or DMSO (control) for 24 h, differential proteins extracted individually for each cell line, and cross-compared across cell lines.

Filtering for significantly reduced proteins across each cell line indicated that 17 core proteins are decreased in at least four out of eight cell lines tested (Fig. 5C). The top ten regulated proteins by USP9X inhibition include some previously validated targets including CEP55 and MKRN2, and further proteomic hits (EDF1, TCF25) previously identified (Clancy et al, 2021), while TOR4A, PDCL3, TYMS, HECTD1, TAGLN and TGFB1I1 were consistently depleted across cell lines but have not previously been associated with USP9X (Fig. 5D). The reduction in CEP55, MKRN2 and new targets, EDF1, CHMP2B, and DTX3L following USP9X inhibition in multiple cell lines were confirmed by western blotting (Fig. 5E; Appendix Fig. S6A,B). These studies also showed that USP9X levels were not affected by compound treatment, and that depletion of CEP55, MKRN2, CHMP2B and DTX3L could be rescued by proteasome inhibition (Appendix Fig. S6A), consistent with ubiquitin-dependent turnover and direct, rather than indirect (transcriptional/translational) regulation.

siRNA-mediated gene silencing of USP9X in MDA-MB-231 cells corroborated our list of common USP9X-regulated proteins (Appendix Fig. S6C,D). Treatment with WEHI-092 in USP9X knockdown (KD) conditions still regulated a small subset of proteins (TYMS, PDCL3, TOR4A, DTX3L, ACBD6, TGFB1I1); this could be due to remaining USP9X activity, considering a ~90% achieved USP9X depletion (Appendix Fig. S6E). We also detected previously reported USP9X substrates CEP131 and ZNF598 to be depleted during KD in presence of WEHI-092, potentially indicating substrates responding to low USP9X levels. In addition to decreases in protein levels, several of the top ten downregulated proteins across all cell lines had corresponding ubiquitinated peptides in the ubiquitinomics studies, including CEP55, CHMP2B, DTX3L, and EDF1 (Fig. 4E; Appendix Fig. S6F). The identification of 17 proteins that were consistently depleted between cell lines revealed a valuable set of biomarkers for USP9X inhibition.

## Global analysis of USP9Xi-regulated proteins reveals cell-line-specific patterns with functional clusters

Analysis of proteomic datasets for individual cell lines revealed a relatively small number of proteins (between 5 and 30) that changed significantly upon USP9X inhibition at the chosen time point. To identify global patterns of proteome changes by USP9X inhibition using pathway analysis, we combined the results from all cell lines, analysing proteins that were significantly downregulated in at least one of the cell lines. This generated a list of 99 proteins, which were then visualised in context of their cell line origin (Fig. 6A). Strikingly, while there was a set of overlapping proteins (which included proteins discussed above and in Fig. 5), most proteome changes were cell-line-specific (Fig. 6A). The observed depletion upon USP9X inhibition could not be explained by abnormal expression patterns of targets across the tested cell lines (Appendix Fig. S7A). However, the observed clusters of cell-line-specific WEHI-092-regulated proteins, associated with transcriptionally upregulated proteins for individual cell lines (Appendix Fig. S7B).

All this suggests disparate cell-type-specific roles of USP9X which may explain why many of the previously suggested targets of

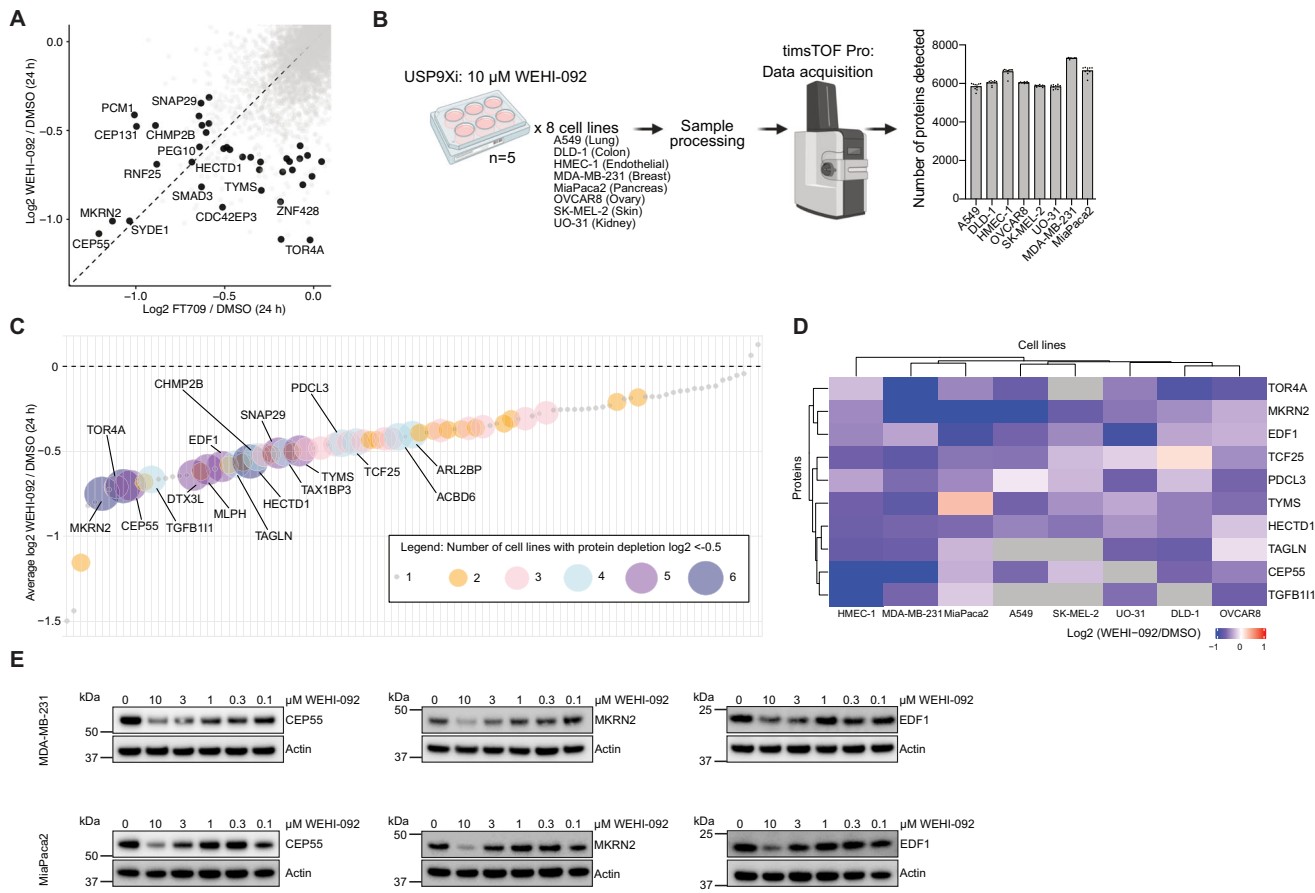

**Figure 5. Global proteomics studies identify commonly regulated proteins across cell lines upon USP9X inhibition.**

(A) Comparison of depleted proteins (log2 <−0.585-fold change over untreated control, adjusted *P* value < 0.05) in MDA-MB-231 cells upon USP9X inhibitor treatment with either FT709 or WEHI-092 (10 μM, 24 h each). Each datapoint represents the mean of four or five biological repeats per condition. The dashed line at *y* = *x* was included for visualisation purposes. (B) Schematic workflow of global proteomics experiments in eight cell lines. The bar graph on the right shows the number of detected proteins per cell line (mean). Each datapoint represents one of five biological repeats per condition (with and without WEHI-092). (C) Identification of commonly regulated proteins upon WEHI-092 treatment (10 μM, 24 h) across eight cell lines. Plotted is the average log2 fold change over untreated control across all eight cell lines, with four or five biological repeats per cell line and condition. The size of each dot indicates in how many cell lines the respective protein was depleted by log2 <−0.5-fold change WEHI-092 treated vs. untreated control treatment. The data was pre-filtered for proteins that are depleted by log2 <−0.585 (adjusted *P* value < 0.05) over the untreated control in at least one out of the eight cell lines. Proteins that are depleted in at least four/eight cell lines are labelled. The *y* axis limit was set to −1.5, and one datapoint outside of this was plotted at *y* = −1.5. (D) Heatmap visualisation of the top ten commonly regulated proteins. Data from (C) was filtered for the top ten proteins with a significant (adjusted *P* value < 0.05) protein level decrease by log2 <−0.585 over the untreated control in at least three out of the eight cell lines with four or five biological repeats per cell line and condition. Hierarchical clustering was performed on proteins (rows) and cell lines (columns) using Euclidean distance as the similarity metric. Heatmap colours indicate fold changes in protein expression, whereby grey indicates that the protein was not detected. (E) Western Blot validation of a subset of commonly regulated USP9X substrates from (C, D). Treatments were performed for 24 h. Representative blots from at least two independent biological repeats are shown with Actin as a sample loading control. Source data are available online for this figure.

USP9X, such as MCL-1 (Schwickart et al, 2010), ERG (Wang et al, 2014), Ets-1 (Potu et al, 2017), YAP1 (Li et al, 2018; Biber et al, 2025), NUAK1 (Fritz et al, 2020), and Itch (Mouchantaf et al, 2006) were not observed as significantly changing proteins in any of the cell lines we tested (see 'Discussion').

We next undertook enrichment analysis of the proteomic changes across different cell lines to understand the pathways and cellular processes affected by USP9X, wherein individual proteins were systematically categorised by gene ontology (GO) terms and subsequently aggregated by biological pathways into clusters (Zhou et al, 2019). Stringent filtering criteria were applied (protein level reduction >50%, adjusted *P* value < 0.05) to identify enriched GO-terms associated with the most significant hits across cell lines. In

this enriched ontology map, nodes representing similar biological processes and pathways were identified and clustered (Fig. 6B).

Analysis of all 99 proteins across cell lines, unveiled distinct biological pathways with which USP9X has previously been associated, including, TGFβ-signalling (Dupont et al, 2009; Jie et al, 2021; Yang et al, 2022), ubiquitin-dependent catabolism (Mouchantaf et al, 2006; Xie et al, 2013), cytoskeleton architecture (Homan et al, 2014) and endosomal trafficking (Savio et al, 2016). The most prominent cluster of regulated proteins upon USP9X inhibition enriched for GO terms associated with mitotic cell cycle regulation and chromosome segregation (Fig. 6B) consistent with the reported role of USP9X in cell cycle regulation and mitosis (Engel et al, 2016; McGarry et al, 2016; Li et al, 2017; Skowyra et al,

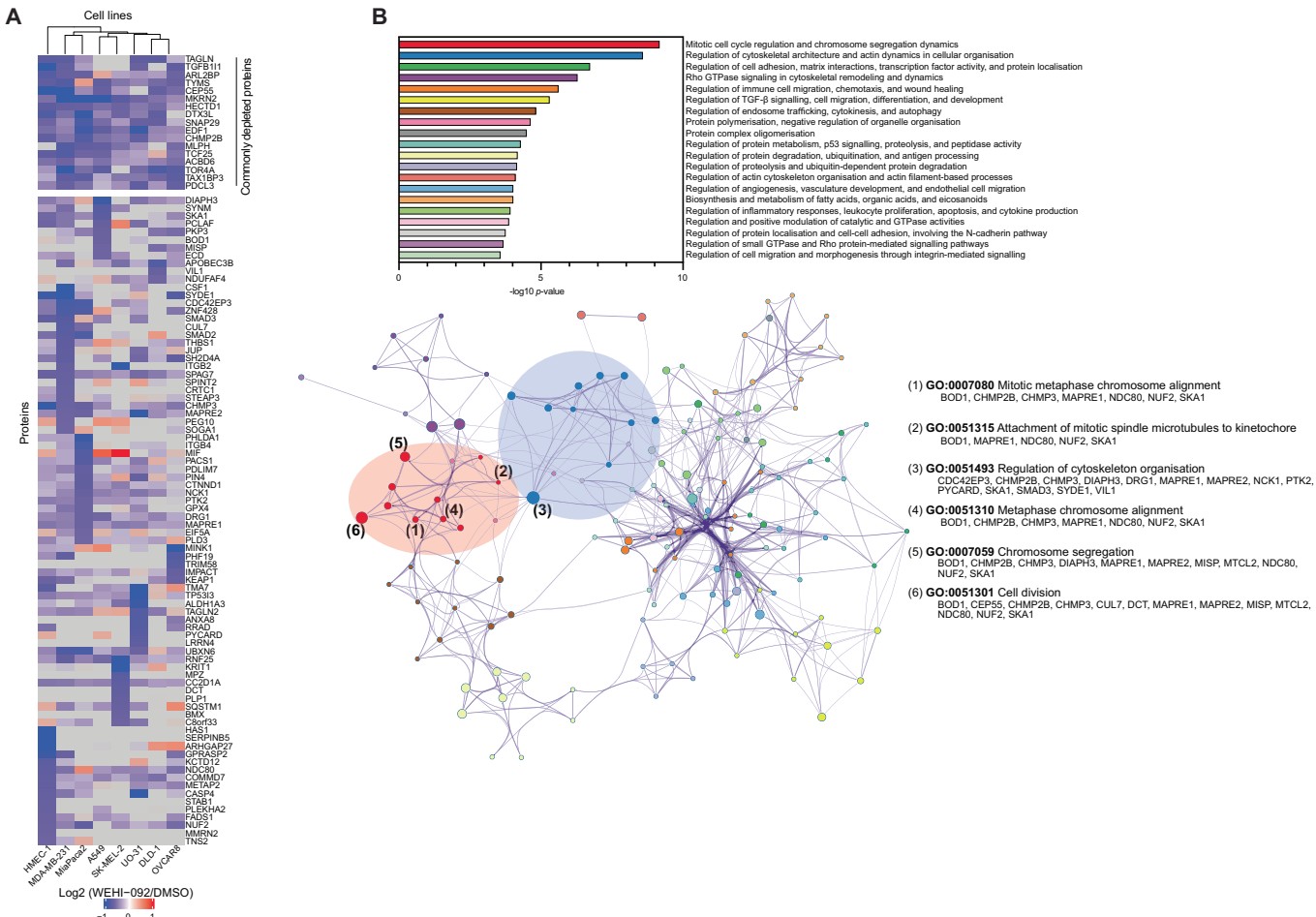

**Figure 6. Global proteomics across eight different cell lines reveals cell line-specific regulation of proteins upon WEHI-092 treatment.**

(A) Heatmap visualisation of proteins with altered abundance upon WEHI-092 treatment in eight cell lines. The data were pre-filtered for proteins that are depleted by log2 <−0.585 (adjusted P value < 0.05) over untreated control in at least one out of the eight cell lines with four or five biological repeats per cell line and condition. Hierarchical clustering was performed on cell lines (columns) using Euclidean distance as the similarity metric. Heatmap colours indicate log2 fold changes in protein expression, whereby grey indicates that the protein was not detected. In total, 17 proteins in the top part are commonly depleted across the cell lines tested (see Fig. 5). (B) Pathway and process enrichment analysis of proteomic dataset from (A) using metascape.org (Zhou et al, 2019). Similar biological processes and pathways were clustered algorithmically, colour-coded for differentiation, and summarised. Clusters were quantitatively ranked (P values determined from hypergeometric testing) in a bar plot (top) and visualised as a network (bottom) where each node corresponds to a specific gene ontology term, with node diameters proportional to the number of contributing proteins. Nodes representing similar biological processes and pathways from the same cluster are shown in the same colour used in the bar plot. Within the network, nodes from within the top two clusters are highlighted through a colour backdrop. The top six most significant individual gene ontology terms are numbered and labelled with their respective gene ontology term and with the protein names from our dataset falling into the term. Source data are available online for this figure.

2018; Kodani et al, 2019; Dietachmayr et al, 2020). Specifically, in addition to the previously reported CEP55, which is depleted across all cell lines, we found multiple centrosomal/kinetochore proteins, including BOD1, SKA1 and kinetochore complex component NDC80, to be USP9X targets, albeit in different cell lines (Fig. 6B).

## USP9X inhibition arrests cells in metaphase

To functionally validate our global proteomic findings, we selected five cell lines from our panels in Fig. 3 to study the effects of WEHI-092 in mitosis using lattice light sheet imaging. Four out of the five selected cell lines (MDA-MB-231, MiaPaca2, SK-MEL-2 and UO-31) showed sensitivity to compound treatment in clonogenic potential assays, whereas HMEC-1 cells were largely unresponsive

to WEHI-092. We analysed cell division by following individual cells over a 24 h window by imaging tubulin and DNA in cells every 10 min and tracked cell divisions with and without WEHI-092 treatment (Appendix Fig. S8A–D). MDA-MB-231, MiaPaca2, and SK-MEL-2 cells showed significant defects in cell division following treatment with WEHI-092 (Fig. 7A,B,D; Appendix Fig. S8E). While the chromosomes condense and the spindle apparatus forms successfully with moderately delayed kinetics, there was no progression beyond metaphase in treated cells (Fig. 7A,B; Movies EV1–4). In contrast, HMEC-1 cells, which were unresponsive to WEHI-092 treatment in growth or survival assays, proceeded through mitosis normally (Fig. 7C,D). Notably, metaphase arrest did not trigger apoptosis in MDA-MB-231 and MiaPaca2 cells, as assessed by Annexin V uptake (Fig. 7E)

suggesting that USP9X inhibition induces a non-lethal mitotic blockade in susceptible cell lines likely explaining effects on clonogenic potential, and growth arrest in distinct cancer cell lines (Fig. 3). Of note, the sensitive cell line UO-31 did not display the arrested mitosis phenotype (Appendix Fig. S8F; Movie EV5). Together, these analyses demonstrate how meta-analysis of proteomic changes induced by a specific DUB inhibitor across various cell lines can provide significant insights into global DUB function.

## Discussion

We here identify a new chemical class of USP9X inhibitors with a distinct piperazine core (Clancy et al, 2021) that is more chemically tractable than the bicyclic core in FT709, with similar in vitro activity, cellular efficacy, and, most importantly, remarkable USP9X specificity in vitro and in cells.

We explain the specificity of WEHI-092 and FT709 through HDX-MS experiments, which reveal a single six-amino acid region, notably located in the Fingers subdomain of USP9X. This six-amino acid motif is unique to USP9X (Ye et al, 2009) among USP family members and differs even in its closest human paralogue, USP24. A USP9X chimera with five USP24-mimicking point mutations was no longer inhibited by WEHI-092 or FT709. Hence, we have pinpointed the binding site for USP9X-specific inhibition, which remarkably is located away from the catalytic centre and represents a previously unrecognised compound binding pocket in the Fingers subdomain. Despite significant effort, we were unable to obtain a co-crystal structure, and it remains to be established how WEHI-092 precisely binds to USP9X. These findings further demonstrate that USP DUBs are remarkable for the many ways that the catalytic domain can accommodate chemically distinct scaffolds and be inhibited (Kazi et al, 2025).

A new USP9X inhibitor scaffold presented an opportunity to clarify the role of USP9X in cancer cell biology. Despite more than 130 papers on this topic, the USP9X field remains conflicted as to whether USP9X is a viable cancer target (Dewson et al, 2023). By using WEHI-092 across a large panel of cancer cell lines combined with performing deep ubiquitinomic and proteomic analyses, we confirm pleiotropic, cell-type-specific effects of USP9X, which may have contributed to an abundance of diverse USP9X targets that have been published but have not been (or could not be) confirmed by others. Our broad study also unveils a multitude of centrosomal substrates, and corresponding regulation of mitosis by USP9X, across multiple cell lines, corroborating previous literature. We further find that USP9X inhibition affects a unique set of proteins as compared to USP7 inhibition, which clearly demonstrates that these two abundant and ubiquitous DUBs have distinct sets of protein substrates. USP9X inhibition increases the abundance of >1000 ubiquitination sites in >500 proteins, 10% of which were also depleted in cells upon inhibitor treatment. Effect size and proportions were comparable to studies on USP7 inhibition (Steger et al, 2021). Whether the remaining 90% of ubiquitination sites are non-degradative events or did not result in protein degradation in the time window studied remains unclear. When we focused on proteome changes after USP9X inhibition, we demonstrated that USP9X regulates distinct sets of proteins in a highly cell-type-specific manner. This observation can at least partially explain the

differences and lack of consistency and/or reproducibility of previously published results with regard to cellular roles of USP9X.

Importantly, we identify a core set of 17 proteins uniformly regulated by USP9X inhibition across highly diverse cell lines. Our data suggests that these proteins represent a consistent set of USP9X targets, and will be useful as proximal biomarkers for future studies. Indeed, proximal biomarker discovery has been a persistent challenge for DUB inhibitor research, and our multi-cell line approach may be more broadly applicable for other DUBs for which specific inhibitors are available.

The 17 commonly regulated proteins, in the context of all proteins changed across all cell lines, allowed GO-term meta-analysis (Zhou et al, 2019) to unveil critical cellular processes regulated by USP9X inhibition, with the most prominent being the regulation of mitotic processes in metaphase. Other pathways included TGFβ signalling, translation, and regulation of cytoskeletal architecture. Here, our data were consistent with a subset of previous studies demonstrating regulation of cell cycle progression (Vong et al, 2005; Engel et al, 2016; McGarry et al, 2016; Skowyra et al, 2018), SMAD signalling (Dupont et al, 2009; Stegeman et al, 2013), and ribosomal quality control (Clancy et al, 2021) following genetic deletion of USP9X. It remains unclear how USP9X identifies its substrates and regulates these pathways. Previous data have shown that USP9X appears to co-locate with centrosomal proteins, and has been localised to specific sites in cells (Murray et al, 2004; Reijnders et al, 2016; Li et al, 2017; Wang et al, 2017), which may explain how USP9X appears to regulate multiple proteins in centrosomal complexes. However, it is also possible that specific loss of a critical protein leads to collateral destabilisation of larger complexes and indirect protein depletion. A further insight from our proteomic analysis is the identification of numerous E3 ligases, including DTX3L, HECTD1, MKRN2, PJA2 and RNF135 within the top regulated proteins across cell lines, and numerous additional E3 ligases in individual cell lines, suggesting that USP9X may sit atop a hierarchy of UPS processes, with USP9X inhibition unleashing a cascade of secondary events that may induce further proteome changes. This observation may also account for the unique proteomic changes observed in individual cell lines upon USP9X inhibition that may be dependent on secondary processes.

Importantly, however, we confirm the role of USP9X in cell cycle regulation through our proteomic analysis and cell imaging studies, which show that WEHI-092 leads to mitotic arrest in cancer cells, which intriguingly was not associated with cell death. This explains observed phenotypes in cancer cell line panels where WEHI-092 leads to mild growth arrest but not death, with few exceptions, that are exaggerated in longer-term studies assessing clonogenic potential.

It was striking to note that UO-31 cells, a papillary renal cell carcinoma cell line, was highly sensitive to USP9X inhibition. In this cell line, USP9X appears to be synthetically lethal with (an) other protein(s) or pathway(s). Despite significant efforts to understand differences at the transcriptional, proteomic, and ubiquitinomic levels, the specific co-dependencies in UO-31 cells remain opaque and will require further study.

Establishing WEHI-092 as an agent that arrests mitosis could be beneficial in exploiting cancer vulnerabilities. Mitotic spindle agents such as Taxol are often front-line chemotherapy treatments and can be boosted by co-treatment with secondary mitosis inhibitors or DNA-damaging agents. It would be interesting to test

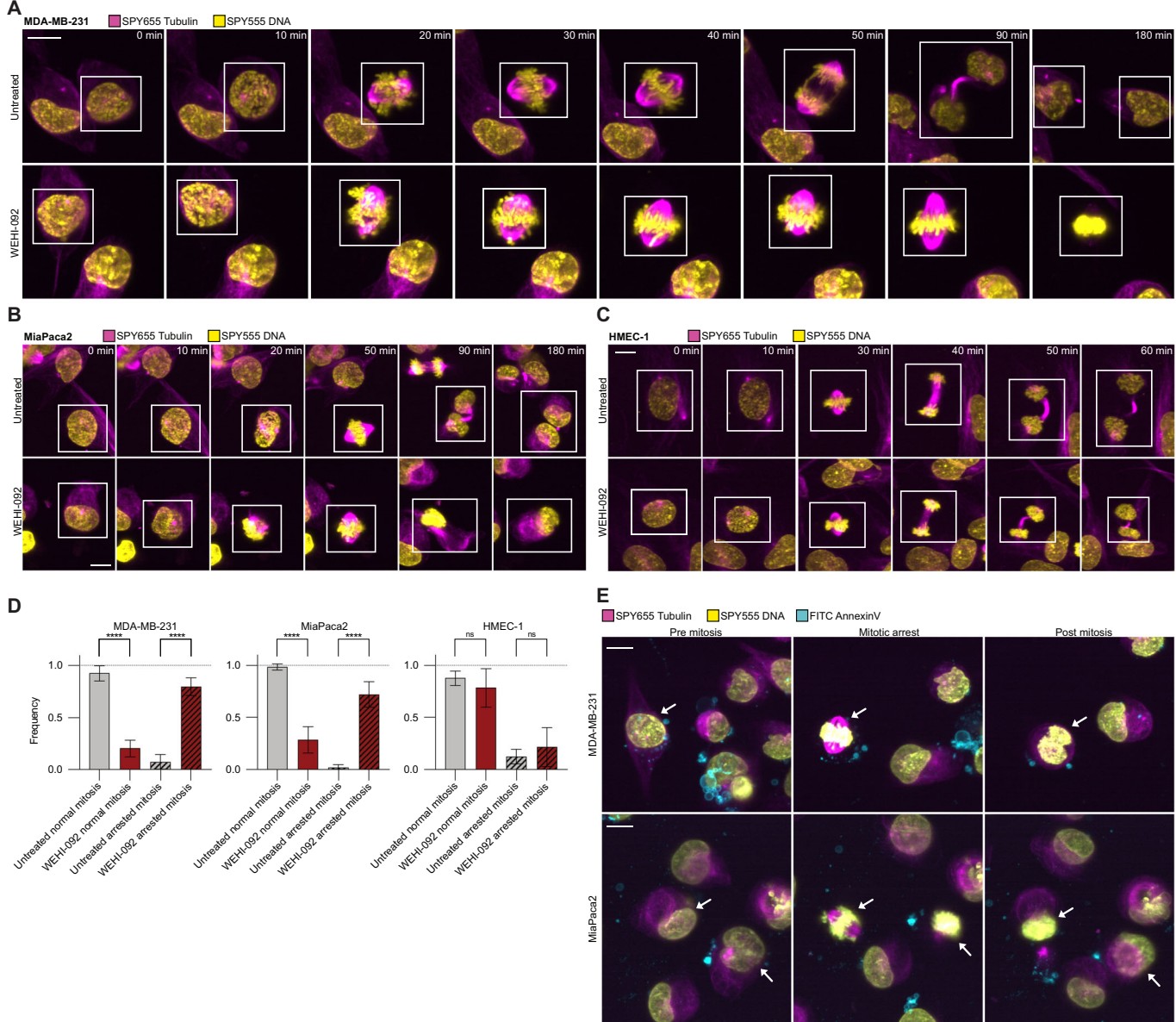

**Figure 7. Inhibition of USP9X causes arrested mitosis in cancer cells.**

(A) Lattice light sheet imaging of MDA-MB-231 cells upon WEHI-092 treatment. Cells were treated with 15 μM WEHI-092 for 24 h before imaging commenced. Times indicated refer to the initiation of mitosis for the cells highlighted in the white boxes. DNA is shown in yellow, and tubulin is shown in magenta. Images were taken every 10 min and are a representative of three independent biological repeats with a total of 29 divisions (untreated) and 25 divisions (WEHI-092), respectively. Scale bar: 10 μm. (B) Lattice light sheet imaging of MiaPaca2 cells upon WEHI-092 treatment. Conditions and colour channels are similar to (A). Images were taken every 10 min and are a representative of three independent biological repeats with a total of 92 divisions (untreated) and 39 divisions (WEHI-092). Scale bar: 10 μm. (C) Lattice light sheet imaging of HMEC-1 cells upon WEHI-092 treatment, also as in (A). Images were taken every 10 min and are a representative of three independent biological repeats with a total of 30 divisions (untreated) and 34 divisions (WEHI-092). Scale bar: 10 μm. (D) Quantification of mitosis phenotypes in MDA-MB-231, MiaPaca2 and HMEC-1 cells. Cell divisions were categorised into normal and arrested mitosis based on the phenotypes observed in (A–C). In the bar graphs, the mean across three independent biological repeats is shown with a total of 29 divisions (untreated) and 25 divisions (WEHI-092) for MDA-MB-231 cells; 92 divisions (untreated) and 39 divisions (WEHI-092) for MiaPaca2 cells; 30 divisions (untreated) and 34 divisions (WEHI-092) for HMEC-1 cells. Error bars, s.d. One-way ANOVA with post-hoc multiple comparison Tukey test was performed for significance testing. Significance level ****$P < 0.0001$, ns not significant. (E) Cells with arrested mitosis phenotype are not stained positive for Annexin V. MiaPaca2 and MDA-MB-231 cells were treated as described in (A, B). DNA is shown in yellow, tubulin is shown in magenta, and Annexin V is shown in cyan. Scale bar: 10 μm. Representative images of three independent experiments. Arrows indicate the same cell before, during and post mitosis. Source data are available online for this figure.

USP9X inhibitors, which had little or no toxicity on primary cells, in combination therapy, which may sensitise cancer cells to chemotherapy.

Overall, our work exemplifies how specific DUB inhibitors can help unravel DUB biology and increase understanding of DUB function in diseases such as cancer. Broad cell line screening with high-quality compounds is becoming feasible, can reveal biomarkers, and can also overcome limitations arising from genetic manipulation. Our work, together with previous complementary studies (Pinto-Fernández et al, 2019; Steger et al, 2021; Varca et al, 2021; Doherty et al, 2022; Chan et al, 2023), paves the way to streamline DUB drug discovery and refine target rationales for DUB inhibitors for human disease.

## Methods

### Reagents and tools table

| Reagent/resource | Reference or source | Identifier or catalogue number |
| --- | --- | --- |
| **Cell lines (human)** | | |
| A375 | In-house | CVCL_0132 |
| A498 | In-house | CVCL_1056 |
| A549 | In-house | CVCL_0023 |
| ACHN | In-house | CVCL_1067 |
| BxPC3 | In-house | CVCL_0186 |
| CAKI-1 | In-house | CVCL_0234 |
| DLD-1 | In-house | CVCL_0248 |
| HEK293T | In-house | CVCL_0063 |
| HepG2 | In-house | CVCL_0027 |
| HMEC-1 | In-house | CVCL_0307 |
| Huh7 | In-house | CVCL_0336 |
| Human dermal fibroblasts (HDF) | Lonza | 19TL178960 |
| HUVEC | StemCell Technologies | 200-0630 |
| MCF-7 | In-house | CVCL_0031 |
| MDA-MB-231 | In-house | CVCL_0062 |
| MDA-MB-468 | In-house | CVCL_0419 |
| MiaPaca2 | In-house | CVCL_0428 |
| OVCAR8 | In-house | CVCL_1629 |
| PANC-1 | In-house | CVCL_0480 |
| SK-MEL-2 | In-house | CVCL_0069 |
| SK-MEL-28 | In-house | CVCL_0526 |
| THP-1 | In-house | CVCL_0006 |
| UO-31 | In-house | CVCL_1911 |
| **Recombinant DNA** | | |
| pOPIN-S_USP9X_CD | This study | n/a |
| pOPINS_USP9X_CD_avi | This study | n/a |
| pOPIN-S_USP9X_CD_chimUSP24 | This study | n/a |
| pOPIN-S_USP9X_CD_chimUSP24_dLoop | This study | n/a |

| Reagent/resource | Reference or source | Identifier or catalogue number |
| --- | --- | --- |
| pOPIN-S_USP9X_CD_dLoop | This study | n/a |
| **Antibodies** | | |
| Mouse anti-Actin HRP | Santa Cruz Biotechnology | sc-47778 |
| Rabbit anti-CEP55 | Cell Signaling Technology | 81693 |
| Rabbit anti-CHMP2B | Cell Signaling Technology | 76173 |
| Rabbit anti-DTX3L | Cell Signaling Technology | 14795 |
| Rabbit anti-EDF1 | Abcam | ab174651 |
| Rabbit anti-MKRN2 | Abcam | ab72055 |
| Rabbit anti-USP9X | Bethyl Laboratories | A301-350A |
| **Oligonucleotides and other sequence-based reagents** | | |
| siRNA negative control | Qiagen | 1027280 |
| siRNA USP9X | Qiagen | SI03101973 |
| Full_USP9X_pOPINS_fwd | This study | 5'–3': CAGATCGGTGGTCGTCCG |
| Full_USP9X_pOPINS_rev | This study | 5'–3': GGTCTAGAAAGCTTTAGCTAATG |
| pOPINS_prom | This study | 5'–3': GCGAACAGATCGGTGGT |
| T7_prom | This study | 5'–3': TAATACGACTCACTATA |
| T7_term | This study | 5'–3': GCTAGTTATTGCTCAGCG |
| USP9X_seqPrim_fwd | This study | 5'–3': CGACGAAAAACAAGATAACGAG |
| USP9X_seqPrim_rev | This study | 5'–3': CCCCTTCCAACTTTGCTAC |
| **Chemicals, enzymes and other reagents** | | |
| 5X In-Fusion® HE enzyme premix | Takara Bio | ST0345 |
| Acetonitrile (ACN) | Thermo Fisher Scientific | A955-4 |
| Adenosine 5'-triphosphate (ATP) disodium salt hydrate | Sigma-Aldrich | A2383 |
| Deuterium chloride | Cambridge Isotope Laboratories, Inc. | DLM-2-50 |
| Deuterium oxide | Cambridge Isotope Laboratories, Inc. | DLM-4-100 |
| Dimethyl pimelimidate dihydrochloride | Merck | D8388-250MG |
| Dimethyl sulfoxide (DMSO) | Sigma-Aldrich | 472301 |
| Dithiothreitol (DTT) | Fluorochem | M02712 |
| DMEM/F12 medium | Thermo Fisher Scientific | 10565-018 |
| DMEM medium | Thermo Fisher Scientific | 11885-084 |
| DNase I | Roche | 11284932001 |
| EOAI3402143 | MedChemExpress | HY-111408 |
| Foetal bovine serum (FBS) | Bovogen | SFBS-F S00JF |
| Formic acid (FA) | Thermo Fisher Scientific | T85178-AD |

| Reagent/resource | Reference or source | Identifier or catalogue number |
|---|---|---|
| FT671 | Focus Bioscience | HY-107985-5MG |
| FT709 | MedChemExpress | HY-145967 |
| GlutaMAX™ | Thermo Fisher Scientific | 35050061 |
| Human epidermal growth factor (EGF) | Sigma-Aldrich | E9644 |
| Human recombinant basic fibroblast growth factor (bFGF) | StemCell Technologies | 78003 |
| Isopropyl-b-D-thiogalactoside (IPTG) | GoldBio | I2481C100 |
| Lipofectamine™ RNAiMAX | Thermo Fisher Scientific | 13778150 |
| Lysozyme | Glentham Life Science | GE8228 |
| Propidium Iodide (PI) | Sigma-Aldrich | P4170 |
| Q5® Hot Start High-Fidelity 2X master mix | New England Biolabs | M0494S |
| Quinoline-Val-Asp-Difluorophenoxy-methylketone (QVD) | MedChemExpress | HY-12305 |
| Sodium deoxycholate (SDC) | Sigma-Aldrich | D5670 |
| Sodium dodecyl sulfate (SDS) | Sigma-Aldrich | 436143-100 G |
| SOLu-Trypsin Protease | Sigma-Aldrich | EMS0004 |
| SPY555-DNA | Spirochrome, Inc. | SC201 |
| SPY650-Tubulin | Spirochrome, Inc. | SC503 |
| T4 DNA ligase | Promega | M1808 |
| Trypsin, treated with L-1-tosylamido-2-phenylethyl chloromethyl ketone (TPCK) | Thermo Fisher Scientific | 20233 |
| Ubiquitin Rhodamine110 | UbiQ Bio | UbiQ-126 |
| WP1130 | Merck | US1681685-10MG |
| **Software** | | |
| Biacore Insight Control Software | Cytiva | v5.0.18.22405 |
| ChronosHDX software | Waters | n/a |
| Clariostar software | BMG Labtech | v5.70 R3 |
| IncuCyte software | Intellicyt | v2021A_20211.2.0.0 |
| SnapGene software | n/a | v7.0.1 |
| Zen Blue | Zeiss | v3.6 |
| RStudio | n/a | v2024.12.0 + 467 |
| GraphPad Prism | n/a | v10.3 |
| Deuteros | n/a | v2.0 |
| Biacore Insight Evaluation Software | Cytiva | v5.0.18.22102 |
| ImageJ2/Fiji | ImageJ | v2.14.0/1.54 f |
| Image Lab | Bio-Rad | v6.1 |

| Reagent/resource | Reference or source | Identifier or catalogue number |
|---|---|---|
| Microsoft Excel | Microsoft | v16.58 |
| **Other** | | |
| ÄKTA (with fraction collector) | Cytiva | n/a |
| Biacore 8 K+ | Cytiva | n/a |
| IncuCyte S3 (live-cell imaging) | Sartorius | n/a |
| Lattice Lightsheet 7 | Zeiss | n/a |
| Mass spectrometer: | | n/a |
| Astral | Thermo Fisher Scientific | |
| Eclipse | Thermo Fisher Scientific | |
| SYNAPT G2-Si | Waters | |
| timsTOF Pro | Bruker | |

## Medicinal chemistry

Details on medicinal chemistry, including the synthesis of compounds used in this study, can be found in the Appendix Material and Methods.

## Protein biochemistry

### Molecular biology

An expression construct for the USP9X catalytic domain (aa 1551–1970, NCBI Reference Sequence: Q93008-1), was codon optimised for bacterial expression, synthesised in gBlock format (Integrated DNA Technologies) and cloned into the pOPIN-S vector (Berrow et al, 2007), which was digested with KpnI and HindIII using In-Fusion™ HD cloning (Takara Clontech). For SPR, constructs were ordered with a GGS linker followed by a C-terminal AviTag™ (aa sequence: GLNDIFEAQKIEWHE) and were cloned as above. For the USP9X Δβ catalytic domain a deletion construct ΔE1924-K1943 and for USP9X/USP24 chimera, a construct with USP9X Y1757-A1766 replaced with the USP24 (NCBI Reference Sequence: NP_056121) aa motif FVRGEVLEGS was used. All constructs were confirmed by Sanger sequencing (AGRF).

### Protein expression and purification

Protein expression vectors were transformed into *E. coli* BL21 cells. Cells were grown in 2× YT medium at 37 °C (200 rpm) until $OD_{600} = 0.6$ was reached. Protein expression was induced by adding IPTG to a final concentration of 0.4 mM. Cultures were grown overnight (o/n) at 16 °C, 200 rpm. Next, cultures were harvested by centrifugation at 5000×*g* for 10 min at 4 °C. All buffers used for protein purification were used at pH 8.0. Cells were lysed by sonication in lysis buffer (25 mM Tris [pH 8.0], 500 mM NaCl, 5% [v/v] glycerol) supplemented with 1 mM TCEP, EDTA-free protease inhibitor cocktail tablets (Roche), lysozyme (2 mg/mL) and DNase I (100 μg/mL). Lysates were cleared by centrifugation at 20,000 rpm for 30 min at 4 °C. The cleared supernatant was filtered (0.45 μm) and then incubated with Ni-NTA His resin (EMD Millipore). The resin was eluted with elution buffer (25 mM Tris [pH 8.0], 250 mM NaCl, 250 mM imidazole, 1 mM TCEP). The eluate was incubated with 0.1 mg/mL SENP1 protease (purified as

described previously in Pruneda et al, 2016) to cleave the His/SUMO tag in dialysis buffer (25 mM Tris [pH 8.0], 200 mM NaCl, 5 mM 2-mercaptoethanol) o/n at 4 °C. The following day, the concentrated and tag-cleaved protein was incubated with Ni-NTA His resin and the eluate was collected for purification by ion-exchange chromatography using a Resource Q 6 mL column (Cytiva) with buffer A (25 mM Tris [pH 8.0], 50 mM NaCl, 5 mM 2-mercaptoethanol) and buffer B (25 mM Tris [pH 8.0], 1 M NaCl, 5 mM 2-mercaptoethanol). Eluates were pooled and concentrated using a 10 kDa membrane filter (Merck). Protein was further concentrated and purified by size-exclusion chromatography (SEC) using a HiLoad 16/600 SuperDex 75 pg column (Cytiva) in SEC buffer (25 mM Tris [pH 8.0], 150 mM NaCl, 1 mM TCEP). The pure protein fractions were pooled, concentrated (12 mg/mL) and flash-frozen in liquid nitrogen. Protein was stored at −80 °C. Ubiquitin suicide probes were prepared as described previously (Ekkebus et al, 2013; Gersch et al, 2017).

## Ub-Rho cleavage

USP9X catalytic domain (final concentration 7.8 nM) was pre-incubated with USP9X inhibitors in a 12-point, 1 in 3 titration for 30 min at ambient temperature and subsequently mixed with 100 nM Ub-Rh110MP substrate in assay buffer (20 mM Tris [pH 8.0], 0.01% [v/v] Triton X 100, 5 mM DTT, 0.03% [w/v] BSA) at pH 8.0 for 35 min at ambient temperature. The reaction was stopped through the addition of 10 mM citric acid. The assay was performed in 384-well format with a total reaction volume of 16 μL. Data was collected using the ClarioSTAR plate reader (BMG Labtech) at an emission of 535 nm. Blank corrected values were normalised to DMSO control (100% activity remaining) and were used for $IC_{50}$ calculation using GraphPad Prism v10.3 using the inhibitor vs. response analysis (four parameter fit) model. $IC_{50}$ values presented in our study are the mean of three independent biological repeats unless indicated otherwise.

## SPR

To generate a biotinylated protein for SPR studies, a reaction of 150 μM D-Biotin (Sigma-Aldrich), 1.25 μM BirA, 2 mM ATP (Sigma-Aldrich) and 5 mM $MgCl_2$ (Sigma-Aldrich) was mixed with purified protein at a 1 in 4 ratio. The reaction was incubated for 2 h at 4 °C, and successful biotinylation of the USP9X catalytic domain was confirmed by a size shift of ~244 Da detected by MS. Next, the sample was desalted using PD midiTrap™ G-25 (GE Healthcare) columns following the manufacturer's instructions and was further purified by SEC, concentrated, and stored at -80 °C. Enzyme activity was tested in a ubiquitin-probe assay. SPR experiments were performed on a Biacore 8 K+ instrument (Cytiva). USP9X catalytic domain with a C-terminal AviTag™ was immobilised in 10 mM HEPES, 150 mM NaCl at pH 7.4 on a Series S Sensor Chip SA (Cytiva) by coupling. Compounds were diluted from 10 mM stocks in DMSO into PBS-P+ running buffer (20 mM phosphate buffer with sodium phosphate dibasic 39% [v/v] and sodium phosphate monobasic 61% [v/v], 150 mM NaCl, 0.05% [v/v] Tween-20 and 1 mM TCEP at pH 6.2). Running buffer was supplemented with 2% (v/v) DMSO. Multi-cycle kinetics were performed with 120 s associations and 600 s dissociations at 30 μL/min with no further regeneration. Binding constants were determined in Biacore Insight evaluation software (v5.0.18) at steady state, taking the median response over a 5-s window

beginning at 10 s before the end of the association phase. Final $K_D$ values are the mean of three independent experiments.

## Ubiquitin cleavage and ubiquitin-probe assays

For inhibitor activity assays, USP9X catalytic domain (di-ubiquitin cleavage assay: 750 nM, ubiquitin-probe assays: 2.5 μM) was pre-incubated with USP9X inhibitors for 30 min at ambient temperature and subsequently mixed with di-ubiquitin (final concentration 2 μM) in assay buffer (25 mM Tris [pH 8.0], 150 mM NaCl, 10 mM DTT) at pH 8.0 or was mixed with ubiquitin-propargylamide (Ub-PA)/ubiquitin-vinyl sulfonate (Ub-VS) probes at a final concentration of 10 μM in assay buffer. For both assays, the mixture was incubated over a time course, and the reaction was stopped by mixing samples with 2× LDS sample buffer (Invitrogen) supplemented with 3% (v/v) 2-mercaptoethanol. All protein samples were run on a 4–12% Bis-Tris SDS-PAGE gradient gel (Invitrogen) for 60 min at 140 V. The gel was stained with InstantBlue® (Abcam) and imaged.

## WEHI-092 specificity DUB panel

WEHI-092 specificity was assessed using the commercial DUBprofiler™ (DUBprofiler™-FLEX SPT, v8) platform. WEHI-092 powder was supplied, and testing was performed at Ubiquigent™ Limited (Dundee, United Kingdom) at 50 μM WEHI-092 in two technical replicates. 49 human DUBs were incubated with WEHI-092 for 15 min, Ub-Rho substrate was added, and % activity remaining following WEHI-092 exposure was calculated based on fluorescence relative to DMSO control. UAF1 was added to the USP1, USP12 and USP46 reactions (with WDR20 additionally added to the USP12 and USP46 reactions), respectively. Ubiquitin was added to the USP5 reaction at $K_D$. Proteasome-VS was added to the USP14 reaction at $K_D$. OTUD5 p177S was used. The reaction for AMSH-LP was supplemented with Zinc.

## Mass spectrometry-based 3X-FLAG-Ub-VS DUB competition assay with cell extracts (DUB IP-MS)

MCF-7 cells have been profiled for active DUBs previously (Pinto-Fernández et al, 2019). Lysates were generated as described by others (Turnbull et al, 2017) using 30 s freeze-thaw cycles (thrice) in lysis buffer (50 mM Tris [pH 7.4], 5 mM $MgCl_2 \times 6 H_2O$, 0.5 mM EDTA [pH 8.0], 250 mM sucrose and 1 mM DTT) at pH 7.5. For experiments with crude cell extracts, 50 μg MCF-7 lysate was incubated with 50 μM WEHI-092 for 60 min at ambient temperature. A pre-treatment of cell lysates with 0.5 μg Ub-VS was included for a baseline DUB response to the Ub-VS probe. 0.1 μg 3X-FLAG-Ub-VS probe was added to samples and incubated for 5 min at 37 °C. Incubation with the probe was optimised to minimise replacement of non-covalent inhibitor WEHI-092 by the covalent probe. 3X-FLAG-Ub-VS labelled DUBs were captured by incubation with anti-FLAG M2 affinity gel resin (Sigma-Aldrich, A2220) for 3.5 h at 4 °C on a rotator. Samples were washed thrice with lysis buffer and eluted from the resin with lysis buffer supplemented with 1% (w/v) SDS following a 10 min incubation. Two consecutive elutions were pooled, and samples were stored at −80 °C until processing.

## HDX-MS

Deuterium labelling of USP9X catalytic domain was performed as described previously (Asadollahi et al, 2023) at 20 °C for periods of

0, 6, 60, 6000 s using a PAL Dual Head HDX Automation manager (Trajan/LEAP) controlled by the ChronosHDX software. In total, 3 μL of USP9X catalytic domain (~25 μM) was transferred to 55 μL of non-deuterated (50 mM potassium phosphate buffer at pH 7.4 in $H_2O$) or deuterated (50 mM potassium phosphate buffer pD 7 in $D_2O$) buffer and incubated for the respective times with 80 μM WEHI-092 or 30 μM FT709. Quenching was performed by adding 50 μL of the protein mix to 50 μL of quench buffer (50 mM potassium phosphate buffer [pH 2.3], 4 M guanidine hydrochloride and 0.1% [v/v] n-dodecylphosphocholine) at 1 °C. For online pepsin digestion, 80 μL of the quenched sample was passed over an immobilised pepsin column (2.1 × 30 mm Enzymate BEH, Waters) equilibrated in 0.1% (v/v) formic acid (FA) in $H_2O$ at 100 μL/min. To further reduce peptide carryover, n-octyl-β-d-glucopyranoside at 1% (w/w) was added to the pepsin column wash solution (1.5 M guanidine hydrochloride, 4% [v/v] acetonitrile [ACN], 0.8% [v/v] FA). Proteolysed peptides were captured and desalted by a C18 trap column (VanGuard BEH; 1.7 μm; 2.1 × 5 mm [Waters]) and eluted with ACN and 0.1% (v/v) FA gradient (5% to 40% in 8 min, 40% to 95% in 0.5 min, 95% 1.5 min) at a flow rate of 80 μL/min and separated on an ACQUITY UPLC BEH C18 analytical column (1.7 μm, 1 × 50 mm, [Waters] delivered by ACQUITY UPLC I-Class Binary Solvent Manager [Waters]). For MS, an ion mobility equipped SYNAPT G2-Si mass spectrometer (Waters) was used. Instrument settings were: 3.0 KV capillary and 40 V sampling cone with source and desolvation temperatures of 100 °C and 40 °C, respectively. The desolvation and cone gas flow were at 80 L/h and 100 L/h, respectively. High-energy ramp trap collision energy was from 20 to 40 V. All mass spectra were acquired using a 0.4 s scan time with continuous lock mass (Leu-Enk, 556.2771 $m/z$) for mass accuracy correction. Data were acquired in HDMS$^E$ (ion mobility) mode, and peptides from non-deuterated samples were identified using Protein Lynx Global Server (PLGS, v3.0, Waters). To ensure high peptide selection stringency, we applied additional filter constraints of 0.3 fragments per residue, minimum consecutive product of 1, minimum score of 6, minimum intensity of 2500, maximum MH+ error of 5 ppm, retention time RSD of 10% and file threshold of 3 out of 6 HDMS$^E$ files. The deuterium uptake values were calculated for each peptide using DynamX 3.0 (Waters). Deuterium exchange experiments were performed in technical triplicate for each of the timepoints. Peptides with a statistically significant difference in HDX were determined using the Deuteros, v2.0 (Lau et al, 2021) software with a hybrid significance test (Welch's $t$ test) with a 99% confidence interval.

### Multiple sequence alignment of USP catalytic domains
Annotated catalytic domain sequences (USP9X UniProt ID Q93008-1, aa 1557–1956, USP24 UniProt ID Q9UPU5, aa 1689–2042, USP34 UniProt ID Q70CQ2, aa 1894–2239, USP28 UniProt ID Q96RU2, aa 162–650, USP25 UniProt ID Q9UHP3, aa 169–657) were extracted and a multiple sequence alignment (MSA) was performed using Clustal Omega (EMBL-EBI; Madeira et al, 2024).

### AlphaFold3
For the AlphaFold3 (Abramson et al, 2024) models of USP9X and USP24 the following sequences were used: Ubiquitin (UniProt ID P0CG48, aa 1–76), USP9X catalytic domain with ubiquitin (UniProt ID Q93008-1, aa 1549–1970), USP24 catalytic domain

with ubiquitin (UniProt ID Q9UPU5, aa 1688–2042), USP24 full-length (UniProt ID Q9UPU5, aa 1–2620). Source code was downloaded and run on internal servers.

## Studies in cells

### Cell lines
MDA-MB-231, MiaPaca2, UO-31, HMEC-1, SK-MEL-2, OVCAR8, DLD-1 and A549 cells were validated at CellBank Australia. Primary cells HUVEC (STEMCELL Technologies) and HDF (Lonza) were sourced from commercial providers. All other cell lines were sourced in-house. Cell lines were screened monthly for *Mycoplasma* contamination using the MycoAlert® kit (Lonza LT07-318) as per the manufacturer's instructions. All cells used were *Mycoplasma* free. Details on culturing media for cell lines used in this study are listed in Appendix Table S1. All cell lines were cultured at 37 °C, 5% $CO_2$ in a standard tissue culture incubator. HDFs were cultured at 37 °C, 10% $CO_2$.

### Colony formation assays
Cells were seeded at low density (200–4000 cells per well) after initial optimisation in a six-well tissue culture-treated plate (Falcon). Twenty-four hours post seeding, media was replaced, and the inhibitor was added. Cells were left to grow into colonies between 7 and 12 days, depending on the cell line. For analysis, the media was removed, and the cells were fixed with 10% (w/v) formalin for 10 min. Colonies were stained with crystal violet (Sigma-Aldrich, 0.5% [w/v] in 20% [v/v] MeOH) for 10 min and washed thrice with distilled $H_2O$. Plates were imaged using the ChemiDoc imaging system (Bio-Rad) and analysed using the ImageJ ColonyArea plugin as described previously (Guzmán et al, 2014).

### Live-cell imaging assays
Cells were seeded at varying densities (1000–8000 cells per well) after initial optimisation in a 96-well tissue culture-treated plate (Thermo Fisher Scientific). Cells were incubated o/n, and the next day, inhibitors and controls diluted in cell media were added to the cells to a final volume of 200 μL per well. Cell death was measured using PI (Sigma-Aldrich). Cells were imaged with four images per well every 2 h for up to 96 h using the IncuCyte live-cell analysis system (S3, Sartorius). Cell proliferation data were analysed using Microsoft Excel by normalising confluence to control conditions. For cell killing readouts, the number of PI-positive cells was normalised to the control conditions for each timepoint.

### NCI cancer cell line panel
NCI-60 cancer cell panel (National Cancer Institute, USA) was used to assess cell line sensitivity to WEHI-092 and WEHI-680 treatment. WEHI-092 and WEHI-680 were supplied, and experiments were performed at the NCI developmental program (DTP). Inhibitors were tested at 10 μM. Data shown is the mean of two technical replicates and is shown relative to the no-drug control, and relative to the time zero number of cells.

### Cell permeability assessment
The apparent permeability coefficient was assessed using Caco-2 cell monolayers as described previously (Katneni et al, 2018). In

brief, donor solutions were prepared by spiking a compound stock solution into transport buffer (Hank's balanced salt solution with 20 mM HEPES, Thermo Fisher Scientific) at 10 µM (WEHI-092, FT709). Compound flux was assessed over a period of up to 120 min, with samples taken from the acceptor chamber at multiple time points. Samples from the donor chamber were taken at the beginning and at the end of the experiment. The mass balance at the end of the permeability experiment was calculated according to the following equation, where compound mass was calculated as the product of molar concentration and volume of donor or acceptor solution:

$$\text{Mass balance}(\%) = \frac{\text{Mass}_{\text{final donor}} + \text{Mass}_{\text{final acceptor}}}{\text{Mass}_{\text{initial donor}}} \times 100$$

The apparent permeability coefficient ($P_{\text{app}}$) was determined based on the rate of compound appearance in acceptor buffer at steady state using the following equation:

$$P_{\text{app}}(\text{cm/s}) = \frac{dQ}{dt} \times \frac{1}{c_{\text{initial donor}} \times A}$$

$\frac{dQ}{dt}$: apparent steady-state transport rate (µmol/s)

A: surface area of Caco-2 monolayer (0.3 cm$^2$ in the test system used)

$c_{\text{initial donor}}$: concentration in the donor at the start of experiment (µmol/cm$^3$).

### siRNA-mediated USP9X silencing experiments

For KD experiments, siRNA targeting USP9X (Qiagen, #SI03101973, target sequence: GACGATGTATTCTCAATCGTA) was used as recommended by the manufacturer, and a negative control siRNA (Qiagen, #1027280) was included as a control. In brief, MDA-MB-231 cells were plated in six-well tissue culture-treated plates (Falcon) in absence of antibiotics in the culture media, left to adhere o/n and transfected at approximately 70% confluence using a 1:1 mixture of a) 150 µL Opti-MEM™ (Thermo Fisher Scientific) and 9 µL Lipofectamine™ RNAiMAX (Thermo Fisher Scientific) with b) 150 µL Opti-MEM™ and 30 pmol siRNA per well. The mixture was diluted 1 in 8 in cell culture media, and cells were grown for 96 h. KD was confirmed via western blotting.

### Immunoblotting

Cells were plated in 24-well tissue culture-treated plates (Falcon), and one day post plating media was replaced, and cells were treated with WEHI-092. The next day, cells were lysed in DISC lysis buffer (20 mM TRIS/HCl [pH 7.5], 150 mM NaCl, 2 mM EDTA [pH 8.0], 1% [v/v] Triton X 100, 10% [v/v] glycerol) supplemented with 2% (w/v) SDS and PhosSTOP tablets (Roche). To shred DNA, DISC lysates were run through polypropylene columns (Pierce). Proteins were separated by SDS-PAGE on a 4–12% Bis-Tris gradient gel (Invitrogen) for 10 min at 70 V followed by 90 min at 120 V. Samples were transferred to Immobilon-E Transfer PVDF membranes (Merck) by wet transfer at 4 °C for 60 min at 100 V. Membranes were blocked in 5% (w/v) skim milk (Devondale) in TBS-T (TBS supplemented with 0.1% [v/v] Tween-20) for 60 min at ambient temperature. Next, membranes were washed twice in TBS-T and incubated with primary antibodies at 4 °C o/n. Primary antibodies were diluted in antibody dilution buffer (5% [w/v] BSA [pH 5.2, Sigma-Aldrich] in TBS-T

with 0.04% [v/v] sodium azide) as per the manufacturer's instructions. Membranes were washed thrice for 5 min each in TBS-T. Next, secondary anti-rabbit IgG antibody (1 in 10,000, Jackson ImmunoResearch; 111-035-003) conjugated with HRP diluted in 5% (w/v) skim milk in TBS-T was incubated with the membranes for 60 min at ambient temperature. β-actin antibody conjugated to HRP (Santa Cruz; sc-47778 HRP) was used at a 1 in 20,000 dilution in 5% (w/v) skim milk in TBS-T for 40 min at ambient temperature as a sample loading control on all blots. Next, membranes were washed four times for 5 min each in TBS-T. Membranes were developed in ECL (Bio-Rad) and imaged using the ChemiDoc imaging system (Bio-Rad). Images were processed using Image Lab software.

### Protein extraction and digestion (ubiquitinomics)

Cells were harvested directly off the cell culture plates using pre-heated sodium deoxycholate (SDC) buffer (1% [v/v] SDC, 10 mM TCEP, 40 mM 2-chloracetamide [CAA], 75 mM Tris-HCl at pH 8.5). Samples were incubated at 85 °C for 10 min and allowed to cool to ambient temperature prior to the addition of universal nuclease (Thermo Fisher Scientific) and incubated for 5 min prior to centrifugation, and the clarified lysates were collected. Protein concentrations were determined using the Pierce™ BCA Protein Assay Kit (Thermo Fisher Scientific), and proteins were digested with Lys-C (Wako, 129–02541) and TPCK-treated Trypsin (Thermo Fisher Scientific, 20233) mix o/n at 37 °C with a 1:50 enzyme to protein ratio. As described previously, the digestion was stopped by adding three volumes 1% (v/v) trifluoroacetic acid (TFA; Sigma-Aldrich) in 2-propanol and loaded onto SDB-RPS cartridges (Strata™-X-C, 100 mg, Phenomenex Inc.; Hansen et al, 2021). Columns were activated and pre-equilibrated with 3 mL of 30% (v/v) MeOH/1% (v/v) TFA and washed with 3 mL of 0.2% (v/v) TFA. Samples were loaded and washed twice with 3 mL 1% (v/v) TFA in 2-propanol and once with 3 mL 0.2% (v/v) TFA / 2% (v/v) ACN. Peptides were eluted twice with 2 mL of 1.25% (v/v) NH$_4$OH/80% (v/v) ACN and diluted with H$_2$O to a final ACN concentration of 30% (v/v). The eluates were snap-frozen in liquid nitrogen and lyophilised o/n.

### Crosslinking of K-GG antibody

Crosslinking was performed as described previously (Udeshi et al, 2013) using the K-GG antibody licensed to Cell Signaling Technology (PTMscan® Pilot Ubiquitin Remnant Motif K-GG kit #14482). Antibody-bound beads were washed thrice with 100 mM sodium tetraborate (pH 9.0) and then crosslinked for 30 min at ambient temperature with 0.5 mL of 20 mM dimethyl pimelimidate in 100 mM sodium borate (pH 9.0). The crosslinking reaction was then quenched by the addition of 200 mM ethanolamine (pH 8.0) and washed twice before being incubated for 2 h with 200 mM ethanolamine (pH 8.0). The beads were washed thrice with immunoaffinity purification (IAP) buffer (50 mM MOPS [pH 7.2], 10 mM Na$_2$HPO$_4$, 50 mM NaCl).

### K-GG peptide enrichment and LC-MS/MS sample preparation

K-GG peptide enrichment was performed as described previously with some minor modifications (Udeshi et al, 2013). Briefly, peptides were resuspended in 0.5 mL of cold IAP buffer and incubated with 2.5 µL of packed crosslinked K-GG antibody-bead

conjugate (corresponding to 31 µg of antibody per sample) for 2 h at 4 °C with end-over-end rotation. Beads were transferred to glass-fibre filter stage tips and washed twice with 1 mL of IAP buffer and an additional three times with cold $H_2O$ via centrifugation between each wash (Hansen et al, 2021). The glass-fibre filter stage tips were then stacked onto SDB-RPS stage tips. The SDB-RPS stage tips were pre-activated and equilibrated with the addition of 60 µL of 2-propanol, 60 µL 80% (v/v) ACN and 100 µL 0.2% (v/v) TFA with centrifugation occurring between each addition. K-GG peptides were eluted off the beads and directly captured onto the SDB-RPS stage tips via two separate 100 µL of 0.15% (v/v) TFA elutions and centrifugation. Peptides were then desalted by washing the SDB-RPS stage tips with 0.2% (v/v) TFA/2% (v/v) ACN, prior to elution with 60 µL 80% (v/v) ACN/2.5% (v/v) $NH_4OH$ directly into level 3 SureStart 0.2 mL MS vials (Thermo Fisher Scientific). Peptides were Speedvac (Thermo Fisher Scientific) dried and then resuspended in 10 µL of 0.1% (v/v) FA/2% (v/v) ACN, with 4 µL injected into the mass spectrometer.

### Protein extraction and digestion

For global proteomics, cell pellets from one well of a 6-well plate per biological repeat ($n = 5$) were lysed in 200 µL of pre-heated (95 °C) buffer (2.5% [v/v] SDS in 100 mM Tris-HCl, pH 8.5). DNA was hydrolysed with the addition of 2 µL neat TFA, and lysates were neutralised to pH 8.5 by the addition of 1 M Tris-HCl as previously described (Dagley et al, 2019). Protein concentration was determined using Pierce™ BCA Protein Assay Kit (Thermo Fisher Scientific) following the manufacturer's instructions. For DUB IP-MS, samples were processed using S-Trap spin columns (Profiti) by following the manufacturer's instructions. Briefly, samples were reduced with 5 mM TCEP and alkylated by addition of 20 mM CAA with incubation at 55 °C for 15 min each prior to addition of 2.5% (v/v) phosphoric acid. In-column digestion was performed with a Lys-C (Wako, 129–02541) and SOLu-Trypsin (Sigma-Aldrich, EMS0004) at 1 µg per column for 1:45 h at 47 °C. For global proteomics, cell lysates (20 µg protein per replicate) were transferred to 0.5 mL LoBind deep well plate (Eppendorf) prepared for MS analysis using the modified SP3 protocol (Hughes et al, 2019), with some modifications. Briefly, samples were subjected to simultaneous reduction and alkylation with a final concentration of 10 mM TCEP and 40 mM 2-chloracetamide (CAA) followed by heating at 95 °C for 10 min. Prewashed magnetic PureCube Carboxy agarose beads (20 µL, Cube Biotech) were added to all the samples along with ACN (70% [v/v] final concentration) and incubated at ambient temperature for 20 min. Samples were placed on a magnetic rack and supernatants were discarded, and beads were washed twice with 70% (v/v) ethanol and once with neat ACN. ACN was completely evaporated from the tubes using a CentriVap (Labconco) before the addition of digestion buffer (50 mM Tris-HCl, pH 8) containing 1 µg each of enzymes Lys-C (Wako, 129–02541) and SOLu-Trypsin (Sigma-Aldrich, EMS0004). Trypsin-LysC on-bead digestion was performed with agitation (400 rpm) for 1 h at 37 °C on a Thermo-Mixer C (Eppendorf). For DUB IP-MS and global proteomics, the samples were transferred to pre-equilibrated C18 StageTips (2× plugs of 3 M Empore resin, no. 2215) following digestion for sample clean-up. The eluates were lyophilised to dryness before being reconstituted in 150 µL 0.1% (v/v) FA/2% (v/v) ACN ready for MS analysis.

### LC-MS/MS measurements

Peptides were loaded on a 15-cm C18 fused silica column with an integrated emitter tip (IonOptics, ID 75 µm, OD 360 µm, 1.6 µm C18 beads), which was maintained at 50 °C using a column oven. DUB IP-MS was performed on a Neo Vanquish (Thermo Fisher Scientific) directly coupled online with the mass spectrometer (Eclipse, Thermo Fisher Scientific) and peptides were separated with a binary buffer system of buffer A (0.1% [v/v] FA) and buffer B (99.9% [v/v] ACN plus 0.1% [v/v] FA), at a flow rate of 250 nL/min. The gradient started at 2% (v/v) B and increased to 34% (v/v) B in 80 min before increasing to 85% (v/v) B within 3 min and held for 15 min prior to returning to 2% (v/v) B and re-equilibration. The mass spectrometer was operated in positive polarity mode with a capillary temperature of 275 °C. The data-independent acquisition (DIA) methods consisted of a MS1 scan ($m/z = 350$–1650) with an automatic gain control (AGC) target of $1.2 \times 10^6$ and a maximum injection time of 60 ms ($R = 120,000$). DIA scans were acquired at $R = 30,000$, with an AGC target of $3 \times 10^5$, 'auto' for injection time and a default charge state of 3. The spectra were recorded in profile mode, and the stepped collision energy was 25, 27.5, 30% normalised collision energy. 44 non-uniform DIA segments were set to achieve an average of five data points per peak. K-GG samples and siRNA KD samples were acquired with a Neo Vanquish (Thermo Fisher Scientific) directly coupled online with an Astral mass spectrometer (Thermo Fisher Scientific) and peptides were separated with a binary buffer system of buffer A (0.1% [v/v] FA) and buffer B (80% [v/v] ACN plus 0.1% [v/v] FA), at a flow rate of 400 nL/min. The gradient started at 2% B and increased to 34% B in 30 min before increasing to 100% (v/v) B within 0.1 min and held for 3 min prior to returning to 2% (v/v) B and re-equilibration. The mass spectrometer was operated in positive polarity mode with a capillary temperature of 275 °C. The DIA methods consisted of a MS1 scan ($m/z = 380$-980) with an AGC target of $5 \times 10^6$ and a maximum injection time of 5 ms ($R = 240,000$). DIA scans were acquired with the Astral detector with an AGC target of $8 \times 10^4$ and 3 ms maximum time. Fragmentation occurred in the higher-energy collisional dissociation (HCD) cell with a normalised stepped collision energy was 25% and the spectra were recorded in profile mode. 199 non-uniform DIA windows across 380-980 were collected with a maximum injection time of 3 ms and a 0.6 s loop control, which achieved an average of five data points per peak. Global proteomic samples were acquired using a custom nano-flow high-performance liquid chromatography (HPLC) system (Thermo Fisher Scientific Ultimate 300 RSLC Nano-LC, PAL systems CTC autosampler). The HPLC was coupled to a timsTOF Pro (Bruker) equipped with a CaptiveSpray source. Peptides were loaded directly onto the column at a flow rate of 600 nL/min with buffer A (99.9% Milli-Q $H_2O$, 0.1% [v/v] FA) and eluted at 400 nL/min on a 30 min linear analytical gradient of increasing buffer B (90% [v/v] ACN, 0.1% [v/v] FA) from 2% (v/v) to 34% (v/v). The timsTOF Pro MS was operated in diaPASEF mode using Compass Hystar 5.1. The settings on the thermal ionisation MS (TIMS) analyser were as follows: Lock Duty Cycle to 100% with equal accumulation and ramp times of 100 ms, and 1/K0 Start 0.6 Vs/cm² End 1.6 Vs/cm², Capillary Voltage 1400 V, Dry Gas 3 L/min, Dry Temp 180 °C. The DIA methods were set up using the instrument firmware (timsTOF control 2.0.18.0) for data-independent isolation of multiple precursor windows within a

single TIMS scan. The method included two windows in each diaPASEF scan, with window placement overlapping the diagonal scan line for doubly and triply charged peptides in the $m/z$ – ion mobility plane across $16 \times 25$ $m/z$ precursor isolation windows (resulting in 32 windows) defined from $m/z$ 400 to 1200, with 1 Da overlap, and collision induced dissociation (CID) collision energy ramped stepwise from 20 eV at 0.8 Vs/cm$^2$ to 59 eV at 1.3 Vs/cm$^2$.

### Raw data searching and analysis

MS raw files were processed using DIA-NN 1.8.1 as described previously (Steger et al, 2021). Global and DUB IP-MS raw files were searched with the following settings: Trypsin specificity, peptide length of 7–30 residues, cysteine carbamidomethylation enabled as a fixed modification, variable modifications set to N-terminal protein acetylation and oxidation of methionine, the maximum number of missed cleavages at one (DUB IP-MS) and two (global proteomics) and match between runs (MBR) enabled within individual cell lines. For DUB IP-MS, modification UniMod:35 with mass delta 15.9949 at M was considered as a variable. For K-GG experiments, library-free searching with K-GG variable modification enabled with a maximum of two modifications and two-missed cleavages was performed with the reviewed human proteome (Uniprot), with MBR enabled and Robust LC quantification. Precursors were consolidated by retaining the precursor that was detected in the largest number of samples and had the highest intensity. Data were filtered for K-GG peptides that were detected in four out of five samples in at least one condition (unless specified otherwise) and differential expression analysis performed using the DEP R package (Zhang et al, 2018). Data were normalised with variance stabilising normalisation (VSN) and missing values imputed using a mixed Bayesian principal component analysis (BPCA) and Min method (unless stated otherwise in the figure legends). Significance testing of log2-transformed intensities was performed, and Benjamini–Hochberg corrected $P$ values $< 0.05$ were considered significant. K-GG fold changes were corrected for changes in the proteome unless stated otherwise.

Downstream processing and differential expression analysis were performed in R using limma (Ritchie et al, 2015) and DEP packages. For DUB IP-MS and global proteomics, quantities were determined using MaxLFQ (Cox et al, 2014b) for proteins with at least two peptides; data were filtered for proteins that were observed in at least four out of five replicates in at least one condition. VSN normalised and imputed using missing values imputed using a mixed BPCA and MinD method. For DUB IP-MS, Benjamini–Hochberg corrected $P$ values $< 0.05$ were considered as significant and DUBs were considered responders if at least 20% capture was lost upon Ub-VS pre-treatment. For global proteomics samples, significance testing of log2-transformed intensities was performed, and Q-values from limma $<0.05$ were considered significant.

### Lattice light sheet imaging

MDA-MB-231, MiaPaca2, HMEC-1, SK-MEL-2 and UO-31 cells were seeded at $2-15 \times 10^4$ cells/mL in 8-well glass μ-slides (ibidi). The next day, cells were treated with 15 μM WEHI-092/DMSO. SPY650-Tubulin, SPY555-DNA (both Spirochrome, Inc.), and Annexin V FITC (Sigma-Aldrich) were added at a final concentration of 1 in 1000. Cells were left to grow for 24 h before time-lapse live-cell data were acquired using a Lattice Light Sheet 7 (Zeiss). Light sheets (488, 561 and 640 nm) of 30 μm in length with a thickness of 1 μm were created at the sample plane via a $13.3 \times 0.44$ numerical aperture (NA) objective. Fluorescence emission was collected via a $44.83 \times 1$ NA detection objective via a multiband stop, LBF 405/488/561/633, filter. Aberration correction was set to a value of 170 to minimise aberrations as determined by imaging the Point Spread Function using 100 nm fluorescent microspheres. Data were collected with a frame time of 10 ms and a z step of 0.4 μm with 834 frames per 290 μm by 290 μm. Individual regions were imaged in parallel across six wells of the eight-well chamber slide. Data are presented as Maximum Intensity Projections (MIPs), and these MIPs were used for all downstream analysis. Tile regions were imaged for 24 h with 10 min intervals at 37 °C, 5% $CO_2$. Analysis was performed using the MIPs of the different full fields of cells over the 24–48 h time period. Mitosis and mitosis arrest were classified through a combination of automated image analysis and manual validation. MIPs were processed using the TrackMate plugin in ImageJ/Fiji (Schindelin et al, 2012; Ershov et al, 2022). Segmentation was performed using the pre-defined StarDist model (Schmidt et al, 2018) on the SPY555-DNA channel and followed by an auto-threshold using the 'Quality' filter. Small erroneous detected spots were further removed through a manually adjusted size filter. The remaining segmented nuclei were tracked using an Advanced Kalman Tracker capable of splitting and merging tracks, which is necessary for tracking dividing cells. An initial search radius of 30 μm followed by a further search radius of 20 μm and a max frame gap of two were set. The mean intensity of all tracks was then measured to determine mitosis and mitosis arrest. Mitosis arrest was defined when a cell's nuclei increased in mean intensity but did not progress through to division and remained with a high mean intensity value, indicating condensed chromatin. Successful mitosis was defined by cells whose mean intensity increased leading into mitosis, but was followed by a splitting in the subsequent tracks. All identified mitosis and mitosis arrest events were manually validated, and the frame number indicating the entry of mitosis was noted for all events. All videos were generated using ImageJ/Fiji.

### In silico analyses

An oncoprint for the NCI-60 cancer cell line panel was generated using copy-number variation (CNV) and mutation data obtained from cBioPortal (cbioportalR study ID: cellline_nci60). Genes included in the oncoprint were those listed in the catalogue of somatic mutations in cancer (COSMIC) Hallmarks of Cancer Genes dataset (v103, GRCh37; https://cancer.sanger.ac.uk/cosmic/download/cosmic/v103/hallmarksofcancer) with an alteration frequency greater than 10%, and *USP9X* was additionally included. Each gene was assigned a pathway annotation based on the MSigDB Hallmark Gene Set collection (https://www.gsea-msigdb.org/gsea/msigdb/human/collections.jsp). Cell line annotations, including cancer type, sex, and doubling time, were obtained from the CellMiner NCI-60 database (https://discover.nci.nih.gov/cellminer/celllineMetadata.do). Gene expression data for selected cancer cell lines were obtained from the CellMiner RNA-seq - composite expression dataset (https://discover.nci.nih.gov/cellminer/loadDownload.do;jsessionid=

7C8077569EE7FF4B612D44300AD1BC97). Z-score of log2 (fragments per kilobase per million reads [FPKM] + 1) normalisation was performed to allow relative comparison of gene expression levels between cell lines. The oncoprint and gene expression heatmap were generated in R.

### Graphics

Some of the graphics in Fig. 3A, Fig. 4A and Fig. 5B, as well as the synopsis image have been created with BioRender.com.

## Data availability

The datasets produced in this study have been deposited to the proteomics identifications (PRIDE) archive database (Perez-Riverol et al, 2024; https://www.ebi.ac.uk/pride/) under the following identifiers: DUB IP-MS: PXD074545. Global proteomics cancer cell line profiling for A549, DLD-1, HMEC-1, OVCAR8, SK-MEL-2, UO-31 cells: PXD062674; and MDA-MB-2131, MiaPaca2 cells: PXD074729. Global proteomics USP9X siRNA KD: PXD062674. HDX-MS: PXD062754. Ubiquitinomics with matched global proteomics: DLD-1 and MDA-MB-231 PXD073869 (K-GG) / PXD074673, PXD075275 (matched global proteomics). SK-MEL-2 PXD073775 (K-GG) / PXD073798 (matched global proteomics). UO-31 PXD073754 (K-GG) / PXD073798 (matched global proteomics).

The source data of this paper are collected in the following database record: biostudies:S-SCDT-10_1038-S44318-026-00742-y.

## Peer review information

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

## Acknowledgements

The authors would like to thank past and present members of the Ubiquitin Signalling Division at WEHI, especially Marlene F Schmidt for help with statistics. We thank Sylvie Urbé and Michael J Clague (University of Liverpool) for critical comments on the manuscript, and Kum Kum Khanna (Mater Research, Queensland) for early discussions on CEP55. We thank the National Drug Discovery Centre (WEHI, Parkville) for early compound testing. We thank Yelena Khakham for obtaining HRMS data. HDX-MS data were collected at the Bio21 proteomics facility. Proteomics data were collected at the WEHI proteomics facility. Imaging data was collected at the WEHI Centre for Dynamic Imaging. NCI-60 cancer cell panel testing was performed at the NCI-DTP. We thank the Centre for Drug Candidate Optimisation (CDCO), Monash Institute for Pharmaceutical Sciences for performing the Caco-2 permeability study and Ubiquigent Ltd. for performing the DUB panel analysis. PS is supported through a Melbourne Research Scholarship at the University of Melbourne. The National Drug Discovery Centre is supported by the Australian Government Medical Research Future Fund (MRFF) Grant ID EPCD000033, and the Victorian Government. This work was carried out during the tenure of a Cancer Research Project Grant from the Future Health Research and Innovation Fund and Cancer Council Western Australia, CCWA 2025/1401 to PJAE. APN is supported through a Leukaemia Foundation Breakthrough Fellowship. The laboratory of RF is supported by The Galbraith Family Charitable Trust, the K & M Foundation for Women, the Betty Deller King Bequest, Denise and Roberto Cappai, John and Tibby Peterson, the Rae Foundation and the Berwick Opportunity Shop. This work has been supported by an NHMRC Investigator Grant GNT117812 to DK.

## Author contributions

**Philipp Schenk**: Conceptualisation; Resources; Data curation; Formal analysis; Validation; Investigation; Visualisation; Methodology; Writing—original draft; Project administration; Writing—review and editing. **Shane M Devine**: Resources; Data curation; Formal analysis; Supervision; Funding acquisition; Validation; Investigation; Visualisation; Methodology; Writing—original draft; Writing—review and editing. **Simon A Cobbold**: Resources; Data curation; Software; Formal analysis; Supervision; Validation; Investigation; Methodology; Writing—review and editing. **Niall D Geoghegan**: Resources; Formal analysis; Investigation; Visualisation; Methodology; Writing—review and editing. **Elizabeth L Kyran**: Resources; Data curation; Formal analysis; Investigation; Visualisation; Writing—review and editing. **Ching-Seng Ang**: Data curation; Formal analysis; Investigation; Methodology; Writing—review and editing. **Jack A Alexandrovics**: Investigation. **Dale J Calleja**: Resources; Investigation; Methodology; Writing—review and editing. **Dylan H Multari**: Resources; Investigation; Methodology. **Vineet Vaibhav**: Resources; Investigation; Methodology. **Bernadine GC Lu**: Resources; Formal analysis; Investigation; Methodology. **Theresa A Klemm**: Resources; Formal analysis; Methodology. **Laura F Dagley**: Resources; Investigation; Methodology. **Kym N Lowes**: Resources; Methodology; Project administration. **Nicholas A Williamson**: Resources; Methodology; Project administration. **Pieter JA Eichhorn**: Conceptualisation; Writing—review and editing. **Ashley P Ng**: Conceptualisation; Resources; Data curation; Formal analysis; Supervision; Methodology; Writing—review and editing. **Rebecca Feltham**: Conceptualisation; Resources; Supervision; Funding acquisition; Project administration; Writing—review and editing. **David Komander**: Conceptualisation; Supervision; Funding acquisition; Writing—original draft; Project administration; Writing—review and editing.

Source data underlying figure panels in this paper may have individual authorship assigned. Where available, figure panel/source data authorship is listed in the following database record: biostudies:S-SCDT-10_1038-S44318-026-00742-y.

## Disclosure and competing interests statement

DK is the founder, shareholder and SAB member of Entact Bio and Proxima Bio, and co-founder and SAB member of Ternarx. RF is co-founder and scientific lead at Ternarx. APN is co-founder and clinical advisor at Ternarx.

# Expanded View Figures

**Figure EV1.  Time resolved ubiquitome profiling of WEHI-092-sensitive cell lines.**

(**A**) Time resolved profiling of high-confidence USP9X substrates in SK-MEL-2 cells. Venn diagram showing the overlap of proteins with enhanced ubiquitination at 30 min (log2 >twofold, adjusted *P* value < 0.05 over untreated control) and decreased abundance at either 30 min, 360 min or 24 h WEHI-092 treatment (10 μM; log2 <−0.585, adjusted *P* value < 0.05 over untreated control) across four or five biological repeats per condition. Heatmap colours indicate fold change in protein ubiquitination (left) and protein expression (right) of proteins that showed significant induction (fold change log2 > 2, adjusted *P* value < 0.05 over untreated control) of at least one ubiquitination site at 30 min of WEHI-092 treatment (10 μM) and that were significantly downregulated (fold change log2 <−0.585, adjusted *P* value < 0.05 over untreated control) at 30 min/360 min/24 h. The data were averaged across four or five biological repeats per condition, and the data were matched based on gene ID level. Hierarchical clustering was performed on proteins (rows) with Euclidean distance as the similarity metric. (**B**) As in (**A**) but for UO-31 cells. Source data are available online for this figure.

▶

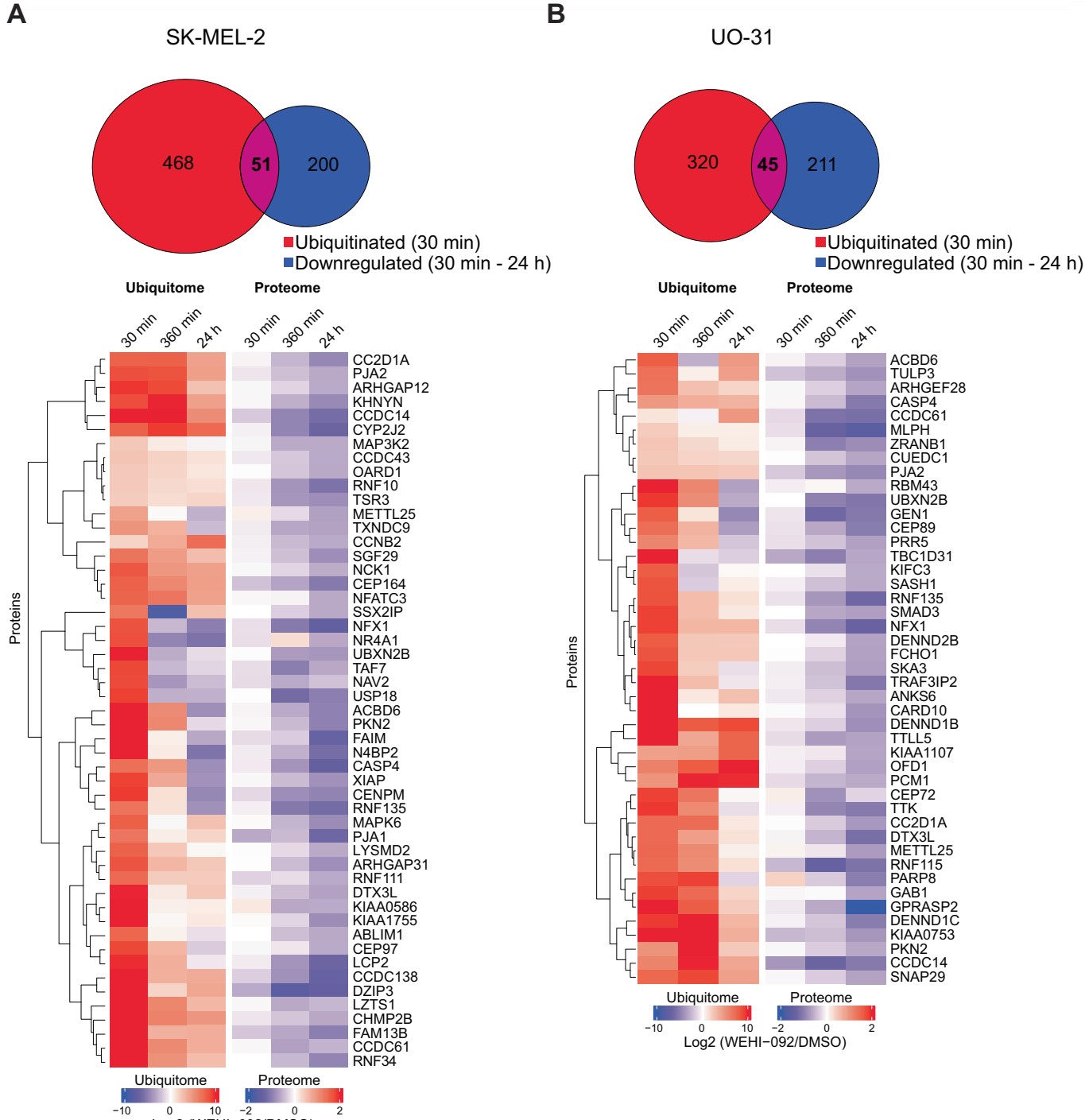

