## [Peer Review File · The EMBO Journal]

Global analysis of cancer cell responses to USP9X inhibition

Philipp Schenk, Shane Devine, Simon Cobbold, Niall Geoghegan, Elizabeth Kyran, Ching-Seng Ang, Jack Alexandrovics, Dale Calleja, Dylan Multari, Vineet Vaibhav, Bernadine Lu, Theresa Klemm, Laura Dagley, Kym Lowes, Nicholas Williamson, Pieter Eichhorn, Ashley Ng, Rebecca Feltham, and David Komander

Corresponding author(s): David Komander (dk@wehi.edu.au)

Review Timeline:

Submission Date:	9th Jun 25
Editorial Decision:	3rd Jul 25
Revision Received:	19th Dec 25
Editorial Decision:	19th Jan 26
Revision Received:	19th Feb 26
Accepted:	20th Feb 26

Editor: Hartmut Vodermaier

Transaction Report:

Prof. David Komander
The Walter and Eliza Hall Institute of Medical Research
Ubiquitin signalling
1G Royal Parade
Parkville
Melbourne, Victoria 3052
Australia

3rd Jul 2025

Re: EMBOJ-2025-121586
Global analysis of cancer cell responses to USP9X inhibition

Dear David,

Thank you for submitting your manuscript on global analysis of USP9X inhibition effects to The EMBO Journal. We have now received reports from three expert referees, copied below for your information. As you will see, the referees appreciate the importance of the subject and the comprehensive analysis, but they also all raise a number of issues and open questions that would need to be addressed prior to publication. These concerns include experimental descriptions, reproducibility and repeats, as well as extension/validation of key results in the most relevant settings.

Should you be able to adequately address these various points, we would be interested in considering a revised manuscript further for publication in our journal. Please be reminded, however, that our single-major-revision-round policy makes it important to diligently respond to each referee point at the time of resubmission; therefore, please do not hesitate to contact me early on in case you would like to clarify any of the referees' points, or discuss plans for how to answer them. We would also be open to extending the revision deadline if that should be helpful. Our scooping protection policy means that competing manuscripts published while your work is under revision will not have a negative effect on our final decision.

Further information on preparing, formatting and uploading a revised manuscript can be found below and in our Guide to Authors. Thank you again for the opportunity to consider this work for The EMBO Journal, and I look forward to receiving your revised manuscript in due time.

With kind regards,

Hartmut

9) To facilitate reproducibility and cross-laboratory adoption of methodologies, please structure the Materials & Methods section as outlined in our guide to authors, including a completed Reagents and Tools Table that can be downloaded from our author guidelines as well (<https://www.embopress.org/page/journal/14602075/authorguide#structuredmethods>).

10) Digital image enhancement is acceptable practice, as long as it accurately represents the original data and conforms to community standards. If a figure has been subjected to significant electronic manipulation, this must be clearly noted in the figure legend and/or the 'Materials and Methods' section. The editors reserve the right to request original versions of figures and the original images that were used to assemble the figure. Finally, we generally encourage uploading of numerical as well as gel/blot image source data; for details see: embopress.org/page/journal/14602075/authorguide#sourcedata

In the interest of ensuring the conceptual advance provided by the work, we recommend submitting a revision within 3 months (1st Oct 2025). Please discuss the revision progress ahead of this time with the editor if you require more time to complete the revisions. Use the link below to submit your revision:

Link Not Available

Referee #1:

Summary and General Assessment

This manuscript describes the development of a novel USP9X inhibitor (WEHI-092) and employs a broad proteomic and cellular profiling approach to characterize its effects. The study presents a technically rich dataset and offers new insights into USP9X biology and selectivity. The use of HDX-MS, AlphaFold3 modeling, and a multi-cell line screening strategy is commendable. However, several conceptual and technical concerns should be addressed to strengthen the validity of the conclusions and ensure the manuscript meets the high standards of The EMBO Journal.

Comments

1. The manuscript performed at least limited SAR studies to achieve WEHI-092. It will be beneficial to discuss if/how the SAR is consistent with the determined binding mode.
2. The cellular activity advantage of WEHI-092 over FT709 (e.g., in UO-31 cells) is noted but not mechanistically explained. Differential permeability, metabolic stability, or off-target effects could underlie this result and should be acknowledged. Binding affinity and in vitro potency data actually favor FT709 (KD = 69 nM vs. 1 μM; IC50 = 124 nM vs. 254 nM), which complicates the claim of comparable or superior efficacy.
3. Substrate validation requires additional support. Core substrates such as CEP55 and SMAD3 are proposed based on

proteomics and ubiquitinomics, but causal validation is limited. Additional MG132 rescue experiments and/or direct ubiquitination assays are needed for at least 1-2 more targets to substantiate claims of direct USP9X regulation.

4. Determinants of cell line sensitivity are not explored. The proteomic heterogeneity across lines is intriguing but unexplained. Were differences in USP9X expression, substrate levels, or mitotic index correlated with inhibitor sensitivity?

5. Statistical inference in proteomic data is absent from some analyses (e.g., Figure 1). Where key conclusions are drawn from relative abundance changes, significance testing should be applied and visualized to help distinguish biological from technical variation. Additionally, different fold-change thresholds are used without explanation or justification throughout. For example, the LFC threshold of 0.3 represents a small change (a FC of ~1.2) which risks capturing noise or experimental variability, especially where the experiment includes technical replicates only and no biological replicates are presented.

6. Figure S1E refers to a "Control compound" which is not defined in the figure, legend or text. Later experiments use WEHI-680 as a control, but this is not at all defined in the context of Figure S1E. It would be helpful to have WEHI-092 and FT709 on the same blot for comparison as well.

7. The manuscript includes WEHI-680 (IC₅₀ 4.3 μM) as a comparator compound. The rationale for selecting this compound and the value of this comparison is unclear. It is being used in these experiments at concentrations exceeding its IC₅₀, so the phenotypic difference isn't necessarily reflective of strength of USP9X binding (side-by-side target engagement could be revealing here). The manuscript should clearly state the rationale for selecting WEHI-680 as a comparator, and if the authors view the results with this compound as being consequential to the study, augment the discussion by interpretation of the results obtained with this compound

8. The Discussion includes speculative questions (e.g., about USP9X's role as a rheostat or spatial organizer) but does not synthesize a model or clearly articulate the new biological insights. These should be revised for clarity on advances and enablement made by the study. Similarly, the suggestion that this study "consolidates the literature" is vague--please specify what previous discrepancies are addressed and what overarching model emerges from this work.

9. The conclusion that profiling six cell lines resolves the variable reports on USP9X function may be overstated. While the authors appropriately highlight heterogeneity, this limited sample cannot fully explain prior inconsistencies across diverse models. This claim should be softened or caveated.

Referee #2:

In the presented manuscript did Schenk et al. present the development and validation of a novel and highly selective DUB inhibitor, WEHI-, which selectively targets USP9X. The group of David Komander used a plethora of tools, including synthesis, DUB explorer assays, activity-based probe analysis, tissue culture, mass spectrometry and immunofluorescent imaging to develop this inhibitor and identify the physiological consequence of USP9X interference.

The enzyme family of deubiquitylases is an exciting and pharmacological underrepresented group of drug targets and every effort to develop new and highly selective drugs is needed, since the clinical data and pre-clinical models suggests DUBs to be an excellent cancer vulnerability. The possibility to target a novel member of this enzyme family in a well defined and stratified patient cohort could be helpful in the future. To this extend is the reported inhibitor a great tool for continued drug development. The manuscript is well written and the experiments very timely and well carried out. The reviewer would like to suggest deepening the understanding of the biological cellular processes upon exposure to USP9X, especially given the wide spectrum of cellular phenotypes observed.

In detail:

Figure 3: The authors used a panel of human cell lines and observed various degrees of cellular responses, ranging from growth inhibition to cell death, while others showed a profound increase in proliferation, such as HL-60T or COLO-205, . It is intriguing that among the cell lines tested SR and UO-31 showed a striking response, along with the AML derived cell line THP-1, which also presented a robust response to the USP9X inhibitor. Could the authors please elaborate on the genetic alterations present in these cells? Is there a gender-bias in the cellular responses observed?

Furthermore, based on public available data, does the expression of USP9X show a wide range in regard to correlation of patient survival. Having the cellular responses to USP9X inhibition available, do the observed cellular responses match the tumour behaviour, meaning, do tumours where USP9X levels would be indicative of poor patient survival show an anti-proliferative phenotype in cellulo?

Figure 4: Next do the authors go into great length to compare the proteome of USP7 or USP9X inhibited cell lines DLD-1 and MDA-MB-231. While this is exciting to observe and the resolution points towards a rapid response in deregulated protein abundance, especially in the upregulated fraction, while the downregulated fraction, at least in MDA-MB-231, is more persistent. The reviewer is wondering if the same was observed for DLD-1 and do the authors think that the pattern would be consistent in other cell lines as well? How does this pattern look like in cell lines that showed a low or no proliferative phenotype? Would this be altered as well?

Since the described novel inhibitor WEHI-092 is extensively used in this manuscript it is not obvious for this reviewer if this inhibitor was immobilized on beads and used for a global IP-MS approach to investigate its putative specificity on a more wider range of proteins? To validate the on target effects of the proposed small molecule inhibitor could the authors please use e.g. a knock down or acute knock out strategy, such as si/shRNA or the Auxin induced degradation model, followed by MS analysis, for cross-validation in 1 or two exemplary cell lines? This would greatly assist in the validation of the biological effects observed by cells exposed to WEHI-092.

Figure 5 and S6: Could the authors please confirm by TUBE, DSK or other means of ubiquitin pulldown that the proposed

targets of USP9X, such as CEP55 or EDF1, are indeed polyubiquitylated upon loss of USP9X? Do the authors suggest that the mentioned proteins are direct targets and interaction partners of USP9X? Do these factors, namely EDF1, CEP55, EDF1 or MKRN2, co-immunoprecipitate with USP9X?

Figure 7: Could the authors please share their thoughts if this mode of action is causative to the strong phenotypes observed in some cell lines, like THP1 or UO-31? Since the statement is very broad, that the inhibition of USP9X arrests cancer cells in metaphase, most cell lines in Figure 3 and S4, showed mild responses to high concentrations of the inhibitor. Could the authors therefore please either elaborate more on the commonality or generalism of this observed arrest, or rephrase the statement, reflecting that a selective number of cell lines halted proliferation?

Minor points:

Could the authors please elude on the specificity of the compounds towards USP9Y?

Do the cell lines used express both variants of USP9? Does USP9X undergo X-chromosome escape?

Does the origin of sex of the cancer cell lines used impact on the susceptibility towards the inhibitor?

Some passages of the manuscript have a connotation comparable to a review, such as:

>> Some of the discrepancies in the literature arise from an undue focus on single-protein relationships (USP9X regulates target X), which was sometimes exaggerated by inappropriate tools. More than 30 papers have assigned the effects of Degrasyn/WP1130 and its derivatives G9/EOAI3402143 (Bartholomeusz et al, 2007a, 2007b) to USP9X inhibition, despite data that WP1130 is a non-specific DUB inhibitor (Ritorto et al, 2014; Kapuria et al, 2010; Peterson et al, 2015).<<
These parts could be moved to the discussion part or rephrased.

Referee #3:

The authors previously reported the identification of a USP9X inhibitor (TF-709). In this study, they present a new-generation USP9X inhibitor, WEHI-092. Its specificity is supported by in vitro deubiquitylation assays, and structural analyses provide insights into its mechanism of action. Functional assays show that WEHI-092 selectively inhibits growth in certain cell lines. Proteomic analyses of global ubiquitylation and protein expression changes further identify potential USP9X-regulated substrates. While the study offers interesting findings, several critical issues need to be addressed.

Note: This reviewer is not an expert in structural biology and has not evaluated that portion of the manuscript.

Major comments:

1. The authors should more clearly demonstrate the reproducibility of their proteomic analyses of ubiquitylation and protein abundance. The number of biological replicates should be explicitly stated. More importantly, quantitative reproducibility between replicates must be shown. In the results section, the authors note that "... the number of ubiquitinated proteins exceeded the number of subsequently degraded proteins, by at least ~6-fold. This observation may indicate either non-degradative (e.g. signalling) ubiquitination events, or ubiquitination events that are not yet functional to induce proteasomal degradation."

However, an alternative explanation is that the false discovery rate is higher for ubiquitylated peptides. Site-level quantification relies on single peptides, while protein-level measurements aggregate multiple peptides, making the latter more robust to noise. This could account for the limited overlap observed and should be addressed explicitly.

Also, what is the difference between "non-degradative (e.g. signalling) ubiquitination events" and "ubiquitination events that are not yet functional to induce proteasomal degradation?"

2. The authors report that USP9X inhibition most strongly impairs growth in MOLT4, SK-MEL-2, SR, and UO-31 cells. Yet, ubiquitylation profiling was performed in DLD1 and MDA-MB-231 cells, which are not among the most sensitive lines. The rationale is unclear. In any case, it would be more informative to perform these analyses in at least two cell lines that exhibit strong growth inhibition in response to USP9X inhibition, to better connect ubiquitylation changes with phenotypic effects.

3. The authors state: "...high-confidence USP9X substrates found in MDA-MB-231 cells, 58 were detected in DLD1 cells at the peptide level, but were seemingly not significantly altered by USP9X inhibitors during the experiment."

This could indicate that the same sites are regulated by different DUBs in different contexts or, alternatively, that measurement noise impact quantification. To clarify this, the authors should report the reproducibility of regulated sites across biological replicates in both cell lines. The number of sites regulated in individual experiments, and their overlap should be shown. This would help distinguish biological variability from quantification noise.

Minor comment:

The authors mention the use of an in-house di-Gly antibody but provide no reference or information on the antibody's source or properties (e.g., monoclonal/polyclonal, host species). This information should be included to ensure experimental reproducibility.

Referee #1:Summary and General Assessment

This manuscript describes the development of a novel USP9X inhibitor (WEHI-092) and employs a broad proteomic and cellular profiling approach to characterize its effects. The study presents a technically rich dataset and offers new insights into USP9X biology and selectivity. The use of HDX-MS, AlphaFold3 modeling, and a multi-cell line screening strategy is commendable. However, several conceptual and technical concerns should be addressed to strengthen the validity of the conclusions and ensure the manuscript meets the high standards of The EMBO Journal.

We thank the reviewer for the summary and appreciation of our work. We have addressed the reviewers' comments with additional experiments and adjustments to the text of our manuscript as detailed point-by-point below. We have highlighted new **Figures** and revised **Figure** sections for the revised manuscript with red boxes.

Comments

1. The manuscript performed at least limited SAR studies to achieve WEHI-092. It will be beneficial to discuss if/how the SAR is consistent with the determined binding mode.

We have included further discussion around the SAR of WEHI-092 and its analogues included in the manuscript (see Results section). Specifically, the indole nitrogen is methylated in WEHI-333 and exhibits similar potency (IC_{50} 382 nM) to WEHI-092. A far less potent compound, WEHI-680 (IC_{50} 4.3 μ M), with the two methyl groups removed from the structure of WEHI-092, acts as a control with a similar chemical scaffold. WEHI-871 does not possess the indole amide portion and is inactive (IC_{50} >100 μ M). We have included the chemical structure of WEHI-871 into the updated **Figure** showing the chemical synthesis of our compound series, see **Fig S1A**. In the absence of a co-crystal structure with compound, our focus was to explain specificity; molecular docking did not lead to sensible models in our hands; this is likely explained by the shallow surfaces identified in the peptide/HDX mapping. How these compounds bind will need to be assessed in future work.

2. The cellular activity advantage of WEHI-092 over FT709 (e.g., in UO-31 cells) is noted but not mechanistically explained. Differential permeability, metabolic stability, or off-target effects could underlie this result and should be acknowledged. Binding affinity and in vitro potency data actually favor FT709 (K_D = 69 nM vs. 1 μ M; IC_{50} = 124 nM vs. 254 nM), which complicates the claim of comparable or superior efficacy.

This was a great observation that also puzzled us. To address this reviewer's query, we assessed WEHI-092 and FT709 in a Caco-2 permeability study as a surrogate model for cellular uptake and active efflux potential. WEHI-092 exhibited high permeability in both apical-to-

basolateral (A-B) and basolateral-to-apical (B-A) directions, with P_{app} values of 42.6×10^{-6} cm/s and 70.4×10^{-6} cm/s respectively. In contrast, FT709 was far less permeable with measured permeability P_{app} of 3.8×10^{-6} cm/s in the A-B direction and 69.4×10^{-6} cm/s in the B-A direction. This translates into a much higher efflux ratio for FT709 (18) compared to WEHI-092 (1.7). This difference likely explains the differences in cellular activity between the two compounds, especially if efflux transporters are present, such as in UO-31 cells. We have added this data now as new **Fig S4G** and discuss it in the revised manuscript.

Compound	Mass balance (%)		P_{app} (10^{-6} cm/s)		Efflux ratio
	A-B	B-A	A-B	B-A	
WEHI-092	90	91	42.6	70.4	1.7
FT709	98	90	3.8	69.4	18

3. Substrate validation requires additional support. Core substrates such as CEP55 and SMAD3 are proposed based on proteomics and ubiquitinomics, but causal validation is limited. Additional MG132 rescue experiments and/or direct ubiquitination assays are needed for at least 1-2 more targets to substantiate claims of direct USP9X regulation.

We had indeed only included a limited set of validation experiments in the first version of our manuscript, and only showed limited MG132 rescue experiments. We now include additional experiments, analysing protein levels of MKRN2, CHMP2B and DTX3L, three further commonly depleted substrates across cell lines as shown in our global proteomics data in **Fig 5C**. For all proteins assessed, USP9X inhibition with WEHI-092 led to protein depletion in Western Blot experiments which is rescued with proteasomal inhibition (see updated **Fig S6A**). These experiments were performed across two cell lines, MiaPaca2 and SK-MEL-2, as representatives for the cell lines used in our global proteomics experiments.

While SMAD3 is not among the commonly depleted proteins highlighted in **Fig 5C**, we have performed MG132 rescue experiments in UO-31 cells, see **Fig R1**.

Figure R1: Western Blot analysis of SMAD3 protein levels in UO-31 cells upon WEHI-092 treatment (24 h). Proteasomal inhibition using MG132 was performed at 10 μ M for 24 h in presence of 5 μ M QVD apoptosis blockage. The blot shown is a representative of two biological repeats.

We found SMAD3 protein levels to be slightly depleted following USP9X inhibition as expected from our global proteomics data and observed rescue from degradation upon proteasomal inhibition further validating our proteomic data.

4. Determinants of cell line sensitivity are not explored. The proteomic heterogeneity across lines is intriguing but unexplained. Were differences in USP9X expression, substrate levels, or mitotic index correlated with inhibitor sensitivity?

We thank the reviewer for pointing this out. In **Fig S4B** of our manuscript we show an analysis of USP9X expression levels across all cell lines used in our proteomic and ubiquitinomic studies. While MiaPaca2 cells appear to express high levels of USP9X compared to the other cell lines, there is no apparent correlation of USP9X expression levels and sensitivity to WEHI-092. For example, there is no difference in USP9X protein expression between the highly sensitive UO-31 cells and the USP9X inhibitor-resistant HMEC-1 cells. As shown in **Fig S7A**, proteins found depleted following USP9X inhibition in individual cell lines appeared to not be expressed abnormally across all proteins detected in our MS analyses.

In addition, we have addressed the reviewers' comment with an analysis of publicly available gene expression data on the list of 99 genes from **Fig 6A** of the cell lines for which data was available, see new **Fig S7B** in our revised manuscript (and shown on the next page).

Following our analyses summarised in **Fig S7B**, there appears to be a trend indicating that high gene expression levels align with depletion of the gene products following USP9X inhibition, such as in the cell line specific protein depletion in SK-MEL-2 cells and UO-31 cells. In the case of UO-31 cells, this is also represented in the protein abundance from our own data, see **Fig S7A**. This trend, however, appears not be applicable across all cell lines tested, as commonly depleted proteins following USP9X inhibition are not expressed at high levels in general. We conclude that USP9X substrate expression levels do not appear to be a predictor of sensitivity to USP9X inhibition. We hypothesise that the unique combination of depleted substrates following USP9X inhibitor treatment may facilitate the phenotypic effects observed in individual cell lines, highlighting the need for testing of USP9X (and DUB) inhibitors in distinct cellular context to draw conclusion on biological implications. We have also included information of cell line sensitivity and doubling time of these cell lines and the broader NCI-60 cancer cell panel experiments into the new **Fig S4H** which is described in more detail in comment 1 from reviewer 2.

In our study, we have focused on discovering the unique USP9X substrate signatures in individual cell lines; further correlation with unique cell line signatures is beyond the scope of our work, as it requires detailed expert analysis and tools, but could be interesting especially for sensitive cell lines (e.g. UO-31).

Figure S7B revised manuscript: Analysis of expression levels of 99 proteins (from Fig 6A). Data shown side by side with the data from Fig 6A is gene expression (Z-score of log2 fragments per kilobase per million reads [FPKM] + 1) from CellMiner NCI-60 database for the cell lines with data available.

5. Statistical inference in proteomic data is absent from some analyses (e.g., Figure 1). Where key conclusions are drawn from relative abundance changes, significance testing should be applied and visualized to help distinguish biological from technical variation. Additionally, different fold-change thresholds are used without explanation or justification throughout. For example, the LFC threshold of 0.3 represents a small change (a FC of ~1.2) which risks capturing noise or experimental variability, especially where the experiment includes technical replicates only and no biological replicates are presented.

We thank the reviewer for pointing this out. We apologise for the inconsistent labelling of technical / biological replicates and have amended this in our revised manuscript. For all proteomic experiments, we have used biological replicates with n=4-5. We have included error bars for the proteomic datasets in **Figure 1**, i.e. **Figs 1G and S1F**. In addition, we apologise for inconsistent description of FDR / p.adj values for our significance testing and have amended this throughout the revised manuscript (see **Figure** legends).

In the case of the DUB IP-MS approach (**Figs 1G and S1F**), USP9X was the only significant hit in our data analysis (FDR < 0.05). These assays have been routinely performed by us and other labs (e.g. Kessler lab at Oxford, first used by us with specific USP7 inhibitors including FT671 in Turnbull *et al*, 2017) and while they were analysed identically, our description was lacking clarity. Differential expression testing was performed using the DEP package comparing WEHI-092 treated versus untreated lysates that were then exposed to Ub-VS. The DEP package uses limma to perform differential expression analysis and correction for multiple hypothesis testing. In this experiment, only USP9X was statistically significant (i.e. adjusted *p*-value < 0.05). We have amended the **Figure** legend and text to clarify this.

For all other ubiquitinomics and proteomics experiment, filtering criteria for the identification of significant changes are stated in the **Figure** legends.

We have used a log₂ fold-change of -0.3 only in case of the DLD-1 ubiquitinomics / paired global proteomics experiment (**Figs S5C-E**). We applied the lower threshold in line with the report by Steger et al. on profiling cellular responses to the USP7 inhibitor FT671 (Steger *et al*, 2021) which we have used to benchmark our workflow to (**Figs S5A-C**). In this experimental setup, the cells were treated only for up to 6 h. Consequently, global protein depletion was less pronounced when compared to the 24 h timepoints used in all our proteomic analysis where we applied a log₂ fold-change of -0.585. As we detected increased ubiquitination of the corresponding peptides, and high overlap to the data of Steger et al. for the experiments in **Figs S5C-E**, as well as significance threshold of <0.05 (adjusted *p*-value), we are confident of the real depletion of these proteins.

In **Fig 5C**, we have used a threshold of log₂ <-0.5 for annotating the number of cell lines in which the individual protein is depleted for visualisation purposes; however, the proteins listed were pre-filtered to the log₂ <-0.585 criteria in at least one individual cell line as described in the **Figure** legend.

6. Figure S1E refers to a "Control compound" which is not defined in the figure, legend or text. Later experiments use WEHI-680 as a control, but this is not at all defined in the context of Figure S1E. It would be helpful to have WEHI-092 and FT709 on the same blot for comparison as well.

We apologise for not properly labelling a control compound (WEHI-871) from our chemical series of WEHI compounds. We have repeated the experiment, included the compounds suggested by the reviewer and updated the **Figure**, see **Fig S1E**.

7. The manuscript includes WEHI-680 (IC₅₀ 4.3 μ M) as a comparator compound. The rationale for selecting this compound and the value of this comparison is unclear. It is being used in these experiments at concentrations exceeding its IC₅₀, so the phenotypic difference isn't necessarily reflective of strength of USP9X binding (side-by-side target engagement could be revealing here). The manuscript should clearly state the rationale for selecting WEHI-680 as a comparator, and if the authors view the results with this compound as being consequential to the study, augment the discussion by interpretation of the results obtained with this compound

We included WEHI-680 as a less potent control compound as it has the same chemical scaffold as WEHI-092, but without the two methyl groups on the indole (see comment above). This compound is ~10-fold less active, and the reduced effects it exerts in Ub-Rho cleavage assays (**Fig S1B**), binding assessed via SPR (**Fig S1C**), diUb cleavage assays (**Fig S1E**), as well as cell killing in the NCI-60 cancer cell panel (**Fig S4C**) and clonogenic assays (**Fig S4D**) compared with WEHI-092 is fully consistent with this drop in activity. This increased our confidence in the biochemical and *in cellulo* data (see reply to comment 1 for more detail). In addition to our steady-state affinity SPR data shown in **Figs 1C and S1C**, we have included kinetic fitting for our SPR data, see **Table R1** below.

Table R1: SPR kinetic fitting data for WEHI-compound series and FT709. Data shown is the mean of three independent biological repeats with standard deviation.

Compound	Kinetic fitting SPR			
	k_{on} (1/ μ Ms)	k_{off} (1/s)	K_D (μ M)	R_{max} (RU)
FT709	1.072 \pm 0.375	0.062 \pm 0.019	0.058 \pm 0.004	29.1
WEHI-092	0.156 \pm 0.019	0.172 \pm 0.006	1.111 \pm 0.133	28.6
WEHI-333	0.09 \pm 0.009	0.119 \pm 0.018	1.323 \pm 0.284	28.8
WEHI-680	0.175 \pm 0.046	0.607 \pm 0.078	3.547 \pm 0.423	26.7
WEHI-871	n.d.	n.d.	n.d.	n.d.

We find that FT709 has a fast on-rate and slow off-rate compared to the WEHI compound series, explaining its slightly higher potency *in vitro*. When comparing WEHI-680 to WEHI-092, the weaker compound activity of WEHI-680 can be explained with a much faster off-rate.

8. The Discussion includes speculative questions (e.g., about USP9X's role as a rheostat or spatial organizer) but does not synthesize a model or clearly articulate the new biological insights. These should be revised for clarity on advances and enablement made by the study. Similarly, the suggestion that this study "consolidates the literature" is vague--please specify what previous discrepancies are addressed and what overarching model emerges from this work.

The pleiotropic effects for USP9X that we uncovered through the use of a specific inhibitor across many cell lines, are indeed difficult to describe, which is likely the reason why our Discussion appeared vague. We have tried to clarify the wording in the Discussion. We have also removed the sentences where we speculate how USP9X may act – these were phrased as open questions as we simply don't know, and we are uncomfortable (and are clearly not encouraged) to speculate.

9. The conclusion that profiling six cell lines resolves the variable reports on USP9X function may be overstated. While the authors appropriately highlight heterogeneity, this limited sample cannot fully explain prior inconsistencies across diverse models. This claim should be softened or caveated.

We agree that we do not reconcile the differences in the hundreds of papers published on USP9X. However we do believe that our paper and analysis, for the first time, unveils the extreme heterogeneity in the literature potentially not just explaining pathophysiologically distinct observations (e.g. tumour suppressor vs oncogene – cell line / cancer specific effects) but perhaps also highlighting that identical substrates cannot be expected for a DUB such as USP9X across many cell lines; this often causes a main caveat (perceived irreproducibility) in the literature. Indeed, we are aware from our work and collaborating labs and companies, that many reported substrates could not be reproduced, which causes frustration, adds to the irreproducibility crisis, but essentially remains unpublished negative data. We have considered the reviewers' point and have softened the statements to be less sweeping in our conclusions.

Referee #2:

In the presented manuscript did Schenk et al. present the development and validation of a novel and highly selective DUB inhibitor, WEHI-, which selectively targets USP9X. The group of David Komander used a plethora of tools, including synthesis, DUB explorer assays, activity-based probe analysis, tissue culture, mass spectrometry and immunofluorescent imaging to develop this inhibitor and identify the physiological consequence of USP9X interference.

The enzyme family of deubiquitylases is an exciting and pharmacological underrepresented group of drug targets and every effort to develop new and highly selective drugs is needed, since the clinical data and pre-clinical models suggests DUBs to be an excellent cancer vulnerability. The possibility to target a novel member of this enzyme family in a well defined and stratified patient cohort could be helpful in the future. To this extend is the reported inhibitor a great tool for continued drug development.

The manuscript is well written and the experiments very timely and well carried out. The reviewer would like to suggest deepening the understanding of the biological cellular processes upon exposure to USP9X, especially given the wide spectrum of cellular phenotypes observed.

We thank the reviewer for the positive feedback on our work and agree with the statement that selective DUB inhibitors are underrepresented in the DUB literature highlighting the importance of our study in the broader DUB inhibitor context. We have addressed the reviewers' comments below with further experiments, *in silico* analyses and adjustments in the text as detailed point-by-point below. We have highlighted new Figures and revised Figure sections for the revised manuscript in red.

In detail:

1. Figure 3: The authors used a panel of human cell lines and observed various degrees of cellular responses, ranging from growth inhibition to cell death, while others showed a profound increase in proliferation, such as HL-60T or COLO-205, . It is intriguing that among the cell lines tested SR and UO-31 showed a striking response, along with the AML derived cell line THP-1, which also presented a robust response to the USP9X inhibitor. Could the authors please elaborate on the genetic alterations present in these cells?

To address the reviewers' comment we have compiled an oncoprint of commonly mutated genes across all cell lines used in our work from the NCI-60 cancer cell panel. The oncoprint includes information on the sex of the cell lines, doubling time as well as sensitivity to USP9X inhibition as assessed in the NCI-60 cancer cell panel (**Fig 3B**). The oncoprint is shown in our revised manuscript in the new supplementary **Fig S4H** (also reproduced on next page).

A detailed comparison between cell lines that showed enhanced growth in the NCI-60 cancer cell panel following USP9X inhibition (HL-60 and COLO 205) revealed limited similarities between these lines. They are of different sex but both have relatively fast doubling times. However, also USP9X inhibitor sensitive cancer cell lines such as SR have fast doubling times. When comparing sensitive cell lines SR, SK-MEL-2 and UO-31, a similarly limited overlap of the mutational status was found with the exception that both, SR and SK-MEL-2 harbour a deletion of *CDKN2A*. Overall, there appears to be limited correlation with observed phenotypes following USP9X inhibitor treatment. One hypothesis could be that the sensitivity of cell lines to WEHI-092 is dependent on as yet undiscovered dependencies / synthetic lethalties with cellular signalling pathways. However, this is not obvious from analysing the cBioPortal annotations using a defined COSMIC gene list (372 Hallmark of Cancer Genes dataset); there was no clear regulation of the same gene in the two highly sensitive cell lines compared to less sensitive cells.

Is there a gender-bias in the cellular responses observed?

There does not seem to be a gender bias. Sensitive cell lines (positive and loss of growth) are of different sex (3x male, 2x female) origin (see **Figure**).

2. Furthermore, based on public available data, does the expression of USP9X show a wide range in regard to correlation of patient survival. Having the cellular responses to USP9X inhibition available, do the observed cellular responses match the tumour behaviour, meaning, do tumours where USP9X levels would be indicative of poor patient survival show an anti-proliferative phenotype in cellulo?

We have analysed gene expression data for USP9X across different cancer types from the Gene Expression Profiling Interactive Analysis (GEPIA) server comparing tumour and normal tissues, see **Fig R2** (next page).

As shown in **Fig R2**, certain cancer types such as breast cancer appear to express higher levels of *USP9X* in tumour tissue when compared to normal tissue. However, in other cancers such as kidney carcinoma, *USP9X* appears to be expressed at lower levels in the tumour, highlighting a diverse expression profile as indicated by the reviewer. While we only identified a small subset of cell lines in short-term cell killing assays to be sensitive to *USP9X* inhibition (**Fig 3B**), a broader range of cancer cell lines was found to be sensitive in clonogenic assays (**Figs 3E and F**) including breast cancer cells MDA-MB-231 and MDA-MB-468 as well as the melanoma line SK-MEL-2 (which was also sensitive in short-term killing assays). We have analysed GEPIA overall survival data on three different cancer types, breast, melanoma and renal cancer in relation to high / low levels of *USP9X* expression, see **Fig R3**.

Figure R2: *USP9X* is overexpressed in tumour tissues when compared to normal tissues. Gene expression data and plot for *USP9X* expression levels taken from GEPIA server at <http://gepia.cancer-pku.cn/index.html> (Tang et al, 2017). The height of the bars represents the median expression of a certain tumour type or paired normal tissue. Abbreviations: ACC – adrenocortical carcinoma, BLCA – bladder urothelial carcinoma, BRCA – breast invasive carcinoma, CESC – cervical squamous cell carcinoma and endocervical adenocarcinoma, CHOL – cholangiocarcinoma, COAD – colon adenocarcinoma, DLBC – lymphoid neoplasm diffuse large B-cell lymphoma, ESCA – oesophageal carcinoma, GBM – glioblastoma multiforme, HNSC – head and neck squamous cell carcinoma, KICH – kidney chromophobe, KIRC – kidney renal clear cell carcinoma, KIRP – kidney renal papillary cell carcinoma, LAML – acute myeloid leukemia, LGG – brain lower grade glioma, LIHC – liver hepatocellular carcinoma, LUAD – lung adenocarcinoma, LUSC – lung squamous cell carcinoma, MESO – mesothelioma, OV – ovarian serous cystadenocarcinoma, PAAD – pancreatic adenocarcinoma, PCPG – pheochromocytoma and paraganglioma, PRAD – prostate adenocarcinoma, READ – rectum adenocarcinoma, SARC – sarcoma, SKCM – skin cutaneous melanoma, STAD – stomach adenocarcinoma, TGCT – testicular germ cell tumours, THCA – thyroid carcinoma, THYM – thymoma, UCEC – uterine corpus endometrial carcinoma, UCS – uterine carcinosarcoma, UVM – uveal melanoma.

Figure R3: Comparison of overall survival for patients with high versus low *USP9X* expression, using the median value as the cutoff. Abbreviations: TPM – transcripts per million reads.

As shown in **Fig R3**, there is a trend towards decreased overall survival in breast cancer patients with high *USP9X* expression, and a similar trend towards decreased overall survival in

renal cancer patients with low *USP9X* expression. There is no apparent correlation between sensitivity to *USP9X* inhibition with overall patient survival in distinct cancer types. We do note that we identify distinct subsets of *USP9X* substrates in individual cell lines that most likely mediate inhibitor sensitivity. A patient stratification into cancer subtypes with distinct expression signatures of *USP9X* substrates might be informative but lies beyond the scope of our work.

3. Figure 4: Next do the authors go into great length to compare the proteome of *USP7* or *USP9X* inhibited cell lines DLD-1 and MDA-MB-231. While this is exciting to observe and the resolution points towards a rapid response in deregulated protein abundance, especially in the upregulated fraction, while the downregulated fraction, at least in MDA-MB-231, is more persistent. The reviewer is wondering if the same was observed for DLD-1 and do the authors think that the pattern would be consistent in other cell lines as well? How does this pattern look like in cell lines that showed a low or no proliferative phenotype? Would this be altered as well?

We apologise if the visualisation for **Figure 4** was not clear. We have included a label for “Ubiquitination sites” and “Proteins” into **Figs 4B and C** where we compare *USP7* and *USP9X* inhibition in DLD-1 cells. This comparison was only performed in DLD-1 cells, and we hypothesise that distinct *USP7* and *USP9X* substrate spectra identified are present across other cell lines as well. Indeed, we here used the *USP7* inhibition solely to illustrate (a) compound specificity and lack of overlapping substrate for two highly abundant human DUBs, and (b) that similar trends, i.e. increases in ubiquitination of substrates and corresponding loss of a substrate of ubiquitinated proteins, can be observed. For *USP7*, this was the basis for the landmark paper by Steger et al (2021), which guided this part of the study.

In MDA-MB-231 cells we show time-resolved substrate profiling across 30 min, 360 min and 24 h *USP9X* inhibitor treatment (**Figs 4D and E**), with a rapid increase in ubiquitination events following *USP9X* inhibition. The same behaviour was also seen in DLD-1 cells (**Fig S5D**), and also for *USP7* inhibition in DLD-1 cells (**Fig S5E**) (consistent with Steger et al). In that regard, the same pattern of early increase in ubiquitination events following *USP9X* inhibition and a later decrease of overall protein abundance was observed in both cellular contexts. However, in both these cell lines, DLD-1 and MDA-MB-231, *USP9X* inhibition was ineffective to kill cells (**Fig 3C**) but showed a reduction in colony formation capabilities (**Figs 3E and F**). So, the reviewer is right that the chosen cell lines for proteomic analysis, were perhaps not fully reflective of the broader phenotypes observed.

Following the reviewers’ comment (and also comment 2 from reviewer 3), we have expanded our ubiquitinomic studies to two additional cell lines, SK-MEL-2 and UO-31, both of which showed proliferation defects in our phenotypic assays. This new data is now incorporated in **Figs 4F and G, and EV Fig 1A and B**. In the new data we show an overlap of 38 ubiquitinated proteins following 30 min WEHI-092 treatment across four distinct cell lines, further strengthening our data resource. Overlap of high confidence substrates (increased ubiquitination and decreased protein abundance) revealed eight high confidence *USP9X* substrates in the cell lines for which time-resolved profiling was performed.

Hence, while we see the same as described in this manuscript, i.e. a highly individual, cell-type specific response to *USP9X* inhibition, the pattern of increased ubiquitination at early time points, and depletion of the ubiquitinated proteins at later time points, is preserved.

In the first version of the manuscript, we compared DLD-1 and MDA-MB-231 cells, however, we only had short time point data for DLD-1 cells (as this was matched to literature protocols,

Steger et al). In this updated manuscript we now compare 4 cell lines at 30 min, and 3 cell lines at 24 h inhibition.

4. Since the described novel inhibitor WEHI-092 is extensively used in this manuscript it is not obvious for this reviewer if this inhibitor was immobilized on beads and used for a global IP-MS approach to investigate its putative specificity on a more wider range of proteins?

This was an interesting idea that we would consider if the inhibitor would show severe off-target activity (see next point). Since we have tight SAR of the compound, it is not immediately clear where and how to attach an affinity tag; again, a structure is required for this question. We were hence unable to perform this experiment at this point.

5. To validate the on target effects of the proposed small molecule inhibitor could the authors please use e.g. a knock down or acute knock out strategy, such as si/shRNA or the Auxin induced degradation model, followed by MS analysis, for cross-validation in 1 or two exemplary cell lines? This would greatly assist in the validation of the biological effects observed by cells exposed to WEHI-092.

We thank the reviewer for this useful suggestion. We have performed siRNA knockdown (KD) experiments in MDA-MB-231 cells with subsequent global proteomics analyses. We have included this data in our revised manuscript, see new **Figs S6C-E**.

We have detected 195 proteins depleted following siRNA-mediated gene silencing of USP9X. We found many proteins from **Fig 5** such as CEP55, MKRN2, EDF1, CHMP2B, TCF25, and others to be depleted in USP9X KD conditions, creating highly overlapping datasets between genetic depletion and small molecule inhibition in this cellular context. In addition, previously reported substrates such as PCM1 and CEP131 were also found depleted following USP9X KD (see also **Fig 5A** for data in MDA-MB-231 cells comparing FT709 and WEHI-092 effects). We note the different biological setups of the two approaches (KD and small molecule inhibition) that might lead to distinct protein signatures and could indicate a scaffolding role of USP9X for some of the 195 proteins depleted in the KD condition based on the higher number of depleted proteins compared to small molecule inhibition.

In addition, we included WEHI-092 treatment in the MDA-MB-231 USP9X KD cells in our experiments (**Fig S6E**) to study potential off-target effects of the compound.

47 proteins appeared to still be downregulated in this condition. Most of the proteins from **Fig 5C** are absent in this setup indicating USP9X-specific regulation of these proteins. However, a subset of proteins from **Fig 5C** (TYMS, TOR4A, ACBD6, PDCL3, TGFB111, DTX3L) appear to be further depleted in the USP9X KD condition upon WEHI-092 treatment. This could either be due to residual USP9X activity present in this setting as shown in the bar plot highlighting raw intensities of USP9X in our proteomic analysis, or reflect true off-target effects. Also, proteins previously not detected in our analyses (CYP1B1, CUL7, KEAP1) could indicate off-target effects of our inhibitor; these however have not featured as significant hits when USP9X was present. A third possibility is that compound treatment causes further stress on cells already experiencing USP9X loss (via an siRNA), and that further protein depletions are compounding collateral, i.e. indirect effects.

In sum, we believe that validation of direct targets of USP9X via distinct MS analysis, Western blotting, inhibitors, and depletions, provides a highly detailed and sophisticated canvas, in which compound off-target effects are likely negligible.

6. Figure 5 and S6: Could the authors please confirm by TUBE, DSK or other means of ubiquitin pulldown that the proposed targets of USP9X, such as CEP55 or EDF1, are indeed polyubiquitylated upon loss of USP9X?

We thank the reviewer for this great suggestion to focus on the ubiquitination events in more detail. In fact, our datasets already included this information, yet we had not explicitly analysed this connection.

In our original manuscript we stated that several of the top ten downregulated proteins across all cell lines had corresponding ubiquitinated peptides in the ubiquitinomics studies including CEP55, EDF1, MKRN2, HECTD1. This can be seen from **Figs 4E and S5D**, where at 30 min of WEHI-092 treatment all proteins shown in the heatmaps have strongly increased ubiquitination in our high-resolution (peptide-level) ubiquitinomics analyses. We have now studied the commonly depleted, and Western Blot validated targets CEP55, CHMP2B, DTX3L, and EDF1 in the anti-GG datasets in MDA-MB-231 cells in more detail. Each protein shows increased ubiquitination on one or multiple sites in the ubiquitinomics dataset. We have extracted this in bar graphs resolved on the ubiquitinated peptide-level, now shown in new **Fig S6F**.

We have cross-validated the ubiquitination sites identified using the PhosphoSitePlus repository (<https://www.phosphosite.org>) and found overlap with our data with reported sites K22, K48 in CEP55; K81, K90, K107 in CHMP2B; and K521 in DTX3L as previously reported ubiquitination sites in these proteins. Indeed, identifying these known ubiquitination sites as being direct USP9X-regulated sites is a nice additional feature of our study and a useful resource.

We consider this data superior to what the reviewer has asked for, namely a TUBE-enrichment Western Blot. While such biochemical experiments work nicely in some cases, many antibodies do not recognise ubiquitinated proteins.

In addition, we have performed further MG132 rescue experiments in MiaPaca2 and SK-MEL-2 cells following WEHI-092 treatment that are now shown in the revised **Fig S6A** (see reviewer 1, comment 3 for more detail). These experiments strengthen our conclusion of direct USP9X regulation of CEP55, MKRN2, CHMP2B, and DTX3L.

Do the authors suggest that the mentioned proteins are direct targets and interaction partners of USP9X? Do these factors, namely EDF1, CEP55, EDF1 or MKRN2, co-immunoprecipitate with USP9X?

We would like to point the referee also to previous studies on CEP55 and MKRN2, which have shown co-immunoprecipitation data for these direct substrates of USP9X (Wang *et al*, 2017; Clancy *et al*, 2021). We did not repeat these literature experiments.

7. Figure 7: Could the authors please share there thoughts if this mode of action is causative to the strong phenotypes observed in some cell lines, like THP1 or UO-31? Since the statement is very brought, that the inhibition of USP9X arrests cancer cells in metaphase, most cell lines in Figure 3 and S4, showed mild responses to high concentrations of the inhibitor. Could the authors therefore please either elaborate more on the commonality or generalism of this observed arrest, or rephrase the statement, reflecting that a selective number of cell lines halted proliferation?

To address the reviewers' comment in detail, we have extended our imaging studies in the revised manuscript: We have profiled the sensitive cell line UO-31 in lattice light sheet imaging experiments as well as the partially sensitive SK-MEL-2 cells to follow individual cells upon treatment with WEHI-092 similar to the experiments shown in Fig 7. In SK-MEL-2 cells, we detected the mitotic arrest similar to MDA-MB-231 and MiaPaca2 cells. The quantification of the SK-MEL-2 data is shown in the new Fig S8E. In the UO-31 cells we found a strong cell death phenotype (consistent with Fig 3) that appeared independent of the mitotic arrest seen in other cell types such as MDA-MB-231 and MiaPaca2 cells. The quantification of the imaging data is shown in new Fig S8F. We have also compiled a new **Supplementary Movie 5** showing this phenotype in UO-31.

From these experiments, we conclude that the arrested mitosis is not causative of the strong death phenotype seen in UO-31 cells. The unique sensitivity of UO-31 cells to WEHI-092 is dependent on yet undiscovered dependencies / synthetic lethalties with cellular signalling pathways (see also out reply to comment 1 form the same reviewer).

We have also softened our conclusions on the generalisability of the mitotic arrest in our revised manuscript.

Minor points:

Could the authors please elude on the specificity of the compounds towards USP9Y?

The USP9Y catalytic domain shares >92% sequence identity with USP9X and the overall 3D-architecture of the catalytic domains are very similar, see **Fig R4**.

Figure R4: USP9X (green) and USP9Y (blue) catalytic domains superimposed from an AlphaFold3 prediction highlighting very high overall structural similarity.

The WEHI-092 binding site we have identified in USP9X (aa 1757-1762 YVKGD~~L~~), differs in the USP9Y sequence only in one residue, with Ile in position 1760 instead of Val (position 1758) in the USP9X sequence (the respective motif in USP9Y is aa 1759-1764 YIKGD~~L~~). To answer the reviewers' question, we have modelled a USP9Y structure using AlphaFold3 with ubiquitin bound (similar to **Fig 2E** for USP9X), as shown below in **Fig R5**.

Figure R5: Comparison of the detailed view on the WEHI-092 binding site in USP9X (**Fig 2E**, left) with the corresponding motif in USP9Y (residues YIKGD~~L~~; right), with blue nitrogen and red oxygen atoms, with transparent surface of modelled ubiquitin (pink carbon atoms).

The only residue within the WEHI-092 binding site that is different from USP9X, Ile in position 1760, is pointing away from the ubiquitin binding interface in our AlphaFold3 model (**Fig R5**).

Other key residues, such as Tyr1759 and Asp1763 are positioned very similar to the AlphaFold model for USP9X (see **Fig 2E**) and appear within hydrogen binding distance with ubiquitin. Disturbance of these residues in our USP9X/USP24 chimera abolished WEHI-092 activity. We therefore conclude from our AlphaFold3 model that WEHI-092 would likely inhibit USP9Y due to the sequence similarity in the binding motif and the apparent unimportance of Ile1760 for mediating ubiquitin binding.

Do the cell lines used express both variants of USP9? Does USP9X undergo X-chromosome escape?

We have analysed *USP9Y* gene expression levels across the NCI-60 cancer cell lines similar to **Fig S7B**. *USP9Y* expression in comparison to *USP9X* expression is shown below, see **Fig R6**.

Figure R6: NCI-60 cancer cell panel comparison of USP9X and USP9Y gene expression levels, sorted by cancer type. Data shown in the heatmap is the fragments per kilobase per million reads (FPKM) from CellMiner NCI-60 database. WEHI-092 response refers to NCI-60 panel testing (**Figs 3B and S7B**).

Overall, *USP9Y* appears to be expressed at much lower levels when compared to *USP9X*. *USP9Y* is absent from female tissue and only six cell lines express *USP9Y* at higher levels, none of which have shown growth phenotypes in our *USP9X* inhibitor assays. According to literature data, *USP9X* escapes X-chromosome inactivation (*Jones et al, 1996*).

Does the origin of sex of the cancer cell lines used impact on the susceptibility towards the inhibitor?

This comment relates to comment 1 of the same reviewer and has been addressed with our oncoprint, see new **Fig S4H**. We did not observe a correlation between sex of the cancer cell lines and susceptibility to USP9X inhibition.

Some passages of the manuscript have a connotation comparable to a review, such as:
>> Some of the discrepancies in the literature arise from an undue focus on single-protein relationships (USP9X regulates target X), which was sometimes exaggerated by inappropriate tools. More than 30 papers have assigned the effects of Degrasyn/WP1130 and its derivatives G9/EOAI3402143 (Bartholomeusz et al, 2007a, 2007b) to USP9X inhibition, despite data that WP1130 is a non-specific DUB inhibitor (Ritorto et al, 2014; Kapuria et al, 2010; Peterson et al, 2015).<< These parts could be moved to the discussion part or rephrased.

These sentences are in the introduction of our manuscript, and provide important background context for the study and rationale for generating new inhibitors, which as the reviewer remarked in his intro, is an important advance required to study DUBs better, with more advanced tools. We felt that we need to point out the previous misuse of non-specific inhibitors, which has been a bane in DUB research.

This sentence is a short summary of the larger section on USP9X in our recent review (Dewson *et al*, 2023) but the points are not forcefully made in other more biological reviews which needs awareness by the field.

Referee #3:

The authors previously reported the identification of a USP9X inhibitor (TF-709). In this study, they present a new-generation USP9X inhibitor, WEHI-092. Its specificity is supported by in vitro deubiquitylation assays, and structural analyses provide insights into its mechanism of action. Functional assays show that WEHI-092 selectively inhibits growth in certain cell lines. Proteomic analyses of global ubiquitylation and protein expression changes further identify potential USP9X-regulated substrates. While the study offers interesting findings, several critical issues need to be addressed. Note: This reviewer is not an expert in structural biology and has not evaluated that portion of the manuscript.

We thank the reviewer for critically reviewing our work and have addressed the reviewers' comments point-by-point as detailed below.

Major comments:

1. The authors should more clearly demonstrate the reproducibility of their proteomic analyses of ubiquitylation and protein abundance. The number of biological replicates should be explicitly stated. More importantly, quantitative reproducibility between replicates must be shown.

We thank the reviewer for pointing this out. We apologise for the inconsistent labelling of technical / biological replicates and have amended this in our revised manuscript in all **Figure legends**. For all proteomic experiments, we have used biological replicates with n=5.

In the results section, the authors note that "... the number of ubiquitinated proteins exceeded the number of subsequently degraded proteins, by at least ~6-fold. This observation may indicate either non-degradative (e.g. signalling) ubiquitination events, or ubiquitination events that are not yet functional to induce proteasomal degradation." However, an alternative explanation is that the false discovery rate is higher for ubiquitylated peptides. Site-level quantification relies on single peptides, while protein-level measurements aggregate multiple peptides, making the latter more robust to noise. This could account for the limited overlap observed and should be addressed explicitly.

We agree with the reviewer that K-GG ubiquitinomics is inherently a more variable assay compared to conventional quantitative proteomics. As the reviewer points out, a major factor driving this variation is the use of only one peptide (generally observed in one charge state but sometimes two) to monitor a remnant ubiquitination site compared to using multiple peptides to derive a consolidated protein measurement. This is reflected in the distribution of CVs (coefficient of variation) between both types of experiments, with K-GG ubiquitinomics generally exhibiting a median CV of 25%, whereas global proteomics exhibiting a median CV < 10%.

Another factor is the incredibly low occupancy of ubiquitination as wonderfully described by Chuna Choudhary's group (Prus *et al*, 2024). Ubiquitination has a median occupancy 0.0081% compared to 28% for phosphorylation and a median half-life of 12 minutes. Even with an optimised enrichment for K-GG peptides and starting with 2-3 mg of trypsin-digested peptides, many detected K-GG peptides are likely at the limit of detection and a greater proportion remain undetectable.

Considering these limitations, it was nonetheless encouraging for us to reproduce the USP7 inhibition experiment described by Steger *et al*. and observe over 50% overlap in specific K-GG peptides that were significantly increased following USP7 inhibition. So, while K-GG

ubiquitinomics is inherently 'noisier' than global proteomics there is robust reproducibility between different labs repeating the same experiments.

We would argue there are three main drivers of the 6-fold difference in ubiquitinated proteins compared to total protein changes.

1. More immediate and binary regulation of ubiquitination. To alter the ubiquitination status requires a one-step deubiquitination event. Whereas proteasomal degradation of a protein requires a more complex number of steps and regulatory interventions.
2. The low occupancy of ubiquitination (0.0081% median occupancy) means there is inherently a greater propensity for large fold changes from only moderate changes to E3 ligase/deubiquitinase activity.
3. K-GG peptide detection close to the limit of detection. Because of the low occupancy/abundance of ubiquitination, K-GG peptide precursors detected on a mass spectrometer are more susceptible to noise or small fluctuations in intensity.

To acknowledge the important point the reviewer has made and to better encapsulate the limitations of the approach but place the work in a wider context we have amended the main text to highlight this limitation and possible mechanisms underpinning the difference in amount of K-GG peptides and proteins changing under the identical conditions.

Also, what is the difference between "non-degradative (e.g. signalling) ubiquitination events" and "ubiquitination events that are not yet functional to induce proteasomal degradation?"

The ubiquitination requirements for efficient proteasomal degradation, has evolved significantly over the last decade, and degradability includes a multitude of steps, from partial unfolding of a substrate (e.g. mediated by p97, or simply by an end-of-life damage to a protein). From the earlier view that 'a K48-tetra-Ub modification' suffices for degradation, the field has evolved to now realise that multisite ubiquitination serves as a timer to enable efficient delivery of substrates into the proteasome. Indeed, we have previously proposed a ubiquitination threshold, that determines proteasome delivery and degradation.

We used the phrase "ubiquitination events that are not yet functional to induce proteasomal degradation" to try to capture this vast and evolving literature. This acknowledges that some proteins assessed at an arbitrary 30 min USP9X inhibition timepoint, may simply not have received sufficient Ub to trigger degradation ('en route').

We do appreciate that this is likely too simplified (as is the model) and that we did not cite the previous idea sufficiently.

Figure 5 A 'ubiquitin threshold' model for proteasomal degradation. Substrate ubiquitination can result in two general outcomes, cellular signaling or proteasomal degradation. Recent evidence supports a model in which multiple short chains (e.g., diubiquitins) or branched ubiquitin are better degradation signals as compared to a single Lys48-linked tetraubiquitin. These findings also suggest that non-degradative ubiquitin signals could be modified into degradative signals through addition of short and/or branched ubiquitin chains to substrates.

Fig reproduced Swatek & Komander, 2016.

2. The authors report that USP9X inhibition most strongly impairs growth in MOLT4, SK-MEL-2, SR, and UO-31 cells. Yet, ubiquitylation profiling was performed in DLD1 and MDA-MB-231 cells, which are not among the most sensitive lines. The rationale is unclear. In any case, it would be more informative to perform these analyses in at least two cell lines that exhibit strong growth inhibition in response to USP9X inhibition, to better connect ubiquitylation changes with phenotypic effects.

We agree with the reviewer that ubiquitinomic experiments in WEHI-092-sensitive cell lines would reveal interesting biology of USP9X inhibition. We have therefore expanded our ubiquitinomic studies to two additional cell lines, in example SK-MEL-2 and UO-31 which both showed phenotypic effects following USP9X inhibition (see Fig 3). In these cell lines, we have performed time-resolved profiling of ubiquitination events and global changes in protein abundance following treatment with WEHI-092, similar to the experiment in Figs 5C and D in MDA-MB-231 cells, generating a comprehensive resource across distinct cellular context. This new data is now incorporated in Figs 4F and G, and EV Fig 1A and B (see also reviewer 2, point 3).

In the new data we show an overlap of 38 ubiquitinated proteins following 30 min WEHI-092 treatment across four distinct cell lines. The vast majority of ubiquitinated proteins however, is detected only in specific cellular contexts.

F

Moreover, overlap of high confidence substrates (increased ubiquitination and decreased protein abundance) revealed eight high confidence USP9X substrates in the cell lines for which time-resolved profiling was performed.

3. The authors state: "...high-confidence USP9X substrates found in MDA-MB-231 cells, 58 were detected in DLD1 cells at the peptide level, but were seemingly not significantly altered by USP9X inhibitors during the experiment." This could indicate that the same sites are regulated by different DUBs in different contexts or, alternatively, that measurement noise impact quantification. To clarify this, the authors should report the reproducibility of regulated sites across biological replicates in both cell lines. The number of sites regulated in individual experiments, and their overlap should be shown. This would help distinguish biological variability from quantification noise.

We thank the reviewer for pointing this out. We had initially included the K-GG profiling in DLD-1 cells compared to MDA-MB-231 comparing high confidence substrates with the caveat that

DLD-1 cells were only profiled up to 6 h (for benchmarking the USP7 inhibition data to Steger et al as shown in **Fig 4B and C**). Following the reviewers' suggestion, we have analysed the overlap of K-GG peptides in both cell lines at the 30 min timepoint, see **Fig R7** below. We found good overlap between the two cell lines highlighting robustness of the method.

Figure R7: Venn diagram showing the overlap of K-GG peptides detected in MDA-MB-231 and DLD-1 at 30 min of treatment with WEHI-092.

Furthermore, we have undergone significant effort to reproduce the data in Steger et al and show the robustness of our K-GG experimental pipeline with the data shown in **Figs S5A-C**.

This is actually very nice as it implies that ubiquitination sites are perhaps more reproducible than previously thought, across cell lines, and across studies. As discussed above, a major drawback in ubiquitination site detection is the problem of a fast dynamic behaviour of ubiquitination. It is a possibility, that this dynamic originates from DUB action. In our experiments and in Steger et al, the corresponding DUB is inactivated, slowing ubiquitination site dynamics, enabling consistent detection.

With regards to reproducibility, we now include **Fig S6F**, describing individual ubiquitinated peptides from this analysis. This **Figure** shows how individual sites on regulated proteins are increased in their ubiquitination upon USP9X inhibition, directly linking Ub sites to a DUB. In this **Figure** we also show individual data points from the biological replicates, highlighting reproducibility across our experiments.

In our revised manuscript, we have included two additional cell lines that were analysed in K-GG experiments (see comment above). Therefore, the comparison of MDA-MB-231 and DLD-1 on the high confidence protein level with the suboptimal comparison of different late timepoints (6 h and 24 h) has been replaced now with higher quality data, see new **Fig 4F**. We have also amended the section in the text referring to this.

Following our expansion of K-GG studies to two additional cell lines (see above), we have replaced the Figures that showed overlap in ubiquitinated peptides and high confidence substrates between MDA-MB-231 and DLD-1 only (former **Figs 4F and G**) with the overlap between all of the cell lines tested (30 min ubiquitination: MDA-MB-231, DLD-1, SK-MEL-2, UO-31 and full time-resolved profiling for MDA-MB-231, SK-MEL-2 and UO-31).

Minor comment: The authors mention the use of an in-house di-Gly antibody but provide no reference or information on the antibody's source or properties (e.g., monoclonal/polyclonal, host species). This information should be included to ensure experimental reproducibility.

We apologise for the incorrect labelling of the K-GG antibody we used for our experiments. The antibody used in our experiments is the K-GG antibody licensed to Cell Signaling Technology (PTMscan® Pilot Ubiquitin Remnant Motif K-GG kit #14482). We have amended this section in the Methods.

Bibliography Schenk et al. replies to reviewer comments

Clancy A, Heride C, Pinto-Fernández A, Elcocks H, Kallinos A, Kayser-Bricker KJ, Wang W, Smith V, Davis S, Fessler S, *et al* (2021) The deubiquitylase USP9X controls ribosomal stalling. *J Cell Biology* 220: e202004211

Dewson G, Eichhorn PJA & Komander D (2023) Deubiquitinases in cancer. *Nat Rev Cancer*: 1–21

Jones MH, Furlong RA, Burkin H, Chalmers IJ, Brown GM, Khwaja O & Affara NA (1996) The Drosophila Developmental Gene Fat Facets Has a Human Homologue in Xp11.4 Which Escapes X-inactivation and Has Related Sequences on Yq11.2. *Hum Mol Genet* 5: 1695–1701

Prus G, Satpathy S, Weinert BT, Narita T & Choudhary C (2024) Global, site-resolved analysis of ubiquitylation occupancy and turnover rate reveals systems properties. *Cell* 187: 2875–2892.e21

Steger M, Demichev V, Backman M, Ohmayer U, Ihmor P, Müller S, Ralser M & Daub H (2021) Time-resolved in vivo ubiquitinome profiling by DIA-MS reveals USP7 targets on a proteome-wide scale. *Nat Commun* 12: 5399

Swatek KN & Komander D (2016) Ubiquitin modifications. *Cell Res* 26: 399–422

Tang Z, Li C, Kang B, Gao G, Li C & Zhang Z (2017) GEPIA: a web server for cancer and normal gene expression profiling and interactive analyses. *Nucleic Acids Res* 45: W98–W102

Turnbull AP, Ioannidis S, Krajewski WW, Pinto-Fernandez A, Heride C, Martin ACL, Tonkin LM, Townsend EC, Buker SM, Lancia DR, *et al* (2017) Molecular basis of USP7 inhibition by selective small-molecule inhibitors. *Nature* 550: 481–486

Wang Q, Tang Y, Xu Y, Xu S, Jiang Y, Dong Q, Zhou Y & Ge W (2017) The X-linked deubiquitinase USP9X is an integral component of centrosome. *J Biol Chem* 292: 12874–12884

Prof. David Komander
The Walter and Eliza Hall Institute of Medical Research
Ubiquitin signalling
1G Royal Parade
Parkville
Melbourne, Victoria 3052
Australia

19th Jan 2026

Re: EMBOJ-2025-121586R
Global analysis of cancer cell responses to USP9X inhibition

Dear David,

Thank you again for submitting your revised manuscript to The EMBO Journal. Two of the original referees have now assessed it once more, and I am happy to say they both were fully satisfied with the revisions. After incorporation of the following remaining editorial issues, we should therefore be able to proceed with formal acceptance of the study:

- Please adjust the order and the headers of the different manuscript sections: Title page with complete author information, Abstract, Keywords, Introduction, Results, Discussion, Methods, Data Availability, Acknowledgements, Disclosure and Competing Interests Statement, References, Main Figure Legends, Tables, Expanded Figure Legends.
- On the abstract page of the manuscript, please include 4-5 general keyword terms to enhance searchability.
- As we are switching from a free-text author contribution statement towards a more formal statement based on Contributor Role Taxonomy (CRediT) terms, please remove the present Author Contribution section and instead specify each author's contribution(s) directly in the Author Information page of our submission system during upload of the final manuscript. See <https://casrai.org/credit/> for more information.
- Please carefully go through the reference list and make sure that all references have their complete citation information - including year, journal name & volume, and page/locator numbers - such information is currently missing for several of them. Also, please adjust the format for citation of preprints as specified in our author guidelines:
The citation in the text should be: "(preprint: [author name1] et al, [year])"
The citation in the reference list: "[author name1], [author name2], ... , [author name10] (et al), [year] [article title]. bioRxiv doi: [doi xxx] (preprint)"
- Please update the format of the "Data Availability" section, which should only mention data deposited in external repositories. Please include a general access link to each of the utilized databases (e.g. ProteomeExchange/PRIDE), plus the respective accession codes for the various datasets. Please also remove referee access information now, and ensure that all datasets become publicly accessible at this point.
- Please note that the funding information has to be added as separate entries for each of the funders, and cannot be simply listed as in the Comments box.
- Please note that the use of BioRender should not only be acknowledged in the Acknowledgements section, but also in a separate "Graphics:" paragraph at the end of the Methods section, in the format:
"Some of the graphics/Figure # graphics/synopsis graphics [etc] have been created with BioRender.com"
- Please rename the supplementary movies as Expanded View movies (in-text callouts: "Movie EV1/2/..."). Their legends should be moved out of the Appendix and into individual text files, each of which should be combined with the respective movie file into a separate ZIP file and uploaded as such.
- Please make sure to reference Appendix Table S1 on at least one occasion, e.g. from the Methods section
- Please note that the exact p values need to be provided in the legend of figure 7D
- Please indicate the statistical test used for data analysis in the legends of figures 1G, 2B, 6B
- For the Appendix, please remove all mentions of the obsolete term "supplementary", starting the title page with "Appendix for..." and followed by a proper, itemized Table of Contents including page numbers. Please make sure to reference the contents

throughout as "Appendix Figure S...", "Appendix Table S1", and "Appendix Material and Methods" (for the section now titled "Medicinal Chemistry" - this should also apply to the references in the main text). The NMR data shown this last section should be turned into a proper (multipanel) Appendix Figure S with minimal title/legend, and references as such from the Appendix M&M section. Information on Movies should be removed.

- Finally, please provide suggestions for a short 'blurb' text prefacing and summing up the conceptual aspect of the study in two sentences (max. 250 characters), followed by 3-5 one-sentence 'bullet points' with brief factual statements of key results of the paper; they will form the basis of an editor-written 'Synopsis' accompanying the online version of the article. Please also upload a synopsis image, which can be used as a "visual title" for the synopsis section of your paper. The image should be ideally in JPG format, and please make sure that it remains in the modest dimensions of (exactly) 550 pixels wide and between 300-500 pixels high.

I am returning the manuscript to you for a final round of minor revision, solely to allow you to make these modifications and upload the revised files. Once we will have received them, we should be ready to proceed with formal acceptance and production of the manuscript.

With kind regards,

Hartmut

*** PLEASE NOTE: All revised manuscript are subject to initial checks for completeness and adherence to our formatting guidelines. Revisions may be returned to the authors and delayed in their editorial re-evaluation if they fail to comply to the following requirements. As a first step please read our guidelines for revised submissions:
<https://link.springer.com/journal/44318/submission-guidelines#cms-Revised-submissions>

1) Every manuscript requires a Data Availability section (even if only stating that no deposited datasets are included). Primary datasets or computer code produced in the current study have to be deposited in appropriate public repositories prior to resubmission, and reviewer access details provided in case that public access is not yet allowed.

4) Each main and each Expanded View (EV) figure should be uploaded as individual production-quality files (preferably in .eps, .tif, .jpg formats). For suggestions on figure preparation/layout, please refer to our Figure Preparation Guidelines:
<https://media.springernature.com/original/springer-cms/rest/v1/content/27825798/data/v1>

6) Please complete our Author Checklist, and make sure that information entered into the checklist is also reflected in the manuscript; the checklist will be available to readers as part of the Review Process File.

8) Please note that supplementary information at EMBO Press has been superseded by the 'Expanded View' for inclusion of additional figures, tables, movies or datasets; with up to five EV Figures being typeset and directly accessible in the HTML version of the article.

9) To facilitate reproducibility and cross-laboratory adoption of methodologies, please structure the Materials & Methods section

as outlined in our guide to authors, including a completed Reagents and Tools Table.

10) Digital image enhancement is acceptable practice, as long as it accurately represents the original data and conforms to community standards. If a figure has been subjected to significant electronic manipulation, this must be clearly noted in the figure legend and/or the 'Materials and Methods' section. The editors reserve the right to request original versions of figures and the original images that were used to assemble the figure. Finally, we generally encourage uploading of numerical as well as gel/blot image source data.

In the interest of ensuring the conceptual advance provided by the work, we recommend submitting a revision within 3 months (19th Apr 2026). Please discuss the revision progress ahead of this time with the editor if you require more time to complete the revisions. Use the link below to submit your revision:

Link Not Available

Referee #2:

The authors have addressed all my concerns, and I am pleased to recommend acceptance of the article.

Referee #3:

The authors have adequately addressed all my points, and the manuscript can be recommended for publication in EMBO J.

- Please adjust the order and the headers of the different manuscript sections: Title page with complete author information, Abstract, Keywords, Introduction, Results, Discussion, Methods, Data Availability, Acknowledgements, Disclosure and Competing Interests Statement, References, Main Figure Legends, Tables, Expanded Figure Legends.

We have adjusted the order.

- On the abstract page of the manuscript, please include 4-5 general keyword terms to enhance searchability.

We have included the following keywords: USP9X, DUB inhibitor, Cancer, Ubiquitinomics, Substrate identification.

- As we are switching from a free-text author contribution statement towards a more formal statement based on Contributor Role Taxonomy (CRediT) terms, please remove the present Author Contribution section and instead specify each author's contribution(s) directly in the Author Information page of our submission system during upload of the final manuscript. See <https://casrai.org/credit/> for more information.

We have removed the Author Contribution section and included the information in the submission portal.

- Please carefully go through the reference list and make sure that all references have their complete citation information - including year, journal name & volume, and page/locator numbers - such information is currently missing for several of them.

Also, please adjust the format for citation of preprints as specified in our author guidelines:

The citation in the text should be: "(preprint: [author name1] et al, [year])"

The citation in the reference list: "[author name1], [author name2], ... , [author name10] (et al), [year] [article title]. bioRxiv doi: [doi xxx] (preprint)"

We have updated the reference list. There are no preprints cited in the updated reference list.

- Please update the format of the "Data Availability" section, which should only mention data deposited in external repositories. Please include a general access link to each of the utilized databases (e.g. ProteomeExchange/PRIDE), plus the respective accession codes for the various datasets. Please also remove referee access information now, and ensure that all datasets become publicly accessible at this point.

We have updated the Data Availability section. The PXD accession IDs are listed in the manuscript, and we will make them publically accessible once we have a doi to link it to.

Note: Three PXD access codes are missing in the Data Availability section (marked XXX), we are in the process of obtaining them.

- Please note that the funding information has to be added as separate entries for each of the funders, and cannot be simply listed as in the Comments box.

We have included the information in the submission portal.

- Please note that the use of BioRender should not only be acknowledged in the Acknowledgements section, but also in a separate "Graphics:" paragraph at the end of the Methods section, in the format: "Some of the graphics/Figure # graphics/synopsis graphics [etc] have been created with BioRender.com"

The paragraph "Graphics: Some of the graphics in Fig 3A, Fig 4A and Fig 5B, as well as the synopsis image have been created with BioRender.com." has been included.

- Please rename the supplementary movies as Expanded View movies (in-text callouts: "Movie

EV1/2/..."). Their legends should be moved out of the Appendix and into individual text files, each of which should be combined with the respective movie file into a separate ZIP file and uploaded as such.

We have incorporated the changes as suggested.

- Please make sure to reference Appendix Table S1 on at least one occasion, e.g. from the Methods section

We have included a reference for Appendix Table S1 in the Methods section.

- Please note that the exact p values need to be provided in the legend of figure 7D

The statistics software used reports $p < 0.0001$ and does not report exact values below that. Thus we report $p < 0.0001$ in the figure legend. For Appendix Fig S8E we have provided the exact p values.

- Please indicate the statistical test used for data analysis in the legends of figures 1G, 2B, 6B

The information has been included for the figures mentioned.

- For the Appendix, please remove all mentions of the obsolete term "supplementary", starting the title page with "Appendix for..." and followed by a proper, itemized Table of Contents including page numbers. Please make sure to reference the contents throughout as "Appendix Figure S...", "Appendix Table S1", and "Appendix Material and Methods" (for the section now titled "Medicinal Chemistry" - this should also apply to the references in the main text). The NMR data shown this last section should be turned into a proper (multipanel) Appendix Figure S with minimal title/legend, and references as such from the Appendix M&M section. Information on Movies should be removed.

We have included the changes as suggested. We have included the new Appendix Fig S9A containing the NMR data in the Appendix.

- Finally, please provide suggestions for a short 'blurb' text prefacing and summing up the conceptual aspect of the study in two sentences (max. 250 characters), followed by 3-5 one-sentence 'bullet points' with brief factual statements of key results of the paper; they will form the basis of an editor-written 'Synopsis' accompanying the online version of the article. Please also upload a synopsis image, which can be used as a "visual title" for the synopsis section of your paper. The image should be ideally in JPG format, and please make sure that it remains in the modest dimensions of (exactly) 550 pixels wide and between 300-500 pixels high.

We have included the information in the separate documents "Schenk_SynopsisText.docx" and "Schenk_SynopsisImage.pdf".

General changes to the manuscript:

- Fixed typos and misleading wording
- Updated titles in the Movie files
- Fixed typos in Reagent Table

Prof. David Komander
The Walter and Eliza Hall Institute of Medical Research
Ubiquitin signalling
1G Royal Parade
Parkville
Melbourne, Victoria 3052
Australia

20th Feb 2026

Re: EMBOJ-2025-121586R1
Global analysis of cancer cell responses to USP9X inhibition

Dear David,

Thank you for submitting your final revised manuscript for our consideration. I am pleased to inform you that we have now accepted it for publication in The EMBO Journal.

You may qualify for financial assistance for your publication charges - either via a Springer Nature fully open access agreement or an EMBO initiative. Check your eligibility: <https://link.springer.com/journal/44318/how-to-publish-with-us>

With kind regards,

Hartmut

Please note that it is The EMBO Journal policy for the transcript of the editorial process (containing referee reports and your response letters) to be published as an online supplement to each paper. If you should prefer removal of any referee-only figures included in the point-by-point response(s), e.g. because they may still be used for future publication or because they have been reproduced from published work by others, please do let us know immediately via response email.

More information is available here: <https://link.springer.com/partners/embo-press/editorial-policies#Peer%20review>